# Generative Proto-Sequence: Sequence-Level Decision Making for Long-Horizon Reinforcement Learning

**Netanel Fried**\*                                                                     *friednet@post.bgu.ac.il*
*Department of Computer Science and Information*
*Ben-Gurion University of the Negev*

**Liad Giladi**\*                                                                          *giladili@post.bgu.ac.il*
*Department of Computer Science and Information*
*Ben-Gurion University of the Negev*

**Gilad Katz**                                                                            *giladkz@bgu.ac.il*
*Department of Computer Science and Information*
*Ben-Gurion University of the Negev*

**Reviewed on OpenReview:** *https://openreview.net/forum?id=fSG2DRHtOg*

## Abstract

Deep reinforcement learning (DRL) methods often face challenges in environments characterized by large state spaces, long action horizons, and sparse rewards, where effective exploration and credit assignment are critical. We introduce Generative Proto-Sequence (GPS), a novel generative DRL approach that produces variable-length discrete action sequences. By generating entire action sequences in a single decision rather than selecting individual actions at each timestep, GPS reduces the temporal decision bottleneck that impedes learning in long-horizon tasks. This sequence-level abstraction provides three key advantages: (1) it facilitates more effective credit assignment by directly connecting state observations with the outcomes of complete behavioral patterns; (2) by committing to coherent multi-step strategies, our approach facilitates better exploration of the state space; and (3) it promotes better generalization by learning macro-behaviors that transfer across similar situations rather than memorizing state-specific responses. Evaluations across diverse maze navigation tasks of varying sizes and complexities demonstrate that GPS outperforms leading action repetition and temporal methods in the large majority of tested configurations, where it converges faster and achieves higher success rates.

## 1 Introduction

Deep reinforcement learning (DRL) has demonstrated impressive performance across diverse applications (Mnih et al., 2015; Silver et al., 2016; Levine et al., 2016). However, significant challenges remain when dealing with environments characterized by large state spaces, long-range tasks, and sparse rewards. In such contexts, traditional DRL methods that select actions sequentially often suffer from inefficient exploration and poor credit assignment (Mesnard et al., 2020; Raileanu & Rocktäschel, 2020; Ecoffet et al., 2021), leading to difficulties in learning effective policies for tasks that require coordinated, multi-step strategies. These challenges are further exacerbated by sparse reward signals, whose limited feedback hinders the agent's ability to discover and reinforce successful behaviors (Arjona-Medina et al., 2019; Hung et al., 2019).

Recent research efforts have attempted to address these challenges using diverse techniques such as hierarchical DRL (Kulkarni et al., 2016; Xu et al., 2022), temporal abstraction (Biedenkapp et al., 2021; Zhang et al., 2022b; Saanum et al., 2023; Patel & Siegelmann, 2024), sequence modeling (Chen et al., 2021; Janner et al., 2021; Giladi & Katz, 2023), and action repetition strategies (Srinivas et al., 2017; Sharma et al., 2017; Dabney et al., 2020). By creating sub-tasks or capturing higher-level behavioral patterns (Rosete-Beas et al.,

---

\*Equal contribution

2023; Vezzani et al., 2022; Wang et al., 2023), these techniques aim to reduce decision frequency and enhance learning efficiency in long-horizon tasks. Although these methods offer promising results, they often require careful sub-task design (Ajay et al., 2023), reward shaping (Liu et al., 2022), or complex training procedures (Seo & Abbeel, 2024; Rosete-Beas et al., 2023). While there are temporal abstraction methods that generate multi-step action sequences, they often depend on iterative rollouts, autoregressive decoding, or model-based simulations. These solutions introduce computational overhead and restrict the ability to efficiently generate diverse action sequences. To our knowledge, no method supports the generation of *coherent, variable-length action sequences directly from state observations in a single decision step.*

In this study, we propose Generative Proto-Sequence (GPS), a novel actor-critic architecture capable of producing variable-length action sequences. Instead of actions, our Actor generates a *proto-sequence* embedding, which is then decoded into a discrete action sequence using a Decoder component. The Critic evaluates the state and the entire generated sequence jointly, with gradients flowing from the critic through the Decoder to the Actor, facilitating end-to-end learning of strategic, multi-step action sequences. This design enables the agent to generate and execute complex exploratory behaviors in a single decision, enhancing both generalization and long-horizon credit assignment.

We evaluated GPS on a set of challenging maze environments with varying sizes and configurations, including rooms, corridors, and randomly generated obstacles. Our results demonstrate that GPS learns more efficiently across most configurations, generalizes better to novel maze layouts, and outperforms leading baselines in terms of success rate and convergence speed, particularly in large and complex mazes. Our contributions are as follows:

- We introduce a novel architecture that enables end-to-end generation and evaluation of variable-length discrete action sequences, facilitating improved credit assignment and exploration.

- We demonstrate that producing multi-step action sequences in a single shot can lead to improved generalization and faster convergence in maze navigation tasks, particularly in large and complex configurations.

- We provide empirical results on challenging maze benchmarks, showing improvements over top-performing action repetition and temporal methods baselines in metrics such as convergence speed and success rate. We make our code publicly available[1].

## 2 Related Work

### 2.1 Temporal Abstraction Through Action Repetition

Various approaches in DRL have explored temporal abstraction by repeating primitive actions to extend decision horizons. Earlier works (Srinivas et al., 2017; Sharma et al., 2017) introduced frameworks for dynamic action selection and repetition, though their repetition policies operated independently from chosen actions, limiting strategic development.

DAR (Srinivas et al., 2017) augments discrete action spaces by duplicating each base action with multiple repetition rates. While this expansion can improve learning in environments benefiting from temporal abstraction, it produces an inefficient representation – duplicated actions are treated as unrelated, preventing the agent from exploiting their shared underlying behavior and leading to slower learning and imbalanced trade-offs between coarse and fine control. FiGAR (Sharma et al., 2017) addresses this by decoupling behavior and repetition into two jointly trained policies; however, the repetition policy operates independently from the chosen action, limiting the development of nuanced, action-specific repetition strategies.

The authors of Dabney et al. (2020) proposed an exploration strategy repeating actions for random durations to reduce inefficient dithering. Temporl (Biedenkapp et al., 2021) advanced this by enabling agents to determine both action and repetition duration, improving learning efficiency. However, its hierarchical structure artificially decouples action selection from duration determination. Metelli et al. (2020) instead study action persistence in batch reinforcement learning, treating repetition duration as an environmental

---

[1]Code available at `https://github.com/liadgiladi/Generative-Proto-Sequence`

configuration parameter that modifies the effective control frequency. They introduce Persistent Fitted Q-Iteration (PFQI), which learns value functions for different fixed repetition factors from a single batch of experience and provides theoretical bounds on the performance loss induced by persistence as well as a heuristic for selecting an approximately optimal repetition duration. Other recent work includes multi-frequency control, where Lee et al. (2020) introduce Action-Persistent Actor-Critic (AP-AC) to handle cases where different action dimensions persist for different durations (e.g., repeating leg movements every two steps while repeating arm movements every four steps). Additionally, Sabbioni et al. (2023) introduce the All-Persistence Bellman Operator to enable agents to learn from experiences collected with varying repetition durations simultaneously, by decomposing and bootstrapping across different temporal scales. They provide theoretical guarantees of convergence and empirically demonstrate improved exploration and sample efficiency through their ability to reuse experience collected at any repetition duration. Despite showing promise, these studies share a limitation: *temporal abstraction is achieved solely through simple repetition of primitive actions*, without generating coherent, variable-length action sequences.

## 2.2 Multi-Step Action Sequence Generation

Beyond single-action repetition, several methods focus on generating and partially committing to multi-step action sequences – an approach often called action chunking, where multiple consecutive actions are predicted simultaneously rather than one at a time. The authors of Zhang et al. (2022a) introduced a generative planning method (GPM) that produces multi-step plans. Since GPM is trained by maximizing value, the plans generated from it can be regarded as intentional action sequences to reach high-value states and improve sample efficiency. PrAC (Coad et al., 2022) enables agents to generate n-step plans and commit to them while being predictable, balancing adaptability and control stability. The work of Saanum et al. (2023) incentivizes compressible action sequences by integrating sequence priors, while Patel & Siegelmann (2024) introduced a model-based sequence RL framework (SRL) reducing decision frequency through action chunking. Action chunking approaches have also been explored in imitation learning, where methods like Action Chunking Transformers (ACT) (Zhao et al., 2023) and Diffusion Policy (Chi et al., 2025) generate multi-step behaviors from expert demonstrations.

Despite recent progress, most existing methods for generating multi-step action sequences still face major limitations. Many rely on computationally intensive processes such as iterative rollouts, autoregressive decoding, or model-based simulation, which can be slow and inflexible (Li et al., 2024; Li, 2023; Zhang et al., 2025). For example, methods like PrAC and SRL use learned environment models for both planning and training, adding extra model-based complexity (Kumar et al., 2024; Luo et al., 2024). To stay adaptable, some approaches also use external switching mechanisms or mid-sequence re-planning, as seen in GPM and PrAC. This treats long-term planning as an add-on to a step-by-step framework rather than as a core design principle. As a result, sequence generation and evaluation are often optimized separately, which can lead to poor credit assignment (Dai et al., 2018). One case is the use of handcrafted regularization, such as rewarding shorter or more "compressible" sequences (Saanum et al., 2023). However, when objectives are split in this way, it becomes unclear whether failures come from a bad plan or from breaking the secondary constraint, making end-to-end training harder and reducing stability during execution.

## 2.3 Temporal Abstraction Using Hierarchies and LLMs

Hierarchical methods have advanced multi-action decision-making through skill discovery and sequencing. TACO-RL (Rosete-Beas et al., 2023) learns latent skills from unstructured data for long-horizon tasks. ASPiRe (Xu et al., 2022) accelerated RL by combining specialized skill priors. The work of Vezzani et al. (2022) introduced a skill scheduler sequencing pretrained skills, while SHRL (Wang et al., 2023) combined high-level policies with low-level skills for visual navigation. These approaches improve temporal abstraction by leveraging reusable skills rather than primitive actions.

Recent works have leveraged large language models and value-based reinforcement learning methods with action discretization for action sequence generation. CQN-AS (Seo & Abbeel, 2024) proposed a value-based algorithm learning precise value functions from noisy action sequences. AlphaMaze (Dao & Vu, 2025) improved LLMs' spatial reasoning by combining supervised fine-tuning with policy optimization.

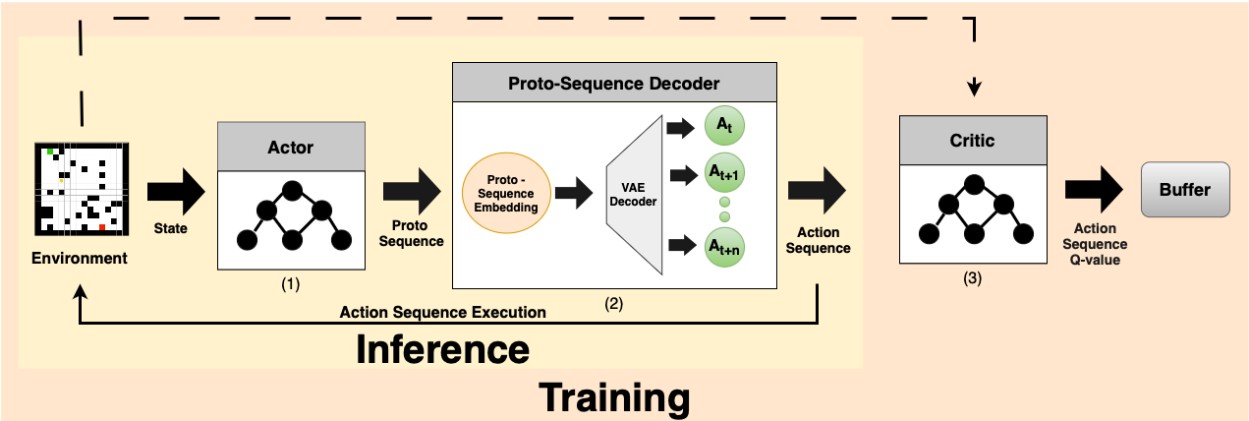

Figure 1: The three components of our proposed approach: (1) The Actor encodes the current state to produce a proto-sequence embedding. (2) The Decoder translates this latent embedding into a variable-length discrete action sequence. (3) The Critic evaluates the state-action-sequence pair and assigns it a Q-value representing the expected cumulative reward (3). During inference, only the actor and decoder components are used.

Our work draws inspiration from Dulac-Arnold et al. (2015), who generated embedding representations of proto-sequences mapped to discrete actions. In GPS, we propose key improvements: our approach is fully differentiable and trainable end-to-end, unlike Dulac-Arnold et al. (2015) whose k-nearest neighbors mapping broke the computation graph. Additionally, by using a VAE-based decoder instead of clustering, we automatically create sequence representations without manual embedding design. This enables efficient generation of coherent, variable-length action sequences that extend beyond simple repetition or skill sequencing.

## 3    Method

**Overview.** Our proposed approach is presented in Figure 1. GPS consists of three components: *Actor*, *Proto-Sequence Decoder (PSD)*, and *Critic*. The Actor receives the current state as input, and produces a *proto-sequence* – an embedding-based representation of a sequence of actions. The PSD receives the proto-sequence as input, and translates it into a discrete set of actions (e.g., $a_t, a_{t+1}, ..., a_{t+L}$), which are then executed sequentially by the agent. Finally, our Critic receives the sequence and predicts the expected cumulative reward obtained from its execution.

GPS differs from previous studies in several important aspects. First, unlike previous studies (Dulac-Arnold et al., 2015), it is end-to-end differentiable and does not require training workarounds. Secondly, our VAE decoder produces more diverse and flexible action sequences than autoregressive or model-based approaches, and also does so in an efficient, one-shot manner. Thirdly, sequence generation and evaluation are learned jointly, without regularization or switching mechanisms, thus improving credit assignment. Finally, by committing to the entire sequence (unlike the frequent re-evaluation of Zhang et al. (2022a)) we reduce execution overhead and increase behavioral predictability by forcing GPS to learn robust policies.

Another important aspect of our proposed approach is its ability to generate action sequences that differ from those on which it was trained. By creating novel sequences, GPS does not simply "memorize" a fixed set of actions, but is able to generalize to larger action spaces. We elaborate on GPS's capacity to produce novel sequences in Section 5.1. Despite using a frozen decoder pre-trained on a limited set of sequences, GPS can generate action sequences that differ from those in the decoder's training data. The ability to generalize is important to our method, since sequence memorization is not applicable to domains with large action spaces. We elaborate on GPS's capacity to produce novel sequences in Section 5.1.

### 3.1 The Actor

The Actor serves as GPS's policy network. Given a state $s_t$, the Actor analyzes the input and outputs a proto-sequence embedding $k = \pi_{\theta^\pi}(s_t)$, where $\theta^\pi$ and $\pi$ are the parameters of the Actor's neural network and the current policy, respectively. The proto-sequence k is a latent embedding of a sequence of actions, represented as a vector in the embedding space $k \in \mathbb{R}^d$, where d is the dimensionality of the embedding space. This representation provides our Actor with significant flexibility, as it can create action sequences of varying length using a fixed-size representation.

The proto-sequence is next used by the PSD to produce a discrete sequence of actions, and this sequence is evaluated by the Critic (Section 3.3). The parameters $\theta^\pi$ of the Actor are then updated using an actor-critic approach analogous to the Deep Deterministic Policy Gradient (DDPG) algorithm (Lillicrap et al., 2016), leveraging the learning signal provided by the Critic. Specifically, the actor's parameters $\theta^\pi$ are adjusted to produce proto-sequence embeddings $k$ that maximize the expected cumulative reward estimated by the critic, $Q_{\theta^Q}(s_t, \mathbf{a})$. This optimization is achieved by updating $\theta^\pi$ to minimize the negative Q-value provided by the critic $-Q_{\theta^Q}(s_t, g_{\theta^\omega}(\pi_{\theta^\pi}(s_t)))$, using backpropagated gradients from the output of the critic network $Q_{\theta^Q}$. These gradients pass through the decoder network $g_{\theta^\omega}$ and subsequently through the actor network $\pi_{\theta^\pi}$, enabling the update of the latter's parameters $\theta^\pi$.

### 3.2 The Proto-Sequence Decoder

The goal of the PSD is to translate the latent proto-sequence $k$ generated by the Actor into a sequence of executable actions in the original action space $\{a_t, a_{t+1}, ..., a_{t+L}\} \in A$. We define the PSD as a function $g_{\theta^\omega} : K \to A'^{L_{max}}$, parameterized by $\theta^\omega$, where A' extends A with an EOS token to handle variable-length sequences within a fixed-length format, padding shorter sequences as needed. This function maps from the latent proto-sequence space $K$ to sequences of fixed length $L_{max}$.

We use a Variational Autoencoder (VAE) (Kingma & Welling, 2013) as our PSD. We train the architecture on a diverse set of synthetic action sequences of varying lengths. For detailed information on the generation process of these sequences, see Appendix M.1. After training, we discard the VAE's encoder and retain only the learned decoder network $g_{\theta^\omega}$. The decoder is integrated into our agent architecture, transforming the Actor's latent proto-sequence embeddings into sequences of discrete actions. GPS will then execute the full sequence, without changes or early stopping. We chose VAE for its efficiency, ability to generate complete sequences in a single step, and its structured latent space that enables smooth interpolation and principled probabilistic modeling.

We pre-train the PSD and keep its parameters fixed while jointly training the actor and critic. We had several reasons for this separation: *a)* simplicity and robustness – separately training the Decoder reduces the number of moving parts in our architecture, enabling faster training. Separate training also prevents a 'moving target' scenario, where multiple components adapt at the same time, compromising training stability (Sutton et al., 1999); *b)* Diversity and prevention of mode collapse: by training the Decoder separately, we have full control of its training set. We use a diverse training set that ensures that the Decoder can generate action sequences of all types. In addition, by freezing the Decoder after its training, we prevent a scenario where this component "forgets" how to generate specific actions during the other components' training; *c)* Transferability – by ensuring our Decoder can generate action sequences of all types, we can train it once and then use it in all tasks with this particular action space. It should be noted that training GPS end-to-end achieves comparable final results, with some differences in path optimality and efficiency (see Section G in the Appendix).

### 3.3 The Critic

The goal of our Critic is similar to the role of the critic in an actor-critic architecture. The Critic receives the current state $s_t$ and the one-hot encoded discrete actions sequence $\mathbf{A} = (a_t, a_{t+1}, \ldots, a_{t+L})$ produced by the PSD. It then attempts to predict $Q_{\theta^Q}(s_t, \mathbf{A})$, which represents the cumulative discounted reward obtained

by executing **A** and following the policy after the end of the sequence:

$$Q_{\theta^Q}(s_t, \mathbf{A}) \approx \mathbb{E}_{\pi, P}\left[\sum_{k=0}^{L-1} \gamma^k r_{t+k} + \gamma^L V^\pi(s_{t+L})\right]$$

where $V^\pi$ is the value function under policy $\pi$, $\theta^Q$ are the Critic's parameters, and $L = eff\_len(\mathbf{A})$ denotes the effective length of the action sequence **A**.

The Critic's parameters $\theta^Q$ are updated by minimizing the Mean Squared Error (MSE) loss against a Temporal Difference (TD) target $y_t$:

$$L(\theta^Q) = \mathbb{E}_{(s_t, \mathbf{A}, \text{rewards}, s_{\text{next}})}\left[(Q(s_t, \mathbf{A}; \theta^Q) - y_t)^2\right]$$

The target $y_t$ is constructed from the sum of discounted rewards $R_t(\mathbf{A})$ obtained by executing sequence **A**, and the discounted value of the subsequent state $s_{t+\text{L}}$, estimated using target Actor ($\text{Actor}_{\text{target}}$) and target Critic ($Q_{\text{target}}$) networks:

$$y_t = R_t(\mathbf{A}) + \gamma^L Q_{\text{target}}(s_{t+L}, \text{PSD}(\text{Actor}_{\text{target}}(s_{t+L})); \theta^{Q-})$$

This update mechanism, which relies on TD errors and target networks, is characteristic of many actor-critic algorithms, and shares similarities with methods such as DDPG (Lillicrap et al., 2016). While the Critic learns to accurately predict $Q(s_t, \mathbf{A}; \theta^Q)$, the Actor is trained to produce proto-sequences that, when decoded by the PSD, maximize this predicted Q-value.

### 3.4 Training Set Augmentation Using Sequence Subsets and Inference

To enhance learning efficiency and improve credit assignment, our training procedure leverages reward information from subsequences of each executed action sequence. For each sequence $\mathbf{A} = (a_t, \ldots, a_{t+L})$ of length $L$, we extract transitions corresponding to multiple contiguous subsequences $(a_i, \ldots, a_j)$ where $t \leq i < j < t + L$. For each such subsequence starting from an intermediate state $s_i$, we calculate the accumulated discounted reward obtained during its execution. This process effectively generates multiple learning samples of varying temporal lengths from a single interaction sequence, enriching the training data.

The subsequence extraction strategies for these state-subsequence-reward tuples, which we add to the replay buffer, include two primary approaches: (1) prefix extraction, which fixes the starting state while varying the end point, and (2) suffix extraction, which fixes the goal state while varying the starting point. This bidirectional approach diversifies the replay buffer with different time scales and enables the Critic to learn value estimates $Q_{\theta^Q}(s_i, (a_i, \ldots, a_j))$ for sequences of different lengths concurrently. As shown in our analysis in Section 5.2, these extraction strategies significantly accelerate learning and improve overall performance. It is important to note that during inference (test time), our architecture does not utilize the Critic component, since no training takes place. Instead, the Actor and PSD produce the action sequence, and the latter is executed in full.

## 4 Experiments and Results

### 4.1 Evaluation Environment

Mazes are a foundational benchmark in DRL research, commonly used to evaluate an agent's ability to perform complex sequential decision-making and navigation tasks. Their structured yet variable environments provide a controlled setting for evaluating generalization, exploration, and memory, which are central to DRL performance (Pašukonis et al., 2023). We use four types of mazes in our evaluation:

- **Empty.** These mazes have no walls or obstacles, except for their boundaries.

- **Sparse Obstacles.** This setup has randomly placed obstacles in K% of the cells of each maze (e.g., 15%).

Table 1: The setup and properties of the mazes used in the evaluation.

| Environment | Dist. from start to goal | Train Set Size | Train Optimal Avg. Path | Val Set Size | Val Optimal Avg. Path | Test Set Size | Test Optimal Avg. Path |
|---|---|---|---|---|---|---|---|
| 8x8 | [1 - 14] | 100 | 5.14 | 100 | 5.49 | 1000 | 5.31 |
| 16x16 | [16 - 26] | 100 | 18.04 | 100 | 18.0 | 1000 | 17.98 |
| 16x16_obstacles_15% | [20 - 30] | 100 | 21.02 | 100 | 21.35 | 210 | 21.31 |
| 16x16_obstacles_25% | [20 - 30] | 100 | 21.63 | 100 | 21.34 | 400 | 21.54 |
| 16x16_rooms | [20 - 30] | 100 | 20.93 | 100 | 21.01 | 585 | 21.02 |
| 16x16_corridors | [10 - 30] | 100 | 12.84 | 100 | 12.76 | 545 | 13.14 |
| 24x24 | [20 - 30] | 100 | 23.39 | 100 | 23.26 | 1000 | 23.56 |
| 24x24_obstacles_15% | [10 - 20] | 100 | 15.04 | 100 | 14.58 | 1000 | 14.73 |
| 24x24_obstacles_25% | [10 - 20] | 100 | 15.8 | 100 | 15.05 | 1000 | 15.11 |

- **Rooms.** This setup consists of four large rooms with small doors between them. We also add randomly placed obstacles in 5% of open cells.

- **Corridors.** These mazes have only narrow corridors for the agent to navigate.

Our evaluation uses fully observable MDPs where the agent has complete visibility of the entire maze grid, including its position, goal location, and all obstacles.

Similarly to Dao & Vu (2025), we use an LLM to produce the code used in our maze generation. Our code, as well as the mazes generated for our evaluation, are available in the appendix. All information on our generated mazes is presented in Table 1. For each maze size and type, the table presents: *a)* the sizes of our training, validation, and test sets, *b)* the range for the distance between the start and goal positions, and *c)* the average length of the optimal path.

## 4.2 Baselines & Evaluated Methods

We evaluate two versions of GPS and three discrete-action baselines: *DQN*, *TempoRL*, and *DAR*. Full implementation details of our approach are included in the Appendix.

**GPS:** Our primary approach generates action sequences using a VAE-based decoder with Gumbel-Softmax sampling. This stochastic mechanism applies a temperature-controlled softmax to produce action distributions that maintain differentiability while approximating discrete samples. The Gumbel-Softmax technique creates a relaxation of categorical distributions that preserves gradients for backpropagation, facilitating end-to-end training of our actor-critic architecture.

**GPS-D:** A deterministic variant of our approach that uses argmax operations with a straight-through estimator in the decoder instead of Gumbel-Softmax sampling. This version produces consistent, deterministic action sequences for each proto-sequence embedding.

**DQN:** Deep Q-Network (Mnih et al., 2013) is a foundational model-free DRL algorithm that learns state-action values. DQN utilizes experience replay and a target network to stabilize its learning.

**DAR:** Dynamic Action Repetition (Srinivas et al., 2017) extends discrete action spaces by repeating original actions at varying rates. DAR enables the agent to select different levels of temporal control, allowing for some action abstraction.

**TempoRL:** Temporal Reinforcement Learning (Biedenkapp et al., 2021) introduces a proactive approach, where the agent selects both an action and its duration. TempoRL employs a hierarchical structure with a behavior policy for action selection and a skip policy for duration, enabling more fine-grained temporal abstraction and efficient exploration.

## 4.3 Experimental Setup

**State and action representations.** We represent the state using a tensor of shape $(N, M, 3)$, where $N$ and $M$ are the height and width of the maze grid. The channels use a similar encoding to that of MiniGridLibrary (Chevalier-Boisvert et al., 2023): *a) Object type*: identifies all environmental elements including walls, empty spaces, agent position, goal location, and starting position; *b) Object color*: provides distinguishing colors

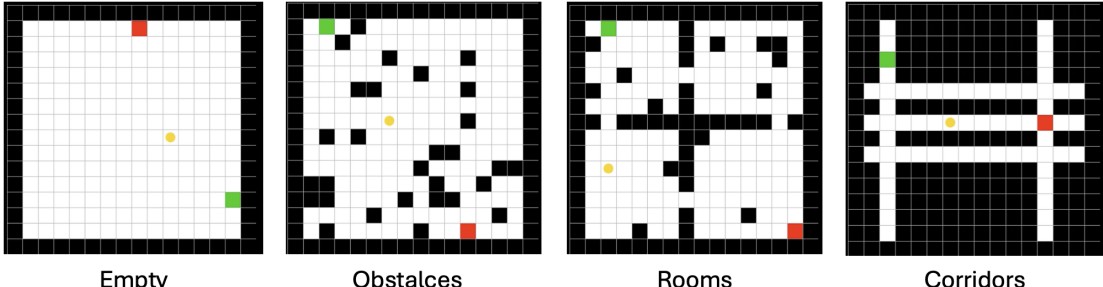

Figure 2: Examples of our generated mazes (16×16). We use four maze environments (left to right): **EMPTY** – open space; **15% Obstacles** – random obstacle placement; **ROOMS** – structured rooms with doorways; **CORRIDORS** – narrow paths requiring precise navigation. Red squares mark start positions, green squares mark goals, and yellow circles show the current agent location.

for the start position, goal location, and current agent position; *c) Placeholder channel*: consistently set to 0, maintaining compatibility with the MiniGrid format. Our discrete environment supports the four basic actions – up, down, left, right – represented as a four-entry one-hot vector

**Reward function.** We define the reward function identically across all methods as follows:

$$R = \begin{cases} 1 & \text{if goal is reached} \\ -1/l_{max} & \text{if a valid action is executed} \\ -3/l_{max} & \text{if an invalid action is executed} \end{cases}$$

where $l_{max}$ is the maximal start-goal distance (see Table 1) acting as a regularizer. When GPS executes a sequence of length L, we sum these Markovian step rewards: $R_{total} = \sum(r_t)$ for $t = 1..L$. This reward structure encourages goal achievement while penalizing excessive steps and invalid actions, ensuring fair comparison across all methods.

**Evaluation metrics.** We use three evaluation metrics:

- **Average success rate (ASR):** the percentage of episodes evaluated where the agent navigates successfully from start to goal position within a predefined number of steps.

- **Path efficiency ratio (PER):** for successfully completed episodes, we calculate the ratio between the episode length and the optimal (minimal) length:

$$PER = \frac{l_{opt}}{l_{episode}}$$

- **Sequence Generation Frequency ($SGF$):** This metric reflects how often the agent generates a new action sequence. It is calculated as the average number of times the Actor is invoked per evaluation episode. Lower values suggest the agent relies on longer-term proto-sequences before needing to generate a new sequence. This metric is relevant to GPS, DAR, and TempoRL baselines.

These metrics are complementary, as they allow us to evaluate the policy's *effectiveness*, *efficiency*, and *decision frequency*, under identical reward optimization across all approaches.

**Neural architecture setup.** All models use a shared CNN feature extractor followed by method-specific linear layers. DQN outputs Q-values for cardinal directions, DAR expands this for multiple repetition rates, and TempoRL implements a branching architecture for action and skip duration. In GPS, actor and

Table 2: ASR Performance at Different Training Steps. Values represent mean ± standard deviation over five seeds.

| Environment | 100k Steps | | | | | 500k Steps | | | | | 1M Steps | | | | | 1.5M Steps | | | | |
|---|---|---|---|---|---|---|---|---|---|---|---|---|---|---|---|---|---|---|---|---|
| | DQN | GPS | GPS-D | TempoRL | DAR | DQN | GPS | GPS-D | TempoRL | DAR | DQN | GPS | GPS-D | TempoRL | DAR | DQN | GPS | GPS-D | TempoRL | DAR |
| 8x8 | 0.75 ± 0.06 | 0.99 ± 0.01 | 0.91 ± 0.03 | 0.82 ± 0.06 | 0.57 ± 0.05 | 0.85 ± 0.06 | 1.00 ± 0.00 | 0.99 ± 0.01 | 0.94 ± 0.03 | 0.73 ± 0.03 | 0.85 ± 0.06 | 1.00 ± 0.00 | 1.00 ± 0.00 | 0.94 ± 0.03 | 0.74 ± 0.04 | - | - | - | - | - |
| 16x16 | 0.19 ± 0.05 | 0.74 ± 0.13 | 0.56 ± 0.11 | 0.36 ± 0.09 | 0.30 ± 0.04 | 0.71 ± 0.08 | 1.00 ± 0.00 | 0.95 ± 0.01 | 0.83 ± 0.07 | 0.62 ± 0.04 | 0.73 ± 0.05 | 1.00 ± 0.00 | 0.97 ± 0.01 | 0.87 ± 0.04 | 0.66 ± 0.04 | 0.74 ± 0.06 | 1.00 ± 0.00 | 0.99 ± 0.01 | 0.88 ± 0.05 | 0.66 ± 0.04 |
| 16x16_obst_15% | 0.08 ± 0.04 | 0.31 ± 0.11 | 0.23 ± 0.11 | 0.06 ± 0.02 | 0.12 ± 0.04 | 0.78 ± 0.04 | 0.76 ± 0.20 | 0.55 ± 0.16 | 0.70 ± 0.04 | 0.44 ± 0.05 | 0.83 ± 0.03 | 0.93 ± 0.05 | 0.76 ± 0.11 | 0.77 ± 0.02 | 0.56 ± 0.06 | 0.84 ± 0.03 | 0.94 ± 0.04 | 0.81 ± 0.07 | 0.80 ± 0.01 | 0.58 ± 0.05 |
| 16x16_obst_25% | 0.03 ± 0.01 | 0.08 ± 0.04 | 0.04 ± 0.03 | 0.03 ± 0.01 | 0.03 ± 0.00 | 0.50 ± 0.13 | 0.41 ± 0.12 | 0.27 ± 0.09 | 0.16 ± 0.06 | 0.04 ± 0.01 | 0.69 ± 0.06 | 0.51 ± 0.13 | 0.40 ± 0.12 | 0.59 ± 0.07 | 0.09 ± 0.03 | 0.72 ± 0.06 | 0.57 ± 0.16 | 0.44 ± 0.15 | 0.73 ± 0.07 | 0.16 ± 0.03 |
| 16x16_rooms | 0.03 ± 0.02 | 0.11 ± 0.05 | 0.05 ± 0.03 | 0.03 ± 0.01 | 0.02 ± 0.00 | 0.55 ± 0.04 | 0.48 ± 0.16 | 0.33 ± 0.13 | 0.42 ± 0.08 | 0.04 ± 0.01 | 0.64 ± 0.04 | 0.80 ± 0.09 | 0.60 ± 0.11 | 0.59 ± 0.06 | 0.09 ± 0.02 | 0.64 ± 0.02 | 0.85 ± 0.07 | 0.66 ± 0.09 | 0.67 ± 0.08 | 0.13 ± 0.05 |
| 16x16_corr | 0.49 ± 0.12 | 0.93 ± 0.06 | 0.77 ± 0.11 | 0.61 ± 0.06 | 0.17 ± 0.03 | 0.80 ± 0.02 | 0.98 ± 0.03 | 0.89 ± 0.05 | 0.89 ± 0.02 | 0.43 ± 0.05 | 0.80 ± 0.02 | 0.98 ± 0.03 | 0.90 ± 0.05 | 0.89 ± 0.03 | 0.61 ± 0.04 | - | - | - | - | - |
| 24x24 | 0.04 ± 0.01 | 0.20 ± 0.09 | 0.13 ± 0.07 | 0.03 ± 0.01 | 0.06 ± 0.01 | 0.14 ± 0.04 | 0.77 ± 0.31 | 0.66 ± 0.39 | 0.28 ± 0.03 | 0.19 ± 0.03 | 0.26 ± 0.04 | 0.88 ± 0.17 | 0.76 ± 0.28 | 0.50 ± 0.04 | 0.26 ± 0.03 | 0.28 ± 0.04 | 0.94 ± 0.10 | 0.80 ± 0.23 | 0.52 ± 0.05 | 0.29 ± 0.03 |
| 24x24_obst_15% | 0.02 ± 0.00 | 0.25 ± 0.09 | 0.11 ± 0.05 | 0.02 ± 0.00 | 0.04 ± 0.01 | 0.10 ± 0.02 | 0.71 ± 0.04 | 0.39 ± 0.05 | 0.06 ± 0.02 | 0.06 ± 0.01 | 0.11 ± 0.01 | 0.79 ± 0.03 | 0.44 ± 0.04 | 0.13 ± 0.02 | 0.09 ± 0.01 | 0.13 ± 0.02 | 0.83 ± 0.01 | 0.46 ± 0.02 | 0.17 ± 0.03 | 0.12 ± 0.02 |
| 24x24_obst_25% | 0.03 ± 0.01 | 0.11 ± 0.04 | 0.04 ± 0.01 | 0.02 ± 0.01 | 0.03 ± 0.01 | 0.07 ± 0.01 | 0.36 ± 0.10 | 0.18 ± 0.05 | 0.03 ± 0.00 | 0.04 ± 0.00 | 0.10 ± 0.01 | 0.46 ± 0.16 | 0.24 ± 0.09 | 0.04 ± 0.01 | 0.04 ± 0.00 | 0.10 ± 0.01 | 0.49 ± 0.16 | 0.27 ± 0.10 | 0.08 ± 0.02 | 0.05 ± 0.01 |

Note: The ASR for each algorithm at specific training step intervals. A gray background indicates the highest ASR achieved for that environment across all steps and algorithms. A yellow background indicates the highest ASR within that specific step interval (excluding any cell already marked gray). '-' indicates unavailable data.

critic networks use separate but identical CNN architectures. The actor produces a 16-dimensional proto-sequence embedding, which the decoder converts into action sequences through a multi-layer network with normalization. Full details are in the appendix.

**Hyperparameters & Hardware.** Unless otherwise noted for specific ablation studies, experiments were conducted using a common set of key hyperparameters, summarized in Tables 14–19 and 22 in the appendix. We selected the values based on preliminary experiments and common practices. All experiments were conducted on a system running Red Hat 5.14 with x86_64 architecture. We used an NVIDIA RTX 2080 GPU with 8GB of VRAM.

**Training Protocol & Model Selection.** We used different training setups based on maze size and type. Detailed step counts are in Table 2. Model selection for final testing used the checkpoint from each run yielding the highest average success rate on a held-out set of validation environments during training. Exploration employed an $\epsilon$-greedy strategy, with random sequences being sampled from the same pool of 400 synthetic sequences used to train the PSD (see Appendix M.1 for generation details). This ensures exploratory sequences follow similar structural patterns to those the decoder was trained to generate. While GPS incurs computational overhead per decision, our analysis shows this is often offset by reduced decision frequency and faster convergence (see Appendix F for detailed trade-offs).

### 4.4 Evaluation Results

#### 4.4.1 Evaluating the Average Success Rate (ASR).

The results of our evaluation are presented in Table 2. GPS outperforms the baselines in the majority of tested maze configurations. Below we present several key observations:

**Ability to learn, converge quickly, and generalize.** GPS demonstrates high sample efficiency and rapid convergence, consistently achieving the highest ASR at 100k steps across all environments. GPS achieves an ASR of 0.74 on the 16×16 empty maze with only 100K training steps, compared to DQN (0.19) and TempoRL (0.36). This early advantage is even more pronounced in larger mazes: in 24×24 with 15% obstacles, GPS reaches 0.25 at 100k steps while all baselines remain below 0.04. GPS also substantially outperforms the baselines with an ASR of 0.77 for 24×24 empty maze at 500k steps, while the top baseline TempoRL only reached 0.28. These results support our hypothesis that modeling action sequences rather than individual actions enables more strategic exploration and faster discovery of goal-reaching strategies. GPS primarily learns to *generate sequences that move the agent in the correct general direction toward goals*, allowing progress in unseen environments even without perfectly optimized paths. The deterministic variant, GPS-D, also shows strong performance, supporting the robustness of the proto-sequence concept.

**Scalability and superior ability to solve complex environments.** The performance gap widens in larger environments. In the empty 24×24 maze, GPS significantly outperforms baselines, achieving an almost perfect ASR of 0.94 at 1.5M steps compared to DQN (0.28) and TempoRL (0.52). In the complex 24×24 environment with 15% obstacles, GPS achieves an ASR of 0.83 after 1.5M steps, significantly outperforming the top performing baselines. Even in the most difficult 24×24 environment with 25% obstacles, GPS

Table 3: Comparative Performance Analysis: Convergence Speed and Efficiency Metrics. Values represent mean ± standard deviation over five seeds.

| Environment | ASR Converge>0.9 Step | | | | | PER | | | | | SGF | | | | |
|---|---|---|---|---|---|---|---|---|---|---|---|---|---|---|---|
| | DQN | GPS | GPS-D | TempoRL | DAR | DQN | GPS | GPS-D | TempoRL | DAR | DQN | GPS | GPS-D | TempoRL | DAR |
| 8x8 | >1M | 100k | 100k | 200k | >1M | 0.99 ± 0.02 | 0.88 ± 0.01 | **0.99 ± 0.00** | 0.80 ± 0.01 | 0.59 ± 0.05 | - | 3.39 ± 0.07 | **2.90 ± 0.07** | 5.37 ± 0.24 | 3.97 ± 0.17 |
| 16x16 | >1.5M | 200k | 400k | >1.5M | >1.5M | 0.95 ± 0.12 | 0.85 ± 0.01 | **0.99 ± 0.00** | 0.94 ± 0.00 | 0.79 ± 0.03 | - | 8.76 ± 0.48 | **7.68 ± 0.64** | 12.73 ± 0.86 | 7.99 ± 0.44 |
| 16x16_obst_15% | >1.5M | 900k | >1.5M | >1.5M | >1.5M | **0.98 ± 0.05** | 0.74 ± 0.02 | 0.96 ± 0.00 | 0.96 ± 0.01 | 0.62 ± 0.05 | - | 14.46 ± 0.82 | 11.05 ± 0.41 | 16.34 ± 2.27 | **10.35 ± 1.96** |
| 16x16_obst_25% | >1.5M | >1.5M | >1.5M | >1.5M | >1.5M | 0.97 ± 0.07 | 0.67 ± 0.01 | N/R | **0.98 ± 0.02** | N/R | - | **17.77 ± 0.87** | N/R | 20.18 ± 1.98 | N/R |
| 16x16_rooms | >1.5M | >1.5M | >1.5M | >1.5M | >1.5M | 0.95 ± 0.12 | 0.68 ± 0.00 | 0.93 ± 0.01 | **0.96 ± 0.01** | N/R | - | 15.96 ± 0.61 | **12.45 ± 2.22** | 17.77 ± 3.92 | N/R |
| 16x16_corr | >1M | 100k | 800k | >1M | >1M | 0.97 ± 0.07 | 0.78 ± 0.03 | **0.97 ± 0.01** | 0.90 ± 0.00 | 0.70 ± 0.04 | - | 9.02 ± 0.90 | 7.23 ± 0.14 | 9.55 ± 0.31 | **6.30 ± 1.69** |
| 24x24 | >1.5M | 1.3M | >1.5M | >1.5M | >1.5M | N/R | 0.78 ± 0.07 | **0.99 ± 0.02** | 0.96 ± 0.00 | N/R | - | 12.95 ± 1.72 | **11.80 ± 4.44** | 16.57 ± 1.16 | N/R |
| 24x24_obst_15% | >1.5M | >1.5M | >1.5M | >1.5M | >1.5M | N/R | 0.59 ± 0.01 | N/R | N/R | N/R | - | **14.47 ± 0.31** | N/R | N/R | N/R |
| 24x24_obst_25% | >1.5M | >1.5M | >1.5M | >1.5M | >1.5M | N/R | N/R | N/R | N/R | N/R | - | N/R | N/R | N/R | N/R |

Note: We present four key performance metrics. The first column shows training steps required to achieve a 90% success rate (lower is better), with highlighted values indicating the fastest convergence. Path Efficiency Ratio (PER) measures trajectory optimality (higher is better, max=1.0), with **bold values** showing best performance. Sequence Generation Frequency (SGF) indicates the average number of decision points needed per episode (lower generally indicates better temporal abstraction). '-' indicates N/A and 'N/R' for ASR < 0.5, indicating insufficient success rate for meaningful PER/SGF evaluation.

maintains a relative advantage, significantly outperforming DQN, the top baseline, with an ASR of 0.49 versus 0.10.

**Performance on medium-sized and structured environments.** In the structured "rooms" environment, GPS achieves an ASR of 0.85 at 1.5M steps, significantly outperforming DQN (0.64) and TempoRL (0.67), while DAR shows a significantly lower score (0.13). This demonstrates GPS's effectiveness in navigating complex structured layouts that require coordinated multi-step strategies. In the challenging 16×16 maze with 15% obstacles, GPS achieves strong performance with an ASR of 0.94, showing significant improvements over DQN (0.84), TempoRL (0.8), and DAR (0.58). In the most challenging evaluation setups with 25% obstacles, GPS achieves an ASR of 0.57 in the 16×16 maze, which is lower than TempoRL (0.73) and DQN (0.72). These results, along with the relatively slower convergence and lower final ASR observed in rooms environment (reaching 0.85 ASR at 1.5M steps compared to near-perfect performance achieved in 100-900k steps in simpler configurations), suggest that extremely dense obstacle distributions and highly constrained spatial layouts can limit the advantages of sequence-level decision making when precise navigation is required. However, our approach significantly outperforms these baselines in the larger 24×24 setup with the same obstacle density, achieving an ASR of 0.49 compared to TempoRL (0.08) and DQN (0.10). These results suggest that the relative performance in dense obstacle configurations is influenced by the interplay between maze scale and obstacle density rather than obstacle density alone.

**Performance characteristics of the baselines.** The baseline methods demonstrate varying strengths across different environments. DQN achieves competitive performance in several cases, particularly in medium-complexity environments like 16x16 with 15% obstacles (0.84), 16x16 with 25% obstacles (0.72) and structured rooms (0.64). TempoRL shows strong performance in specific configurations, excelling in corridors (ASR of 0.89) and achieves solid results in 16x16 obstacle environments (ASR of 0.73-0.80). DAR consistently underperforms, with ASR declining from 0.74 in 8×8 mazes to 0.66 in 16×16 and 0.29 in 24×24 empty mazes. Performance degrades further with obstacles, typically ranging between 0.05–0.16, though it reaches 0.58 in the 16×16 maze with 15% obstacles. The structured "rooms" environment presents similar challenges, with performance at 0.13, suggesting its action repetition strategy is ineffective for complex navigation tasks.

GPS's strong performance stems from operating in the space of action sequences rather than individual actions, enabling more strategic exploration and the discovery of long-horizon rewards that would be difficult to find using single actions or simple repetition methods. Note that for TempoRL and DAR, which might require more extensive training to converge optimally, we observed improved performance with larger training datasets (see Table 4 in the appendix), though computational constraints limited further exploration. While DAR also showed improvements under these conditions, it still achieves relatively lower results in complex structured environments, suggesting that simple action repetition may have inherent limitations for certain maze types.

### 4.4.2 Evaluating the Path Efficiency Ratio (PER).

The results of our evaluation are presented in Table 3. PER is calculated at the final training checkpoint using the total time steps per environment detailed in Table 1. We report PER for GPS, GPS-D, and two baselines introduced in Section 4.2, allowing for direct comparison across methods. A key GPS characteristic is **Self-Correction Through Sequential Decision Points**. GPS can adjust its course at subsequent decision points without requiring an initially perfect action sequence. This sequence-level closed-loop control enables course corrections while retaining the benefits of temporal abstraction (empirical validation of this self-correction capability is provided in Appendix E, demonstrating GPS's ability to generate correction sequences with high success rates (62.4-99.6% immediate correction) across diverse maze environments). Leveraging this capability, GPS adopts a strategy of **Trading Path Efficiency for Robust Navigation**, prioritizing directional correctness over strict path optimality. This approach develops more transferable navigation skills – particularly evident in larger or more obstacle-dense mazes – explaining cases where PER is lower despite higher ASR and faster convergence (see Tables 2 and 3).

GPS-D consistently yields higher PER than GPS across all environments. For example, in the 16×16 empty maze, GPS-D achieves a PER of 0.99 versus GPS's 0.85; in the 24×24 empty maze, GPS-D reaches 0.99 compared to 0.78 for GPS. GPS's Gumbel-Softmax sampling introduces stochasticity that enables broader exploration but can cause path deviations, while GPS-D's deterministic argmax decoder produces more consistent trajectories, trading exploration advantages for improved exploitation. DQN and TempoRL achieve higher PER than GPS in most environments (e.g., DQN reaches 0.98 and TempoRL 0.96 in 16×16 with 15% obstacles, compared to GPS's 0.74), and comparable or slightly lower PER than GPS-D. However, PER is only meaningful when the agent successfully reaches the goal. In many environments where baselines achieve lower ASR, their high path efficiency applies to only a small fraction of episodes. Overall, DAR generally shows lower PER across all environments.

These findings reveal a conceptual trade-off in our approach: GPS's stochasticity boosts exploration and rapid convergence to high ASR, while GPS-D's determinism excels in path efficiency once a good policy is learned. This sequence-level exploration-exploitation trade-off offers practitioners a choice between prioritizing solution discovery (GPS) or execution efficiency (GPS-D) based on their specific requirements.

### 4.4.3 Evaluating Sequence Generation Frequency (SGF).

The results are presented in Table 3, where lower values generally indicate superior temporal abstraction due to fewer policy invocations per episode. For methods reporting SGF, our approaches GPS and GPS-D demonstrate competitive performance across environments. In the 16×16 empty maze, GPS-D achieves an SGF of 7.68, comparable to DAR (7.99) and substantially lower than TempoRL (12.73), while GPS records 8.76. However, in the $16 \times 16$ corridors environment, DAR (6.30) outperforms GPS-D (7.23) and GPS (9.02). Despite this environment-dependent variation, our methods often operate with limited interventions-such as GPS-D's 11.80 SGF in $24 \times 24$ empty mazes versus TempoRL (16.57), demonstrating effective generation of extended proto-sequences.

GPS-D's generally low SGF combined with its high PER indicates capability for efficient, strategic trajectory generation through robust behavioral patterns, making it ideal for scenarios requiring predictable execution or constrained resources. GPS offers a compelling trade-off with competitive PER and favorable SGF compared to TempoRL (e.g., 8.76 vs. 12.73 in $16 \times 16$ empty; 15.96 vs. 17.77 in $16 \times 16$ rooms), alongside faster convergence and higher Average Success Rate as discussed in Section 4.4.1. It balances path efficiency, sequence compactness, and learning speed effectively.

In conclusion, SGF analysis supports our sequence-generation paradigm's effectiveness for temporal abstraction. GPS-D provides efficient, long-term utility with fewer, optimal decisions, while GPS balances competitive SGF, good PER, and rapid ASR. The choice between them depends on application priorities: efficiency vs. predictability or adaptation vs. broader performance.

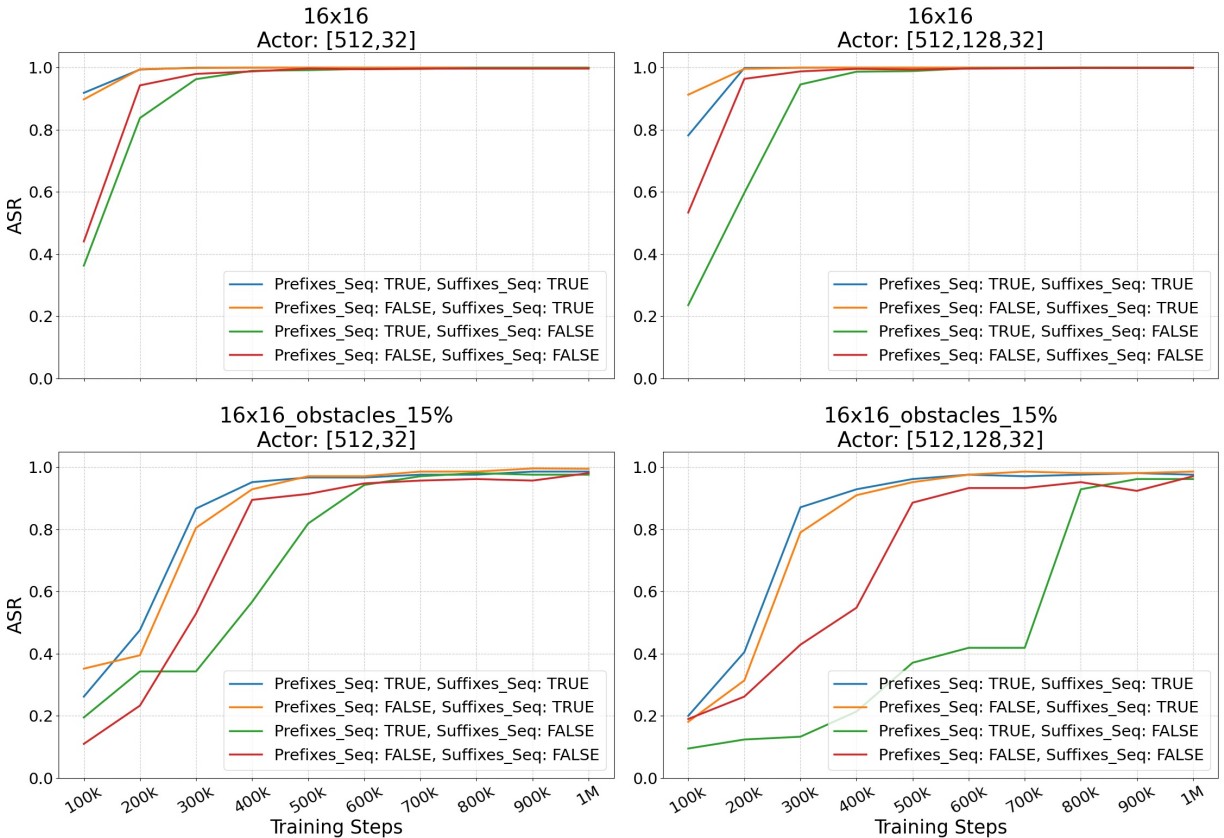

Figure 3: Impact of Subsequence Buffering Strategy on Average Success Rate.

# 5  Analysis and Discussion

## 5.1  Analyzing GPS's Ability to Generate Novel Sequences

While we provided our Decoder with a diverse training set, the latter did not include all possible trajectories. Our reasons were twofold. First, while including all possible action combinations was feasible in our (relatively small) action space, doing the same for larger, more complex action spaces would be infeasible or very costly. secondly, we wanted to evaluate GPS's ability to generalize and produce trajectories that were not in the training set. We consider the ability to generalize important, because the lack of it may limit the usefulness of our approach in large action spaces.

As described in Appendix M.1, the PSD was pre-trained on a set of 400 synthetic sequences generated according to simple, common-sense heuristics for navigation tasks: *a)* each sequence contained at most two distinct action types. *b)* Actions of the same type appeared in contiguous blocks (e.g., "up, up, left" allowed; "up, left, up" disallowed). *c)* No immediately contrasting actions were allowed (e.g., "up, down" prohibited). *d)* Maximum sequence length was capped at $L_{\max}$ (shorter sequences permitted). *e)* Avoidance of loops. After full training, we gathered 15 action sequences by sampling states from the GPS replay buffer and generating the corresponding action sequences through the actor and PSD. Eleven of these did not appear in the PSD's training set and were not fully aligned with at least one of the navigation patterns described above. Using the encoding up→0, down→1, left→2, right→3, the novel sequences were:

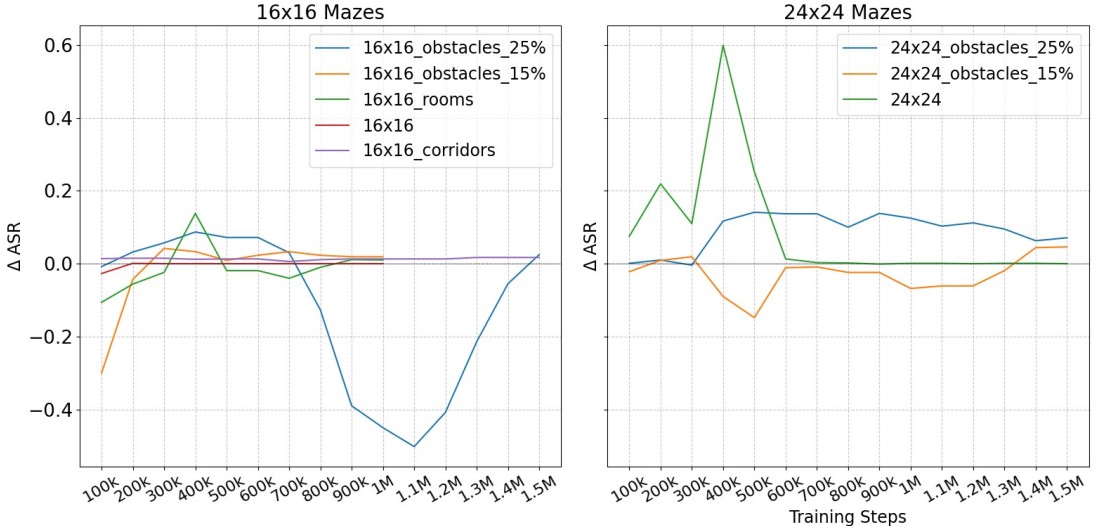

Figure 4: Impact of Actor Network Scaling on Average Success Rate ($\Delta$ ASR) in Mazes.

$$[[1, 2, 1, 1], [1, 1, 0], [3, 1, 0, 1], [0, 0, 3, 1],$$
$$[3, 3, 2], [3, 0, 3], [1, 1, 2, 3], [2, 0, 1],$$
$$[0, 0, 3, 3, 0, 3], [3, 3, 1, 2], [2, 1, 1, 3]]$$

These results show that GPS can create new action sequences not seen during training because it works in a structured embedding space. In this space, sequences with similar structures are grouped together, making it possible to blend known patterns and generate new ones, as shown in Figure 6 in the appendix.

## 5.2 Sequence Subsets Augmentation

We investigate our subsequence buffering approach (Section 3.4), implemented through *prefixes* (fixed start, varying end point) and *suffixes* (fixed goal state, varying starting point). Figure 3 shows that the baseline without subsequence buffering (red) consistently learns most slowly and often converges sub-optimally, while all subsequence buffering variants substantially improve learning efficiency. Using prefixes and suffixes simultaneously (blue) generally produces the most rapid learning, though the suffix-only configuration (orange) performs nearly as well, suggesting backward sampling provides particularly valuable learning signals. The prefix-only approach (green) typically shows slower convergence than other subsequence methods. These performance patterns remain consistent across different maze structures and actor networks.

## 5.3 Performance Under Stochastic Dynamics

Up to this point, we conducted our evaluation in a deterministic setting. We now evaluate GPS's robustness in two stochastic settings, which are more challenging to our approach because of its "commitment" to fully implementing its generated trajectories:

- **"Sticky" actions.** We used the well established "sticky actions" mechanism (Dabney et al., 2020), with 25% probability of repeating the previous action instead of executing the planned one.

- **Random actions.** In this setup, which is also common in the literature (Liu et al., 2024), each action has a 25% probability of being replaces with a random action.

These two setups introduce temporal correlations and execution uncertainty that challenge sequence-based methods, as errors can potentially cascade throughout multi-step sequences.

Results for the sticky actions setup show that GPS maintains strong performance under the sticky actions setup, achieving near-perfect or perfect success rates across most tested environments while preserving competitive convergence times and reasonable path efficiency. In the random actions setup, which proved to be more challenging to all evaluated algorithms (see Appendix C for details), GPS is again the top performer. Moreover, while all evaluated algorithms suffer from a degradation in their performance in this setup, GPS's relative degradation is the smallest. This result is noteworthy, since we would expect DQN, with its single actions, to be the most robust.

The robust performance under stochastic conditions provides evidence that GPS's sequence generation approach may confer resilience beyond deterministic settings. Operating at the sequence level appears to offer some natural buffering against action execution uncertainties, though comprehensive evaluation across diverse stochastic environments would be needed to fully establish this robustness. Detailed experimental details and quantitative results supporting this analysis are presented in Appendix C.

### 5.4 Performance Under Partial Observability

Beyond stochastic dynamics, we evaluated GPS's robustness under partial observability constraints, where agents only perceive a $7 \times 7$ local view window centered on their current position rather than the full maze structure. This setting is particularly challenging for sequence-based methods as it requires generating multi-step action sequences without complete environmental information, testing whether our approach can maintain effective navigation strategies when operating with limited sensory input.

The results reveal environment-dependent performance patterns under partial observability. In empty and corridor environments, GPS demonstrates strong performance, maintaining advantages over DQN in empty 16×16 maze at convergence (0.71 vs 0.60 ASR at 1M steps). Corridor environment show more pronounced advantages, with GPS achieving 3× improvement over DQN (0.79 vs 0.26 ASR at 1M steps), suggesting that multi-step sequences provide inherent benefits for maintaining directional consistency despite limited visibility. However, moderate path efficiency ratios (PER 0.57-0.61) indicate suboptimal navigation compared to DQN's near-perfect efficiency, suggesting that while GPS reaches goals successfully, the generated sequences include unnecessary detours when operating with incomplete information. Notably, in larger 24×24 empty maze, performance converges more closely between methods (0.31 vs 0.30 ASR). This convergence reflects the compounding challenge of limited visibility over longer navigation distances – while the $7 \times 7$ observation window covers a reasonable fraction of $16 \times 16$ mazes, it represents a much smaller proportional view in $24 \times 24$ environments, diminishing GPS's directional consistency advantages as both methods struggle with the fundamentally limited information available for long-distance navigation.

In obstacle-dense and rooms environments, GPS faces increased challenges despite consistently outperforming DQN. In $16 \times 16$ maze with 15% obstacles, both methods struggle to achieve high success rates (GPS: 0.34 ASR, DQN: 0.29 ASR at 1.5M steps), while performance gaps widen in denser configurations (25% obstacles: 0.34 vs 0.21; rooms: 0.45 vs 0.28). These results represent large degradation compared to GPS's full observability performance in most configurations (typically achieving ASR $\geq 0.90$), indicating that partial observability fundamentally challenges sequence-based planning. We hypothesize these difficulties stem from GPS generating multi-step sequences without seeing upcoming barriers or doorways, committing to trajectories that encounter unseen obstacles. While DQN's step-by-step approach theoretically allows immediate incorporation of newly revealed features, its poor performance suggests that reactive single-action decisions may be insufficient for effective exploration under limited visibility. The elevated Sequence Generation Frequency under partial observability reflects GPS's adaptive response, with the proportional increase growing substantially larger in $24 \times 24$ environments, demonstrating that the impact of partial observability on sequence effectiveness scales disproportionately with environment size. No configuration reaches 0.9 ASR within 1.5M steps, indicating substantial sample complexity increases compared to full observability settings, where convergence typically occurs within up to 1M steps. Complete experimental analysis is provided in Appendix D.

### 5.5 Impact of Actor Network Scaling

We examine the impact of actor network size (small: two-layer (512, 32); large: three-layer (512, 128, 32)) on maze navigation performance, measured by $\Delta$ASR (Large - Small) (see Figure 4). In simpler 16x16 mazes (empty or corridor), both architectures performed similarly. With 15% obstacles, the smaller network initially outperformed ($\Delta$ASR $\approx -0.3$ at 100K steps) before convergence at $\sim$300K steps. In denser 25% obstacles, the smaller network outperformed from $\sim$800K steps, peaking at $\Delta$ASR $\approx -0.5$ at 1.1M steps. In larger 24x24 mazes with 15% obstacles, the small network generally led, despite the large network's brief advantage ($\sim$500K steps). However, in the most complex 25% obstacles maze, the large network outperformed throughout training, maintaining $\Delta$ASR between 0.1-0.15. The 24x24 empty maze showed fluctuating performance with occasional spikes for the larger network around 200K and 500K steps.

These results suggest a trade-off: smaller networks suffice or excel in smaller or moderately complex environments (possibly due to better regularization or more stable sequence generation learning), while larger networks demonstrate clear benefits in more complex environments.

### 5.6 Computational Complexity and Runtime

GPS introduces additional runtime overhead compared to baselines, as its Actor-Decoder pipeline makes each decision 1.6–3.3x slower in wall-clock time. However, because GPS operates at a sequence level and makes decisions less frequently, the effective runtime per episode is comparable to baselines in moderately complex settings (e.g., 16x16 mazes). In larger, more difficult environments, GPS shows higher normalized costs, but this is partly because it continues to solve tasks where baselines fail. Despite the per-decision slowdown, GPS converges to high success rates much faster overall (90% in 3.3 hours vs. DQN's 5.83 hours), demonstrating that its superior sample efficiency compensates for runtime overhead in practice. This makes GPS more useful in challenging tasks where training effectiveness outweighs raw inference speed. A comprehensive analysis is presented in Section F in the Appendix.

## 6 Conclusions, Limitations, and Future Work

GPS is a novel actor-critic method that generates variable-length action sequences in a single step. GPS maps state observations to proto-sequences, which are decoded into discrete action sequences. This approach enhances credit assignment and exploration in long-horizon tasks by moving beyond sequential single-action selection. Our evaluation on maze navigation tasks shows GPS surpasses leading action repetition and temporal methods across the majority of tested configurations, achieving higher success rates and faster convergence in environments ranging from simple open spaces to large complex mazes. Performance patterns across different maze scales reveal that GPS's sequence-level approach is particularly advantageous in larger environments, where it achieves considerably higher success rates than baselines even with dense obstacles.

Although our approach shows benefits, particularly in complex environments, several limitations should be acknowledged. First, a new PSD needs to be trained for each unique action space, which adds to the complexity of our approach. GPS has not been evaluated on large action spaces, so adaptations to the decoder component may be needed. The results reveal that performance is influenced by the interaction between environment scale and obstacle density. High-density random obstacles require fine-grained navigation that can limit the advantages of sequence-level decision making. In medium-sized environments with very dense obstacles, this precision requirement offsets GPS's other advantages. In larger environments, however, GPS's strengths outweigh this difficulty, maintaining significant performance advantages over the baselines. Finally, we have not yet adapted GPS to continuous action spaces.

Another important aspect of our approach is its suitability for domains with varying requirements. While GPS outperforms the baselines in the large majority of tested configurations and is able to generate solutions rapidly, the resulting solutions are not always strictly optimal. This trade-off between solution quality and computational efficiency is a well-documented phenomenon in deep reinforcement learning, where methods that prioritize rapid inference may forgo perfect optimality in favor of operational practicality and responsiveness. Similar observations have been reported in recent DRL research (Sohaib et al., 2025; Wu et al., 2023),

highlighting how faster-converging methods frequently produce solutions that, while feasible and effective in real-world applications, may not match the optimality of simpler, slower approaches such as DQN.

Future work will focus on extending to more complex domains, such as those with larger action spaces or continuous control. As part of this research direction, we plan to explore advanced initialization strategies for our Decoder, so that our approach can more efficiently explore large action spaces. Additionally, we will aim to enhance GPS's effectiveness in partial information and stochastic settings, as well as in environments requiring precise local navigation. To this end, we are considering the creation of a re-planning component, and mechanisms that will adapt the maximal length of the generated sequences based on both available information and local environment complexity. Finally, we will explore modifications to our approach that will enable us to define desired trade-offs between path optimality and success. By doing so, our aim is to automatically adapt GPS's strategy to match the requirements of various domains.

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

# A    Analysis of Reward Strategy Impact on ASR

This appendix details the comparison of two step penalty strategies, illustrated in Figure 5. The strategies are the default Max-Level-based penalty ($-1/l_{\max}$, where $l_{\max}$ is a normalization factor related to task depth, e.g., $l_{\max} \approx 30$) and an alternative Map-Size-based penalty ($-1/$maze size, e.g., $-1/256$ for a $16 \times 16$ maze). Their relative efficacy is measured by $\Delta\text{ASR} = \text{ASR}_{\text{Max-Level}} - \text{ASR}_{\text{Map-Size}}$, where positive values indicate superior performance for the Max-Level strategy.

Figure 5 reveals distinct performance patterns across the tested environments.

- In simple $16 \times 16$ mazes, the Max-Level strategy provides a substantial initial learning speedup ($\Delta\text{ASR} \approx +0.8$ at 200k steps), although its final Average Success Rate (ASR) is matched by the Map-Size strategy after approximately 400k training steps.

- When 15% obstacles are introduced in the $16 \times 16$ maze, increasing its complexity, the Max-Level strategy maintains a consistent performance advantage throughout the training. $\Delta\text{ASR}$ peaks at approximately $+0.44$ and remains positive (settling around $+0.1$).

- In larger $24 \times 24$ mazes, the Max-Level strategy's superiority becomes more pronounced. $\Delta\text{ASR}$ dramatically increases after 500k steps, reaching and sustaining a value of approximately $+0.9$. This highlights the diminishing effectiveness of the Map-Size penalty (e.g., $-1/576$ for $24 \times 24$) as it becomes increasingly diluted in larger state spaces.

The superior performance of the Max-Level strategy, particularly in more complex or larger environments, can be attributed to several factors. Firstly, it provides a more **impactful and relevant penalty signal**. The $l_{\max}$-normalized penalty (e.g., $\approx -1/30$) provides stronger and more consistent learning feedback than the Map-Size penalty, which drops off as the maze size increases. Secondly, $l_{\max}$ serves as a normalization factor that likely **correlates better with the intrinsic task difficulty** and typical solution length than the raw cell count of the maze, which does not inherently capture navigational complexity. Consequently, the Max-Level penalty structure appears to offer more **effective exploration guidance** and promotes **greater learning efficiency**.

In summary, normalizing step penalties by $l_{\max}$ (Max-Level-based strategy) leads to a more robust and effective reward scheme for the navigation tasks studied. This approach fosters more efficient learning and achieves higher success rates by aligning the penalty signal more accurately with the inherent challenges of the environment, proving especially advantageous as task complexity and scale increase.

# B    Evaluating the Average Success Rate (ASR) With Larger Train dataset

To assess performance on a larger dataset, we trained the agent on 2000 mazes. Table 4 presents the Average Success Rates (ASR) across various maze configurations.

Table 4: Average Success Rates (ASR) with 2000 mazes in train dataset across maze types.

| Maze Type | Max Steps | Train Size | Dist. from Start to Goal | DQN | GPS | GPS-D | TempoRL | DAR |
|---|---|---|---|---|---|---|---|---|
| 8x8 | 1M | 2000 | $[1 - 14]$ | 1.00 | 1.00 | 1.00 | 1.00 | 0.81 |
| 16x16 | 1M | 2000 | $[16 - 26]$ | 1.00 | 1.00 | 1.00 | 1.00 | 0.89 |
| 16x16_obst_15% | 1.5M | 2000 | $[20 - 30]$ | 0.94 | 0.99 | 0.94 | 0.91 | 0.78 |
| 16x16_obst_25% | 1.5M | 2000 | $[20 - 30]$ | 0.98 | 0.98 | 0.94 | 0.39 | 0.16 |
| 16x16_rooms | 1.5M | 2000 | $[20 - 30]$ | 1.00 | 0.95 | 0.82 | 0.98 | 0.13 |
| 16x16_corr | 1M | 2000 | $[10 - 30]$ | 1.00 | 1.00 | 1.00 | 1.00 | 0.67 |
| 24x24 | 1.5M | 2000 | $[20 - 30]$ | 0.98 | 1.00 | 0.98 | 1.00 | 0.75 |
| 24x24_obst_15% | 1.5M | 2000 | $[10 - 30]$ | 0.05 | 0.09 | 0.05 | 0.03 | 0.13 |

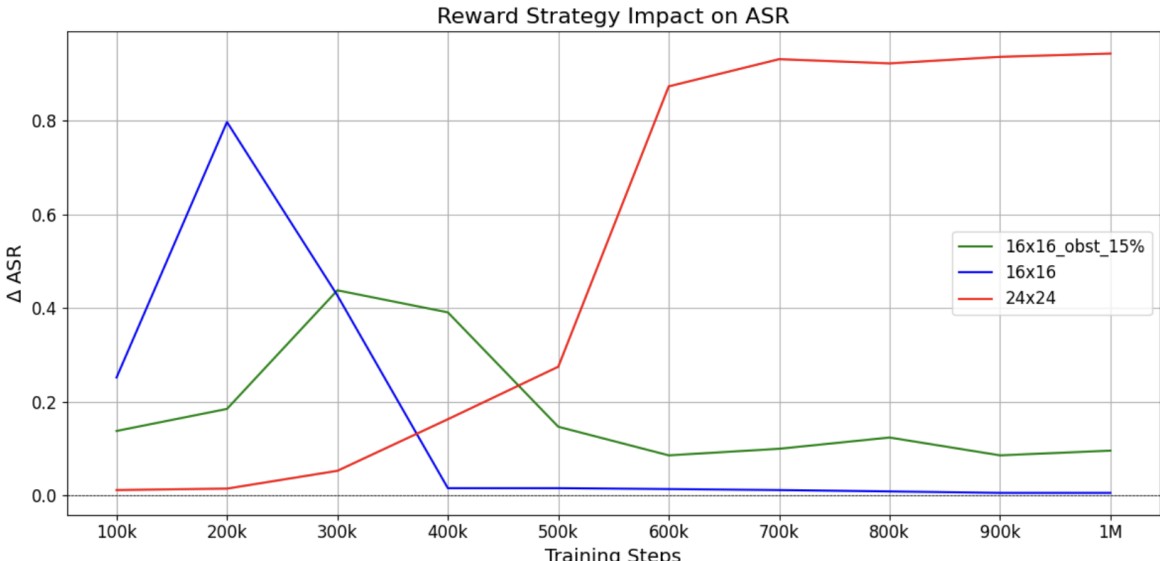

Figure 5: Comparison of average success rate differences ($\Delta$ASR) between two reward strategies: **Max-Level** (the default strategy) and **Map-Size**, which is identical except that it replaces $l_{\max}$ with the total number of cells in the maze (e.g., 256 for a 16×16 maze). The values are evaluated across training steps. A positive $\Delta$ASR indicates that the Max-Level reward strategy yields better performance.

## C  Evaluation on Stochastic Environments

Real-world environments feature inherent stochasticity through action execution noise, sensor uncertainty, and environmental dynamics. To evaluate GPS's robustness under such conditions, we extended our evaluation framework to include two types of stochastic dynamics:

- **"Sticky" actions.** We implemented a setup where there is a 25% probability that the agent executes the previous action in the sequence instead of the current planned action. This "sticky actions" mechanism follows established protocols in reinforcement learning evaluation (Machado et al., 2018; Dabney et al., 2020), introducing temporal correlations and systematic execution biases that create realistic uncertainty and test an agent's ability to adapt to imperfect action execution.

- **Random actions.** In this setup each action has a 25% probability of being replaces with a random action. These setups are also common in the literature, where they are used to simulate adversarial environments (Liu et al., 2024), or environments with noisy signals (Sutton et al., 1999).

These two setups present unique challenges for sequence-based methods like GPS. Since our approach commits to entire action sequences, execution errors can potentially cascade throughout the sequence, leading to significant deviations from intended trajectories. This makes stochastic environments a critical test for the robustness of our sequence generation paradigm.

### C.1  Sticky Actions Results

Table 5 presents GPS performance results under the "sticky" stochastic conditions across our maze configurations, comparing GPS under deterministic conditions versus GPS with sticky actions (GPS-SA) under stochastic conditions. The results demonstrate that GPS exhibits remarkable resilience to stochastic dynamics. Despite the 25% action execution uncertainty, GPS-SA achieves near-perfect or perfect success rates in most environments, reaching ASR = 1.00 in empty mazes (8×8, 16×16, and 24×24) and maintaining strong performance (ASR = 0.95) even in the challenging 16×16 maze with 15% obstacles, and comparable

Table 5: GPS Performance on Stochastic Maze Environments (25% Sticky Actions). GPS-SA denotes GPS with sticky actions. Both GPS and GPS-SA results are based on single-seed experiment.

| Environment | ASR at Training Steps | | | | | | | | ASR Converge >0.9 Step | | PER | | SGF | |
| | 100K | | 500K | | 1M | | 1.5M | | | | | | | |
| | GPS | GPS-SA | GPS | GPS-SA | GPS | GPS-SA | GPS | GPS-SA | GPS | GPS-SA | GPS | GPS-SA | GPS | GPS-SA |
|---|---|---|---|---|---|---|---|---|---|---|---|---|---|---|
| 8x8 | 1.00 | 0.99 | 1.00 | 1.00 | 1.00 | 1.00 | - | - | 100k | 100k | 0.89 | 0.71 | 3.30 | 5.45 |
| 16x16 | 0.71 | 0.77 | 1.00 | 1.00 | 1.00 | 1.00 | 1.00 | 1.00 | 300k | 300k | 0.85 | 0.80 | 8.99 | 8.83 |
| 16x16_obstacles_15% | 0.43 | 0.33 | 0.92 | 0.90 | 0.92 | 0.94 | 0.92 | 0.95 | 300k | 600k | 0.75 | 0.69 | 14.35 | 16.26 |
| 16x16_obstacles_25% | 0.05 | 0.04 | 0.37 | 0.22 | 0.47 | 0.46 | 0.48 | 0.47 | >1.5M | >1.5M | 0.68 | 0.61 | 18.47 | 19.41 |
| 16x16_rooms | 0.21 | 0.06 | 0.72 | 0.34 | 0.91 | 0.54 | 0.91 | 0.81 | 700k | >1.5M | 0.68 | 0.60 | 16.27 | 19.84 |
| 16x16_corridors | 0.93 | 0.95 | 0.99 | 0.99 | 0.99 | 0.99 | - | - | 100k | 100k | 0.79 | 0.71 | 9.11 | 10.43 |
| 24x24 | 0.28 | 0.13 | 1.00 | 1.00 | 1.00 | 1.00 | 1.00 | 1.00 | 200k | 300k | 0.84 | 0.78 | 11.39 | 12.42 |
| 24x24_obstacles_15% | 0.20 | 0.14 | 0.66 | 0.46 | 0.74 | 0.61 | 0.81 | 0.61 | >1.5M | >1.5M | 0.58 | 0.53 | 14.64 | 15.67 |
| 24x24_obstacles_25% | 0.14 | 0.13 | 0.36 | 0.29 | 0.46 | 0.38 | 0.51 | 0.40 | >1.5M | >1.5M | 0.55 | 0.48 | 17.22 | 18.72 |

Table 6: GPS Performance on Stochastic Maze Environments (25% Uniform Random Actions). Both GPS and DQN results are based on single-seed experiment.

| Environment | ASR at Training Steps | | | | | | | | ASR Converge >0.9 Step | | PER | | SGF | |
| | 100K | | 500K | | 1M | | 1.5M | | | | | | | |
| | GPS | DQN | GPS | DQN | GPS | DQN | GPS | DQN | GPS | DQN | GPS | DQN | GPS | DQN |
|---|---|---|---|---|---|---|---|---|---|---|---|---|---|---|
| 8x8 | 0.98 | 0.74 | 0.99 | 0.86 | 1.0 | 0.86 | - | - | 100k | >1M | 0.51 | 0.66 | 7.42 | - |
| 16x16 | 0.39 | 0.10 | 0.79 | 0.53 | 0.91 | 0.59 | 0.91 | 0.60 | 900k | >1.5M | 0.54 | 0.71 | 14.14 | - |
| 16x16_obstacles_15% | 0.11 | 0.05 | 0.35 | 0.58 | 0.63 | 0.66 | 0.83 | 0.69 | >1.5M | >1.5M | 0.52 | 0.72 | 20.85 | - |
| 16x16_obstacles_25% | 0.04 | 0.02 | 0.11 | 0.23 | 0.13 | 0.59 | 0.16 | 0.62 | >1.5M | >1.5M | 0.49 | 0.68 | 20.96 | - |
| 16x16_rooms | 0.06 | 0.03 | 0.13 | 0.43 | 0.27 | 0.52 | 0.32 | 0.52 | >1.5M | >1.5M | 0.47 | 0.66 | 24.29 | - |
| 16x16_corridors | 0.71 | 0.34 | 0.98 | 0.8 | 0.98 | 0.81 | - | - | 200k | >1M | 0.55 | 0.72 | 13.34 | - |
| 24x24 | 0.08 | 0.07 | 0.35 | 0.09 | 0.72 | 0.19 | 0.72 | 0.20 | >1.5M | >1.5M | 0.54 | 0.71 | 19.33 | - |
| 24x24_obstacles_15% | 0.19 | 0.04 | 0.28 | 0.12 | 0.33 | 0.16 | 0.47 | 0.16 | >1.5M | >1.5M | 0.43 | 0.56 | 20.05 | - |
| 24x24_obstacles_25% | 0.09 | 0.03 | 0.17 | 0.08 | 0.21 | 0.14 | 0.21 | 0.16 | >1.5M | >1.5M | 0.42 | 0.57 | 21.97 | - |

results (ASR = 0.61, ASR = 0.40) in 24x24 obstacles mazes. Comparing GPS performance across conditions reveals execution uncertainty's impact: while GPS achieves rapid convergence in deterministic settings, sticky actions extend training requirements. The 24×24 empty maze exemplifies this trade-off: convergence time triples from 200K to 300K steps under sticky actions, yet final ASR remains perfect (1.00) in both conditions. Path Efficiency Ratio shows moderate degradation from deterministic to stochastic conditions (0.84 to 0.78), indicating reduced optimality while maintaining effective navigation. Obstacle-dense environments face more pronounced challenges, with the 16×16 maze requiring doubled convergence time (300K to 600K steps). Although the PER reduction is comparable to that in the 24×24 empty maze, the geometry exposes and amplifies the impact of stochasticity – where the same level of degradation leads to much slower convergence and higher SGF – as execution errors in constrained spaces increase collision likelihood.

The Path Efficiency Ratio (PER) analysis reveals nuanced navigation behavior under stochastic conditions. GPS-SA maintains PER values ranging from 0.48 to 0.80 across environments, demonstrating reasonably efficient pathfinding despite 25% action execution uncertainty. While PER degradation from deterministic to stochastic conditions is relatively uniform across environment types (approximately 0.05-0.10 reduction: e.g., 0.84→0.78 in 24×24 empty, 0.75→0.69 in 16×16 with 15% obstacles, 0.68→0.60 in 16×16 rooms), these moderate declines indicate GPS successfully maintains goal-directed behavior while accommodating stochastic perturbations, avoiding catastrophic navigation failures even when execution reliability is compromised.

The Sequence Generation Frequency (SGF) reveals how GPS responds to execution uncertainty. Under sticky actions, SGF increases across most environments, indicating more frequent sequence generation to handle execution errors. The absolute increase varies by environment without a clear pattern: some simple mazes show notable increases (8×8: +2.15 from 3.30 to 5.45), while others show minimal change (24×24 empty: +1.03 from 11.39 to 12.42), and complex environments similarly show variable increases (16×16 with 15% obstacles: +1.91 from 14.35 to 16.26; 16×16 rooms: +3.57 from 16.27 to 19.84). This variability suggests that the impact of execution uncertainty on decision frequency depends on multiple factors beyond simple environment complexity.

Notably, GPS-SA's performance under stochastic dynamics surpasses baseline methods operating in deterministic environments. GPS-SA achieves perfect ASR (1.00) in the 24×24 maze at 500K training steps, while DQN reaches only 0.26 ASR even at 1.5M steps under deterministic conditions (see Table 2). Similarly, in the 16×16 maze with 15% obstacles, GPS-SA maintains ASR = 0.95 at 1.5M steps despite sticky actions, exceeding DQN's deterministic performance of 0.85, TempoRL's 0.80, and DAR's 0.84. The exception is the 16×16 maze with 25% obstacles, where both GPS (0.48) and GPS-SA (0.47) achieve lower final ASR than TempoRL (0.74) and DQN (0.73) under deterministic conditions, indicating that extremely dense obstacle configurations challenge the sequence-level decision-making paradigm under both deterministic and stochastic conditions. This cross-condition superiority – where GPS under adversarial stochasticity outperforms baselines under ideal deterministic execution – obviates the need for baseline evaluation under sticky actions, as these methods already struggle in the simpler setting.

## C.2 Random Actions Results

The stochastic setup of random actions presents a potential larger challenge to our proposed approach. Although this setup has the same probability as the previous one of injecting an unintended action into the sequence, these actions may be more diverse (and potentially disruptive) than those of the previous setup. For example, an optimal trajectory for a given maze may include only "up" and "right" actions; in the sticky setup, the injected actions may only include these two actions. In the random setup, every action – including those not included in the optimal trajectory – is equally possible.

Table 6 presents our evaluation results comparing GPS to DQN, the baseline most likely to be robust against random action injection due to its single-action-at-a-time decision-making. The results show that both algorithms achieve reduced performance in this stochastic setup: in all evaluated environments except 8×8 (the smallest and easiest), both algorithms underperform compared to their non-stochastic results. GPS maintains advantages across environments of varying complexity. In simple mazes, GPS achieves higher final ASR: 8×8 (1.00 vs 0.86), 16×16 empty (0.91 vs 0.60), and 16×16 corridors (0.98 vs 0.81). In larger mazes, GPS shows pronounced advantages: 24×24 empty (0.72 vs 0.20) and 24×24 with 15% obstacles (0.47 vs 0.16). The exceptions are highly constrained environments with dense obstacles, where DQN's reactive single-action approach proves more resilient: 16×16 with 25% obstacles (0.16 vs 0.62) and 16×16 rooms (0.32 vs 0.52). GPS maintains moderate advantages in obstacle environments with lower density (16×16 with 15% obstacles: 0.83 vs 0.69).

The Path Efficiency Ratio (PER) analysis reveals that uniform random action injection causes more severe and uniform degradation compared to sticky actions. PER values under uniform random conditions cluster tightly between 0.42-0.55 across all environments, representing drops of 0.06-0.26 from sticky action performance. While PER under sticky actions varied by environment (0.48-0.80 range), uniform random perturbations compress nearly all environments to approximately 0.50 PER. When both algorithms succeed, DQN achieves consistently higher PER (0.56-0.72) than GPS (0.42-0.55), indicating more optimal paths but lower success rates overall. This uniform degradation pattern reflects how unpredictable disruptions systematically undermine temporal abstraction of sequence coherence: sticky perturbations preserve some directional consistency by repeating previous actions, whereas uniform random perturbations completely disrupt planned multi-step trajectories, eliminating the path optimality advantages that temporal abstraction provides under more predictable conditions.

Sequence Generation Frequency (SGF) increases under uniform random conditions compared to sticky actions, though the magnitude varies by environment. Simple mazes show large relative increases (8×8: +36% from 5.45 to 7.42; 16×16 empty: +60% from 8.83 to 14.14; 24×24 empty: +56% from 12.41 to 19.33), while obstacle-dense environments show more modest increases (16×16 with 15% obstacles: +28% from 16.26 to 20.85; 16×16 with 25% obstacles: +8% from 19.41 to 20.96). The elevated SGF values demonstrate that uniform random perturbations erode temporal abstraction benefits, as GPS must generate new action sequences more frequently when unpredictable execution errors disrupt the execution of planned trajectories. Nevertheless, GPS's sustained ASR advantage over DQN in most environments indicates that multi-step sequence generation retains value over purely reactive single-action decisions, even under high execution uncertainty.

### C.3 Summary

These findings suggest that GPS's sequence-level decision making provides inherent robustness against action execution noise. By generating coherent multi-step action sequences, the approach creates a natural buffer against individual action failures. The sequence generation paradigm appears to offer resilience to stochastic perturbations, as the method can adapt its behavioral patterns over time while maintaining overall navigational effectiveness despite local execution uncertainties.

The robust performance under stochastic conditions provides initial evidence that temporal abstraction through sequence generation may offer advantages beyond sample efficiency. The ability to operate at the sequence level appears to confer resilience to environmental uncertainties, suggesting potential applicability to scenarios where perfect action execution cannot be guaranteed, though further evaluation across diverse stochastic environments would be needed to establish broader generalizability of these findings.

## D  Evaluation on Partial Observability Environments

Real-world tasks often involve partial observability, where agents must make decisions based on limited local information rather than complete environmental knowledge. To evaluate GPS's robustness under such constraints, we tested our approach with agents restricted to a $7 \times 7$ observation window centered on their current position, obscuring the majority of the $16 \times 16 / 24 \times 24$ maze structure.

We implemented partial observability by modifying the observation space to show only a $7 \times 7$ window around the agent's current position. Cells outside this window are marked as unobserved, preventing the agent from accessing global maze structure or distant goal locations. This constraint fundamentally challenges sequence-based methods like GPS, which must generate entire action sequences without visibility of the complete path to the goal. The partial observability setting introduces limited planning horizons (visibility of only 3 cells in any direction), implicit memory requirements for previously observed areas, and increased exploration complexity without global visibility.

Table 7 presents GPS's performance under partial observability constraints compared to the DQN baseline across all maze configurations except $8 \times 8$. The results reveal distinct performance patterns across environment types that illuminate both the strengths and limitations of sequence-level decision making under uncertainty.

**Performance in Open Spaces and Corridor Environments.** In empty and corridor environments, GPS demonstrates strong early-stage performance under partial observability. In the empty 16×16 maze, GPS shows a substantial initial advantage (0.54 vs 0.27 ASR at 100K steps), though this gap narrows considerably by the time of convergence (0.71 vs 0.60 at 1M steps). In the corridor environment, GPS maintains more pronounced advantages, achieving 0.79 ASR at 1M steps compared to DQN's 0.26, representing a 3× improvement. These patterns suggest that generating action sequences provides inherent advantages in these structurally simple settings: multi-step sequences naturally maintain movement direction and exploration consistency, helping the agent navigate effectively even when it cannot see its destination. The narrow pathways of corridor mazes may provide implicit guidance that reduces the disadvantage of partial observability; once an agent commits to moving down a corridor, the constrained environment naturally channels movement toward valid destinations.

**Scaling Effects and Path Efficiency Trade-offs.** However, GPS's advantages diminish with increasing environment size. In the larger 24×24 empty maze, performance converges more closely between the methods (0.31 vs 0.30 ASR at 1.5M steps), contrasting with the substantial gaps observed in smaller mazes. This convergence likely reflects the compounding challenge of limited visibility over longer navigation distances; while the 7×7 window covers a reasonable fraction of the 16×16 maze, it represents a much smaller proportional view in 24×24 environments, diminishing GPS's advantage from directional consistency as both methods struggle with the fundamentally limited information available for long-distance navigation. The SGF metric further supports this interpretation: GPS requires substantially more frequent sequence generation in 24×24 environments compared to 16×16, and the proportional increase in decision frequency from full to partial observability grows considerably larger as environment size increases. Additionally, the moderate path ef-

Table 7: GPS and DQN Performance Under Partial Observability ($7 \times 7$ view window). Both GPS and DQN results are based on single-seed experiment.

| Environment | ASR at Training Steps | | | | | | | | ASR Converge >0.9 Step | | PER | | SGF | |
|---|---|---|---|---|---|---|---|---|---|---|---|---|---|---|
| | 100K | | 500K | | 1M | | 1.5M | | | | | | | |
| | GPS | DQN | GPS | DQN | GPS | DQN | GPS | DQN | GPS | DQN | GPS | DQN | GPS | DQN |
| 16x16 | 0.54 | 0.27 | 0.67 | 0.60 | 0.71 | 0.60 | 0.71 | 0.60 | >1.5M | >1.5M | 0.61 | 0.96 | 10.88 | - |
| 16x16_obstacles_15% | 0.30 | 0.15 | 0.34 | 0.25 | 0.34 | 0.29 | 0.34 | 0.29 | >1.5M | >1.5M | 0.70 | 1.00 | 12.74 | - |
| 16x16_obstacles_25% | 0.24 | 0.10 | 0.25 | 0.20 | 0.34 | 0.21 | 0.34 | 0.21 | >1.5M | >1.5M | 0.59 | 1.00 | 18.02 | - |
| 16x16_rooms | 0.11 | 0.10 | 0.39 | 0.27 | 0.45 | 0.28 | 0.45 | 0.28 | >1.5M | >1.5M | 0.66 | 0.97 | 14.08 | - |
| 16x16_corridors | 0.48 | 0.19 | 0.79 | 0.26 | 0.79 | 0.26 | - | - | >1M | >1M | 0.57 | 0.97 | 11.79 | - |
| 24x24 | 0.22 | 0.14 | 0.31 | 0.26 | 0.31 | 0.30 | 0.31 | 0.30 | >1.5M | >1.5M | 0.61 | 0.96 | 19.26 | - |
| 24x24_obstacles_15% | 0.17 | 0.05 | 0.32 | 0.05 | 0.33 | 0.05 | 0.33 | 0.05 | >1.5M | >1.5M | 0.47 | 1.00 | 16.65 | - |
| 24x24_obstacles_25% | 0.12 | 0.04 | 0.25 | 0.04 | 0.27 | 0.04 | 0.27 | 0.04 | >1.5M | >1.5M | 0.48 | 1.00 | 18.49 | - |

ficiency ratios (PER 0.61 in empty, 0.57 in corridors) indicate suboptimal navigation compared to DQN's near-perfect efficiency (PER 0.96-0.97), suggesting that while GPS reaches goals successfully, the generated sequences include unnecessary detours when operating with incomplete information. Importantly, DQN's higher PER values are calculated only on its successful episodes, which represent a much smaller and potentially easier subset of test instances compared to GPS's broader success coverage, potentially overstating DQN's true path optimality advantage.

**Complex Spatial Navigation.** GPS faces increased challenges in obstacle-dense and rooms environments under partial observability, though it consistently outperforms DQN across these configurations. In the 16×16 maze with 15% obstacles, while GPS maintains early advantages (0.30 vs 0.15 at 100K steps), both methods struggle to achieve high success rates, with GPS plateauing at 0.34 ASR and DQN at 0.29 ASR by 1.5M steps. The performance gap widens in denser obstacle configurations: in the 16×16 maze with 25% obstacles, GPS achieves 0.34 ASR compared to DQN's 0.21, while in rooms environments, GPS reaches 0.45 ASR versus DQN's 0.28. These results represent substantial degradation compared to GPS's performance under full observability (where it typically achieves 0.90+ ASR in similar environments), indicating that partial observability fundamentally challenges sequence-based planning. In larger mazes, the challenges intensify: 24×24 environments show GPS achieving 0.27-0.33 ASR across obstacle configurations, compared to DQN's 0.04-0.05, with performance substantially below what either method achieves under full observability.

**Sequence Commitment vs. Reactive Adaptation.** We hypothesize these difficulties stem from the compounded uncertainty of partial observability and complex spatial navigation. In obstacle environments, GPS must generate multi-step sequences without seeing upcoming barriers, leading to sequences that commit to trajectories encountering unseen walls. Rooms environments require precise doorway navigation and chamber transitions that are particularly challenging when global layout information is unavailable – GPS may generate sequences directing movement toward a distant goal without knowledge of intervening walls or doorways, resulting in collision-heavy paths. While DQN's step-by-step approach theoretically allows immediate incorporation of newly revealed environmental features, its poor performance suggests that reactive single-action decisions prove insufficient for effective exploration and navigation under limited visibility. GPS's relative advantage despite its commitment to multi-step sequences indicates that maintaining directional consistency through sequences provides more value than reactive flexibility when visibility is severely constrained, though neither approach fully solves the challenge of navigating complex partially-observed spaces.

**Training Efficiency and Sample Complexity.** None of the tested configurations achieved the 0.9 ASR convergence threshold within 1.5M training steps, indicating substantially increased sample complexity under limited visibility compared to full observability settings where convergence typically occurs within up to 1M steps. This represents a significant challenge for sequence-based methods under partial observability, as the compound effect of incomplete information and multi-step commitment makes effective policy learning considerably more sample-intensive.

**Behavioral Adaptation Through Sequence Frequency.** The Sequence Generation Frequency (SGF) metric further illuminate GPS's adaptive response to partial observability. The elevated SGF under partial

observability (10.88-19.26 across environments vs 8.76-17.77 with full visibility) reflects the increased challenges of operating with limited environmental information – more frequent sequence generation becomes necessary to maintain navigation progress when initial sequences prove less effective. The variation in SGF across environment types reveals two distinct dimensions of challenge: simple environments like the empty 16×16 maze shows SGF of 10.88, while more complex obstacle environments reach 16.65-18.49, and the proportional increase from full to partial observability grows substantially larger in 24×24 environments, indicating that both environmental complexity and scale compound the limitations of sequence-based planning under limited visibility. This pattern indicates that GPS generates new sequences more frequently when navigating uncertain terrain, likely because sequences planned with incomplete knowledge of obstacle locations prove less effective during execution, requiring more decision points to navigate around newly discovered barriers. This supports our hypothesis about the fundamental challenges of sequence planning in complex partially-observed spaces, where the commitment to multi-step plans conflicts with the need to respond to newly revealed information.

**Trade-offs and Future Directions.** These findings highlight both the potential and limitations of sequence-based decision making under partial observability. GPS's sustained advantages in simple environments (particularly the 3× improvement in corridors) demonstrate that multi-step action sequences can provide useful navigational structure even with limited sensory input, though performance degrades substantially compared to full observability, particularly in obstacle-dense environments where the increased generation frequency (higher SGF) still proves insufficient to overcome the challenges of planning with incomplete spatial information. These results suggest several promising directions for future work. Confidence-conditioned sequence horizons could allow the decoder to learn to generate shorter sequences when observation uncertainty is high, balancing temporal abstraction benefits with the need for frequent replanning. Alternatively, integrating memory mechanisms that explicitly track previously observed regions could help GPS build a more complete spatial representation over time, reducing the disadvantage of limited immediate visibility and enabling more effective sequence planning in complex environments.

# E   Self-Correction Analysis

To substantiate our claim that GPS can self-correct through sequential sequences (Section 4.4.2), we conducted a comprehensive analysis of GPS's ability to recover from deviations in its trajectory.

## E.1   Methodology

We developed a quantitative framework to measure GPS's self-correction behavior by tracking progress toward the goal using Breadth-First Search (BFS) distance. Our analysis focuses on identifying when GPS deviates from optimal progress and measuring its ability to generate correction sequences in subsequent decisions.

### E.1.1   Deviation Detection

For each action sequence executed during an episode, we calculate:

1. **Initial BFS distance** ($d_{\text{before}}$): The shortest path distance to the goal before sequence execution

2. **Final BFS distance** ($d_{\text{after}}$): The shortest path distance to the goal after sequence execution

A deviation is detected when the sequence does not bring the agent closer to the goal:

$$d_{\text{after}} \geq d_{\text{before}} \tag{1}$$

This indicates the agent either maintained the same distance or moved further from the goal, representing a sequence that requires correction.

### E.1.2    Correction Sequence Classification

When a deviation is identified, we track whether GPS generates a correction sequence in subsequent decisions. We measure correction at multiple time horizons:

- **Immediate Correction**: The very next sequence reduces the optimal path length from its own starting position to its ending position, demonstrating GPS's ability to generate locally optimal moves that recover from the previous suboptimal decision

- **Near-term Correction** (within 2 sequences): Recovery occurs within the next two decision points

- **Medium-term Correction** (within 3-4 sequences): Recovery within 3-4 subsequent decisions

- **Long-term Correction** (5+ sequences): Recovery before episode termination

A correction sequence is considered successful when the agent's BFS distance falls below what would have been expected from optimal execution of the original deviating sequence.

### E.2    Results

Table 8 presents the self-correction analysis results across representative maze environments. The analysis was performed only on successful episodes to focus on GPS's recovery capabilities when it ultimately reaches the goal.

Table 8: GPS Self-Correction Success Rates Across Maze Environments

| Environment | Immediate Correction | Near-term ($\leq$2 seq) | Medium-term ($\leq$4 seq) | Long-term (5+ seq) |
|---|---|---|---|---|
| 8×8 | 0.885 ± 0.095 | 0.889 ± 0.092 | 0.896 ± 0.086 | 1.000 ± 0.000 |
| 16×16 | 0.996 ± 0.006 | 1.000 ± 0.000 | 1.000 ± 0.000 | 1.000 ± 0.000 |
| 16×16_obstacles_15% | 0.753 ± 0.311 | 0.779 ± 0.313 | 0.794 ± 0.292 | 1.000 ± 0.000 |
| 16×16_obstacles_25% | 0.765 ± 0.324 | 0.789 ± 0.298 | 0.808 ± 0.272 | 1.000 ± 0.000 |
| 16×16_rooms | 0.624 ± 0.114 | 0.773 ± 0.088 | 0.805 ± 0.079 | 1.000 ± 0.000 |
| 16×16_corridors | 0.965 ± 0.046 | 1.000 ± 0.000 | 1.000 ± 0.000 | 1.000 ± 0.000 |
| 24×24 | 0.899 ± 0.143 | 1.000 ± 0.000 | 1.000 ± 0.000 | 1.000 ± 0.000 |

*Note*: Values represent mean ± standard deviation across representative models from three different training seeds per maze type. Analysis conducted only on successful episodes.

### E.3    Analysis and Discussion

The results provide strong empirical evidence for GPS's self-correction capability:

**High Immediate Correction Rates**: Across all environments, GPS demonstrates substantial immediate correction rates, ranging from 62.4% in complex room environments to 99.6% in open spaces. This indicates that GPS frequently generates correction sequences at the very next decision point after detecting a deviation.

**Environment-Specific Patterns**:

- **Open spaces** (16×16, 24×24): Nearly perfect immediate correction (>89%), reflecting GPS's ability to quickly identify and generate correction sequences for directional errors

- **Structured environments** (rooms, obstacles): Lower immediate correction rates (62-77%) but strong recovery within 2-4 sequences, suggesting GPS adapts its correction strategy to environmental complexity

- **Corridors**: High immediate correction (96.5%), likely due to the constrained nature limiting deviation possibilities

**Guaranteed Long-term Recovery**: All environments show 100% long-term correction rates in successful episodes, confirming that GPS consistently generates effective correction sequences even after significant trajectory deviations.

**Variance Patterns**: Higher standard deviations in obstacle-rich environments ($\pm 0.3$) compared to open spaces ($\pm 0.006$) indicate that correction difficulty varies with local maze structure, yet GPS maintains robust recovery capabilities.

The evidence presented here directly supports our claim that GPS employs sequence-level closed-loop control, generating correction sequences at subsequent decision points without requiring initially perfect action sequences. This self-correction mechanism helps explain GPS's performance characteristics, where the ability to recover from deviations enables reliable goal-reaching (high ASR) even when individual sequences may not follow strictly optimal paths (resulting in lower PER).

## F   Computational Cost Analysis

While GPS demonstrates superior sample efficiency, we analyze its computational trade-offs compared to baseline methods to provide a complete picture of the method's practicality.

**Pre-training Costs.** The VAE decoder requires approximately 10-15 minutes of pre-training on an Apple M1 Max with 64 GB RAM. This is a one-time cost per action space, and the trained decoder can be reused across different environments with the same action space, amortizing this cost over multiple experiments.

**Inference Overhead.** During inference, GPS requires 1.6-3.3x the wall-clock time per decision compared to baselines due to the Actor-Decoder pipeline. In the 16x16 maze with 15% obstacles, GPS takes 2.72ms $\pm$ 0.25ms per decision versus DAR's 1.39ms $\pm$ 0.31ms (1.96x overhead) and DQN's 1.67ms $\pm$ 0.05ms (1.63x overhead), while being 1.17x faster than TempoRL's 3.18ms $\pm$ 0.76ms. In the larger 24x24 environment, the overhead pattern persists: GPS requires 3.82ms $\pm$ 0.81ms per decision compared to DAR's 1.16ms $\pm$ 0.09ms (3.29x overhead), DQN's 1.68ms $\pm$ 0.05ms (2.27x overhead), and TempoRL's 2.89ms $\pm$ 0.77ms (1.32x overhead). However, since GPS makes decisions less frequently due to its sequence-level abstraction, we also measure normalized episode efficiency—the total episode wall-clock time divided by the optimal path length to normalize for task difficulty and enable fair comparison across algorithms with different success rates. In the 16x16 with 15% obstacles environment, GPS (4.6ms $\pm$ 0.19ms per optimal step) performs comparably to baselines (DQN: 4.03ms, TempoRL: 4.55ms, DAR: 4.65ms), indicating that the per-decision overhead is largely offset by reduced decision frequency. However, in the 24x24 environment, GPS shows higher normalized costs (7.24ms $\pm$ 1.51ms per optimal step) compared to baselines (DQN: 5.83ms, TempoRL: 5.83ms, DAR: 4.92ms). Note that this metric is computed only on successful episodes, which may bias results toward baselines that solve only easier test instances (GPS: 90% ASR vs DQN: 24%, TempoRL: 46%, DAR: 23%), suggesting the true computational cost difference for solving challenging instances may be smaller than indicated.

**Training Time to Convergence.** Despite higher per-step costs, GPS achieves faster wall-clock convergence in medium complexity environments like 16x16 mazes with 15% obstacles due to its superior sample efficiency. In 16x16 mazes with 15% obstacles, GPS reaches 90% ASR in 3.3 hours (500K steps), while other methods fail to reach 90% ASR within the same training budget: DQN achieves only 85% ASR after 5.83 hours (1.5M steps), TempoRL reaches 82% ASR after 9.2 hours (1.5M steps), and DAR achieves 64% ASR after 4.08 hours (1.5M steps). For the larger 24x24 maze, GPS's computational overhead becomes more apparent, with training times becoming comparable to the fastest baseline despite superior convergence and final performance. GPS achieves 90% ASR in 11.5 hours (1M steps), while all baselines fail to reach 90% ASR: DQN reaches only 24% ASR after 10.5 hours (1.5M steps), TempoRL achieves 46% ASR after 13.02 hours (1.5M steps), and DAR reaches 23% ASR after 6.26 hours (1.5M steps), illustrating the trade-off between computational efficiency and learning effectiveness.

**Computational Trade-offs and Practical Considerations.** While GPS incurs additional computational overhead per decision compared to most baselines, its reduced decision frequency often compensates for this cost. The computational trade-offs of GPS vary with environment complexity: in medium complexity environments like 16x16 with 15% obstacles maze, GPS provides clear training time advantages, reaching target performance levels considerably faster than baselines. In larger environments such as 24x24 maze, training times become comparable to the fastest baselines, though GPS achieves substantially superior final performance (90% vs 24% ASR for DQN). This pattern suggests that GPS becomes most practical when learning effectiveness is prioritized over raw computational speed, particularly in scenarios where baseline methods struggle to reach acceptable performance levels rather than purely on training efficiency grounds.

Table 9: Computational cost analysis across environments and methods. Values represent mean ± standard deviation over 5 runs.

| Environment | Metric | GPS | DQN | TempoRL | DAR |
|---|---|---|---|---|---|
| 16x16_obst_15% | Inference time per decision (ms) | 2.72 ± 0.25 | 1.67 ± 0.05 | 3.18 ± 0.76 | 1.39 ± 0.31 |
| | Normalized episode efficiency (ms/opt. step) | 4.6 ± 0.19 | 4.03 ± 0.08 | 4.55 ± 0.75 | 4.65 ± 0.56 |
| | Training wall-clock to 90% ASR | 3.3h | 5.83h | 9.2h | 4.08h |
| | Total training wall-clock (h) | 9.97 ± 0.31 | 5.83 ± 0.35 | 9.2 ± 1.72 | 4.08 ± 0.65 |
| 24x24 | Inference time per decision (ms) | 3.82 ± 0.81 | 1.68 ± 0.05 | 2.89 ± 0.77 | 1.16 ± 0.09 |
| | Normalized episode efficiency (ms/opt. step) | 7.24 ± 1.51 | 5.83 ± 0.17 | 5.83 ± 1.18 | 4.92 ± 0.54 |
| | Training wall-clock to 90% ASR | 11.5h | 10.5h | 13.02h | 6.26h |
| | Total training wall-clock (h) | 17.35 ± 2.11 | 10.5 ± 0.39 | 13.02 ± 1.67 | 6.26 ± 0.12 |
| **VAE Pre-training (one-time)** | | 10-15 minutes on Apple M1 Max | | | |

In summary, GPS represents a favorable computational trade-off for challenging navigation tasks where baseline methods fail to achieve acceptable performance, offering superior learning effectiveness at competitive training costs.

# G Comparison: Pre-trained Decoder vs. End-to-End Training

When training GPS, one needs to choose whether to pre-train the Proto-Sequence Decoder (PSD) or train it jointly end-to-end with the actor and critic networks. To assess the impact of this choice, we conducted a comprehensive comparison across our maze environments. This analysis was motivated by the observation that decoder pre-training could potentially encode implicit biases about feasible action sequences, raising questions about whether GPS's advantages stem from the sequence-level decision-making paradigm itself or from initialization artifacts.

## G.1 Experimental Setup

We evaluated two **GPS** variants:

- GPS: Our default implementation with a pre-trained VAE decoder (frozen during actor-critic training)

- GPS-E2E: End-to-end training where all three components (actor, decoder, critic) are jointly optimized from random initialization

Both variants were evaluated on the same set of maze environments using identical hyperparameters for the actor and critic networks. The decoder architecture remained unchanged; only the training procedure differed.

## G.2 Results and Analysis

Table 10 presents the comparative results across maze environments. The findings reveal several important insights:

Table 10: Performance comparison between GPS with pre-trained decoder and end-to-end training (GPS-E2E). Both GPS and GPS-E2E results are based on single-seed experiment.

| Environment | ASR at Training Steps | | | | | | | | ASR Converge >0.9 Step | | PER | | SGF | |
| | 100K | | 500K | | 1M | | 1.5M | | | | | | | |
| | GPS | GPS-E2E | GPS | GPS-E2E | GPS | GPS-E2E | GPS | GPS-E2E | GPS | GPS-E2E | GPS | GPS-E2E | GPS | GPS-E2E |
|---|---|---|---|---|---|---|---|---|---|---|---|---|---|---|
| 8x8 | 1.00 | 0.98 | 1.00 | 1.00 | 1.00 | 1.00 | - | - | 100k | 100k | 0.89 | 0.97 | 3.30 | 5.28 |
| 16x16 | 0.71 | 0.58 | 1.00 | 1.00 | 1.00 | 1.00 | 1.00 | 1.00 | 300k | 200k | 0.85 | 0.89 | 8.99 | 11.10 |
| 16x16_obstacles_15% | 0.43 | 0.31 | 0.92 | 0.92 | 0.92 | 0.94 | 0.92 | 0.94 | 300k | 300k | 0.75 | 0.82 | 14.35 | 21.81 |
| 16x16_obstacles_25% | 0.05 | 0.07 | 0.37 | 0.40 | 0.47 | 0.89 | 0.48 | 0.93 | >1.5M | 1.4M | 0.68 | 0.79 | 18.47 | 27.03 |
| 16x16_rooms | 0.21 | 0.13 | 0.72 | 0.89 | 0.91 | 0.90 | 0.91 | 0.90 | 700k | 1M | 0.68 | 0.70 | 16.27 | 30.37 |
| 16x16_corridors | 0.93 | 0.96 | 0.99 | 0.99 | 0.99 | 0.99 | - | - | 100k | 100k | 0.79 | 0.87 | 9.11 | 14.95 |
| 24x24 | 0.28 | 0.11 | 1.00 | 1.00 | 1.00 | 1.00 | 1.00 | 1.00 | 200k | 300k | 0.84 | 0.87 | 11.39 | 14.20 |
| 24x24_obstacles_15% | 0.20 | 0.34 | 0.66 | 0.67 | 0.74 | 0.68 | 0.81 | 0.77 | >1.5M | >1.5M | 0.58 | 0.52 | 14.64 | 29.68 |
| 24x24_obstacles_15% | 0.14 | 0.14 | 0.36 | 0.44 | 0.46 | 0.50 | 0.51 | 0.51 | >1.5M | >1.5M | 0.55 | 0.52 | 17.22 | 29.28 |

### G.2.1 Performance Parity in Success Rate and Convergence

The most significant finding is that **GPS-E2E achieves comparable final performance** to GPS with pre-trained decoder across most tested environments. Both variants converge to similarly high success rates, with convergence speeds being environment-dependent (e.g., 16×16: GPS-E2E reaches ASR > 0.9 at 200k steps vs GPS at 300k steps; 24×24: GPS converges at 200k steps vs GPS-E2E at 300k steps). The final ASR values are nearly identical across environments (e.g., 16×16: both reach 1.00; 16×16_obs_15%: 0.92 vs 0.94; 16×16_rooms: 0.91 vs 0.90). This demonstrates that:

1. The performance advantage of GPS stems primarily from the **sequence-level decision-making paradigm** rather than from implicit biases encoded in decoder pre-training.

2. GPS can successfully learn effective action sequence representations from scratch during joint training.

3. Decoder pre-training is not a fundamental requirement. However, as we show below, there are trade-offs with regard to path optimality and the number of sequences that need to be generated per trajectory.

### G.2.2 Path Efficiency vs. Temporal Abstraction Trade-off

An interesting pattern emerges when examining the Path Efficiency Ratio (PER) and Sequence Generation Frequency (SGF) metrics:

- **Higher PER for GPS-E2E**: End-to-end training produces slightly higher path efficiency (e.g., 16×16: 0.89 vs 0.85; 16×16_obs_15%: 0.82 vs 0.75). This indicates that GPS-E2E learns to generate action sequences that more closely follow optimal paths.

- **Lower SGF for GPS**: The pre-trained decoder variant requires fewer sequence generations per episode (e.g., 16×16: 8.99 vs 11.10; 24x24_obstacles_15%: 14.64 vs 29.68; 16×16_rooms: 16.27 vs 30.37), suggesting it produces longer, more temporal abstraction action sequences.

This trade-off reveals an important behavioral difference: GPS with pre-trained decoder appears to develop more aggressive temporal abstraction strategies – generating longer sequences that reduce decision frequency at the cost of some path optimality. In contrast, GPS-E2E learns to balance temporal abstraction with path efficiency, generating somewhat shorter sequences that more closely track optimal trajectories.

We hypothesize this difference arises because:

1. The pre-trained decoder's latent space structure, learned from diverse synthetic sequences, encourages exploration of longer, more varied action patterns

2. End-to-end training jointly optimizes sequence generation with the specific task objectives, potentially leading to more task-specific (and thus more efficient but less exploratory) sequence patterns

This suggests that decoder pre-training may provide a form of **implicit exploration regularization**, encouraging the agent to commit to longer temporal abstractions even when shorter sequences might be locally optimal.

### G.3    Decoder Transferability Across Tasks

An important practical advantage of the pre-trained decoder approach is its reusability. In our experiments, **the same pre-trained decoder was successfully used across**:

- All maze configurations (empty, obstacles with varying densities, rooms, corridors)

- All maze sizes ($8\times8$, $16\times16$, $24\times24$)

- Multiple experimental conditions (deterministic dynamics, stochastic environments and partial observability)

This transferability demonstrates that the decoder learns general-purpose sequence generation capabilities that apply across diverse navigation scenarios within the same action space. For applications involving multiple related tasks with shared action spaces, pre-training the decoder once and reusing it can provide practical benefits. However, our end-to-end results confirm that decoder reusability is an added benefit rather than a necessity for GPS's effectiveness. The choice between variants ultimately reflects the PER-SGF trade-off: pre-training encourages more aggressive temporal abstraction (lower SGF) while end-to-end training optimizes for path efficiency (higher PER).

### G.4    Conclusion

Our analysis demonstrates that GPS achieves strong performance with both pre-trained and end-to-end training, confirming that its advantages stem from sequence-level decision-making rather than initialization artifacts. The comparable performance of GPS-E2E validates that GPS's superior performance compated to the baselines (Section 4.4) is derived by its architectural approach to temporal abstraction.

The pre-trained variant offers decoder reusability and more aggressive temporal abstraction (lower SGF), while the end-to-end variant provides superior path efficiency (higher PER) with comparable success rates. This PER-SGF trade-off suggests decoder initialization influences the exploration-exploitation balance. Practitioners can select either variant based on specific requirements without sacrificing fundamental performance.

Future work could explore combining both approaches' strengths through adaptive decoder training schedules (starting pre-trained for exploration, transitioning to task-specific optimization) or curriculum learning that leverages the complementary benefits of both paradigms.

## H    ASR Statistical Significance Testing

To assess the statistical significance of the differences in Average Success Rates (ASR) between our proposed method (GPS) and the baselines (DQN, TempoRL, DAR), we employed McNemar's test. This section details the methodology and presents the results of these tests.

**Methodology**

McNemar's test is a non-parametric test suitable for paired nominal data. It is used to determine whether there is a significant difference in the proportions of two related samples, such as when two algorithms are evaluated on the same set of test instances. In our context, each maze evaluation episode serves as a paired instance, and the outcome for each algorithm (GPS or DQN for example) is categorized as either a success or a failure.

An episode was deemed a **success** if the agent reached the goal in an episodic length of less than 75 steps. Otherwise, it was considered a **failure**.

For each pair of algorithms (GPS vs. DQN) on a given maze type, we constructed a $2 \times 2$ contingency table based on the outcomes of common evaluation episodes:

|  |  | Algorithm B (DQN) | |
|  |  | Success | Failure |
|---|---|---|---|
| Algorithm A (GPS) | Success | $a$ | $b$ |
|  | Failure | $c$ | $d$ |

Where:

- $a$: Number of episodes where both GPS and DQN succeeded.

- $b$: Number of episodes where GPS succeeded and DQN failed.

- $c$: Number of episodes where GPS failed and DQN succeeded.

- $d$: Number of episodes where both GPS and DQN failed.

McNemar's test focuses on the discordant pairs ($b$ and $c$). The null hypothesis ($H_0$) is that the two algorithms have the same ASR. The test statistic is calculated as:

$$\chi^2 = \frac{(b-c)^2}{b+c}$$

This statistic follows a chi-squared distribution with 1 degree of freedom. We used the version of the test without continuity correction, as implemented in 'statsmodels.stats.contingency_tables.mcnemar'.

The significance level was established at $\alpha = 0.05$. However, in all comparisons, the calculated p-values were substantially lower ($p < 0.0001$), providing strong evidence to reject the null hypothesis and confirming statistically significant differences in ASR performance.

**Results: GPS vs. DQN**

The results of McNemar's test comparing GPS (Algorithm A) to DQN (Algorithm B) across various maze configurations are summarized in Table 11. The Average Success Rate (ASR) reported in the table for each algorithm is based on Table 2:

- ASR (GPS) $= (a+b)/(a+b+c+d)$

- ASR (DQN) $= (a+c)/(a+b+c+d)$

All comparisons in Table 11 yielded p-values below 0.0001, indicating highly statistically significant differences between GPS and DQN across all environments. GPS outperforms DQN in the majority of setups. The sole exception is the 16×16 maze with 25% obstacles, where DQN achieves significantly better performance. As discussed in Section 4.4.1, this result reflects the interplay between maze scale and obstacle density rather than a general limitation in dense environments, given that GPS significantly outperforms DQN in the 24×24 setup with the same obstacle density.

The following sections provide the detailed per-run summaries logged and the specific contingency tables used for McNemar's test for each maze configuration.

**Maze: 8x8**

- *GPS (Algorithm A) Summary:* Total episodes: 4978, Successes: 4978, Failures: 0, Errors: 0

- *DQN (Algorithm B) Summary:* Total episodes: 4978, Successes: 4211, Failures: 767, Errors: 0

*Contingency Table:*

Table 11: McNemar's Test Results for GPS vs. DQN. **All differences are statistically significant** ($p < 0.0001$). Results favor GPS in all cases except where marked with $^\dagger$, which favors DQN.

| Environment | ASR (GPS) | ASR (DQN) | McNemar Stat. |
|---|---|---|---|
| 8×8 | **1.00** | 0.85 | 767.00 |
| 16×16 | **1.00** | 0.74 | 1350.00 |
| 16×16 w/ 15% obs. | **0.94** | 0.84 | 75.98 |
| 16×16 w/ 25% obs. | 0.57 | **0.72** | 114.57$^\dagger$ |
| 16×16 rooms | **0.85** | 0.64 | 414.42 |
| 16×16 corridors | **0.98** | 0.80 | 448.63 |
| 24×24 | **0.94** | 0.28 | 3244.29 |
| 24×24 w/ 15% obs. | **0.83** | 0.13 | 3385.79 |
| 24×24 w/ 25% obs. | **0.49** | 0.10 | 1674.61 |

| | | DQN (Alg. B) | |
|---|---|---|---|
| | | **Success** | **Failure** |
| **GPS (Alg. A)** | *Success* | 4211 (a) | 767 (b) |
| | *Failure* | 0 (c) | 0 (d) |

Common episodes for comparison: 4978

McNemar's Statistic: 767.0000, p-value: < **0.0001**

## Maze: 16x16

- *GPS (Algorithm A) Summary:* Total episodes: 4994, Successes: 4994, Failures: 0, Errors: 0

- *DQN (Algorithm B) Summary:* Total episodes: 4994, Successes: 3644, Failures: 1350, Errors: 0

*Contingency Table:*

| | | DQN (Alg. B) | |
|---|---|---|---|
| | | **Success** | **Failure** |
| **GPS (Alg. A)** | *Success* | 3644 (a) | 1350 (b) |
| | *Failure* | 0 (c) | 0 (d) |

Common episodes for comparison: 4994

McNemar's Statistic: 1350.0000, p-value: < **0.0001**

## Maze: 16x16 Obstacles 15%

- *GPS (Algorithm A) Summary:* Total episodes: 1050, Successes: 981, Failures: 69, Errors: 0

- *DQN (Algorithm B) Summary:* Total episodes: 1050, Successes: 882, Failures: 168, Errors: 0

*Contingency Table:*

| | | DQN (Alg. B) | |
|---|---|---|---|
| | | **Success** | **Failure** |
| **GPS (Alg. A)** | *Success* | 867 (a) | 114 (b) |
| | *Failure* | 15 (c) | 54 (d) |

Common episodes for comparison: 1050

McNemar's Statistic: 75.9767, p-value: < **0.0001**

**Maze: 16x16 Obstacles 25%**

- *GPS (Algorithm A) Summary:* Total episodes: 2010, Successes: 1143, Failures: 867, Errors: 0

- *DQN (Algorithm B) Summary:* Total episodes: 2010, Successes: 1451, Failures: 559, Errors: 0

*Contingency Table:*

|  |  | DQN (Alg. B) | |
|  |  | Success | Failure |
|---|---|---|---|
| **GPS (Alg. A)** | *Success* | 883 (a) | 260 (b) |
|  | *Failure* | 568 (c) | 299 (d) |

Common episodes for comparison: 2010
McNemar's Statistic: 114.5700, p-value: < **0.0001**

**Maze: 16x16 Rooms**

- *GPS (Algorithm A) Summary:* Total episodes: 2930, Successes: 2496, Failures: 434, Errors: 0

- *DQN (Algorithm B) Summary:* Total episodes: 2930, Successes: 1888, Failures: 1042, Errors: 0

*Contingency Table:*

|  |  | DQN (Alg. B) | |
|  |  | Success | Failure |
|---|---|---|---|
| **GPS (Alg. A)** | *Success* | 1746 (a) | 750 (b) |
|  | *Failure* | 142 (c) | 292 (d) |

Common episodes for comparison: 2930
McNemar's Statistic: 414.4215, p-value: < **0.0001**

**Maze: 16x16 Corridors**

- *GPS (Algorithm A) Summary:* Total episodes: 2725, Successes: 2671, Failures: 54, Errors: 0

- *DQN (Algorithm B) Summary:* Total episodes: 2725, Successes: 2182, Failures: 543, Errors: 0

*Contingency Table:*

|  |  | DQN (Alg. B) | |
|  |  | Success | Failure |
|---|---|---|---|
| **GPS (Alg. A)** | *Success* | 2160 (a) | 511 (b) |
|  | *Failure* | 22 (c) | 32 (d) |

Common episodes for comparison: 2725
McNemar's Statistic: 448.6323, p-value: < **0.0001**

**Maze: 24x24**

- *GPS (Algorithm A) Summary:* Total episodes: 4999, Successes: 4675, Failures: 324, Errors: 0

- *DQN (Algorithm B) Summary:* Total episodes: 4999, Successes: 1366, Failures: 3633, Errors: 0

*Contingency Table:*

|  |  | DQN (Alg. B) | |
| --- | --- | --- | --- |
|  |  | **Success** | **Failure** |
| **GPS (Alg. A)** | *Success* | 1333 (a) | 3342 (b) |
|  | *Failure* | 33 (c) | 291 (d) |

Common episodes for comparison: 4999
McNemar's Statistic: 3244.2907, p-value: $< 0.0001$

### Maze: 24x24 Obstacles 15%

- *GPS (Algorithm A) Summary:* Total episodes: 4997, Successes: 4098, Failures: 899, Errors: 0

- *DQN (Algorithm B) Summary:* Total episodes: 4997, Successes: 615, Failures: 4382, Errors: 0

*Contingency Table:*

|  |  | DQN (Alg. B) | |
| --- | --- | --- | --- |
|  |  | **Success** | **Failure** |
| **GPS (Alg. A)** | *Success* | 565 (a) | 3533 (b) |
|  | *Failure* | 50 (c) | 849 (d) |

Common episodes for comparison: 4997
McNemar's Statistic: 3385.7910, p-value: $< 0.0001$

### Maze: 24x24 Obstacles 25%

- *GPS (Algorithm A) Summary:* Total episodes: 5000, Successes: 2443, Failures: 2557, Errors: 0

- *DQN (Algorithm B) Summary:* Total episodes: 5000, Successes: 489, Failures: 4511, Errors: 0

*Contingency Table:*

|  |  | DQN (Alg. B) | |
| --- | --- | --- | --- |
|  |  | **Success** | **Failure** |
| **GPS (Alg. A)** | *Success* | 326 (a) | 2117 (b) |
|  | *Failure* | 163 (c) | 2394 (d) |

Common episodes for comparison: 5000
McNemar's Statistic: 1674.6123, p-value: $< 0.0001$

This detailed breakdown for each environment shows the specific data underlying the McNemar's tests.

### Results: GPS vs. TempoRL

Table 12 summarizes the Average Success Rates (ASR) for GPS and TempoRL, along with the McNemar test statistics and p-values derived from common paired evaluation episodes.

All comparisons in Table 12 yielded p-values below 0.0001, indicating highly statistically significant differences between GPS and TempoRL across all environments. GPS outperforms TempoRL in the majority of setups. The sole exception is the $16 \times 16$ maze with 25% obstacles, where TempoRL achieves significantly better performance. As discussed in Section 4.4.1, this result reflects the interplay between maze scale and obstacle density rather than a general limitation in dense environments, given that GPS significantly outperforms TempoRL in the $24 \times 24$ setup with the same obstacle density.

The following sections provide the detailed per-run summaries and the specific contingency tables used for McNemar's test for each maze configuration when comparing GPS with TempoRL.

Table 12: McNemar's Test Results for GPS vs. TempoRL. **All differences are statistically significant** ($p < 0.0001$)**.** Results favor GPS in all cases except where marked with $^\dagger$, which favors TempoRL.

| Environment | ASR (GPS) | ASR (TempoRL) | McNemar Stat. |
|---|---|---|---|
| 8×8 | **1.00** | 0.94 | 311.00 |
| 16×16 | **1.00** | 0.88 | 609.00 |
| 16×16 w/ 15% obs. | **0.94** | 0.80 | 86.78 |
| 16×16 w/ 25% obs. | 0.57 | **0.73** | 105.01$^\dagger$ |
| 16×16 rooms | **0.85** | 0.67 | 268.69 |
| 16×16 corridors | **0.98** | 0.89 | 176.14 |
| 24×24 | **0.94** | 0.52 | 1782.46 |
| 24×24 w/ 15% obs. | **0.83** | 0.17 | 2956.75 |
| 24×24 w/ 25% obs. | **0.49** | 0.08 | 1746.57 |

**Maze: 8x8**

- *GPS (Algorithm A) Summary:* Total episodes: 4978, Successes: 4978, Failures: 0, Errors: 0

- *TempoRL (Algorithm B) Summary:* Total episodes: 4978, Successes: 4667, Failures: 311, Errors: 0

*Contingency Table:*

| | | TempoRL (Alg. B) Success | Failure |
|---|---|---|---|
| **GPS (Alg. A)** | *Success* | 4667 (a) | 311 (b) |
| | *Failure* | 0 (c) | 0 (d) |

Common episodes for comparison: 4978
McNemar's Statistic: 311.0000, p-value: < **0.0001**

**Maze: 16x16**

- *GPS (Algorithm A) Summary:* Total episodes: 4994, Successes: 4994, Failures: 0, Errors: 0

- *TempoRL (Algorithm B) Summary:* Total episodes: 4994, Successes: 4385, Failures: 609, Errors: 0

*Contingency Table:*

| | | TempoRL (Alg. B) Success | Failure |
|---|---|---|---|
| **GPS (Alg. A)** | *Success* | 4385 (a) | 609 (b) |
| | *Failure* | 0 (c) | 0 (d) |

Common episodes for comparison: 4994
McNemar's Statistic: 609.0000, p-value: < **0.0001**

**Maze: 16x16 Obstacles 15%**

- *GPS (Algorithm A) Summary:* Total episodes: 1050, Successes: 981, Failures: 69, Errors: 0

- *TempoRL (Algorithm B) Summary:* Total episodes: 1050, Successes: 834, Failures: 216, Errors: 0

*Contingency Table:*

| | | TempoRL (Alg. B) | |
| --- | --- | --- | --- |
| | | **Success** | **Failure** |
| **GPS (Alg. A)** | *Success* | 783 (a) | 198 (b) |
| | *Failure* | 51 (c) | 18 (d) |

Common episodes for comparison: 1050

McNemar's Statistic: 86.7831, p-value: < **0.0001**

## Maze: 16x16 Obstacles 25%

- *GPS (Algorithm A) Summary:* Total episodes: 2010, Successes: 1143, Failures: 867, Errors: 0

- *TempoRL (Algorithm B) Summary:* Total episodes: 2010, Successes: 1462, Failures: 548, Errors: 0

*Contingency Table:*

| | | TempoRL (Alg. B) | |
| --- | --- | --- | --- |
| | | **Success** | **Failure** |
| **GPS (Alg. A)** | *Success* | 818 (a) | 325 (b) |
| | *Failure* | 644 (c) | 223 (d) |

Common episodes for comparison: 2010

McNemar's Statistic: 105.0165, p-value: < **0.0001**

## Maze: 16x16 Rooms

- *GPS (Algorithm A) Summary:* Total episodes: 2930, Successes: 2496, Failures: 434, Errors: 0

- *TempoRL (Algorithm B) Summary:* Total episodes: 2930, Successes: 1944, Failures: 986, Errors: 0

*Contingency Table:*

| | | TempoRL (Alg. B) | |
| --- | --- | --- | --- |
| | | **Success** | **Failure** |
| **GPS (Alg. A)** | *Success* | 1653 (a) | 843 (b) |
| | *Failure* | 291 (c) | 143 (d) |

Common episodes for comparison: 2930

McNemar's Statistic: 268.6984, p-value: < **0.0001**

## Maze: 16x16 Corridors

- *GPS (Algorithm A) Summary:* Total episodes: 2724, Successes: 2670, Failures: 54, Errors: 0

- *TempoRL (Algorithm B) Summary:* Total episodes: 2724, Successes: 2426, Failures: 298, Errors: 0

*Contingency Table:*

| | | TempoRL (Alg. B) | |
| --- | --- | --- | --- |
| | | **Success** | **Failure** |
| **GPS (Alg. A)** | *Success* | 2379 (a) | 291 (b) |
| | *Failure* | 47 (c) | 7 (d) |

Common episodes for comparison: 2724

McNemar's Statistic: 176.1420, p-value: < **0.0001**

**Maze: 24x24**

- *GPS (Algorithm A) Summary:* Total episodes: 4999, Successes: 4675, Failures: 324, Errors: 0

- *TempoRL (Algorithm B) Summary:* Total episodes: 4999, Successes: 2611, Failures: 2388, Errors: 0

*Contingency Table:*

| | | TempoRL (Alg. B) | |
| --- | --- | --- | --- |
| | | **Success** | **Failure** |
| **GPS (Alg. A)** | *Success* | 2448 (a) | 2227 (b) |
| | *Failure* | 163 (c) | 161 (d) |

Common episodes for comparison: 4999
McNemar's Statistic: 1782.4669, p-value: < **0.0001**

**Maze: 24x24 Obstacles 15%**

- *GPS (Algorithm A) Summary:* Total episodes: 4997, Successes: 4098, Failures: 899, Errors: 0

- *TempoRL (Algorithm B) Summary:* Total episodes: 4997, Successes: 860, Failures: 4137, Errors: 0

*Contingency Table:*

| | | TempoRL (Alg. B) | |
| --- | --- | --- | --- |
| | | **Success** | **Failure** |
| **GPS (Alg. A)** | *Success* | 706 (a) | 3392 (b) |
| | *Failure* | 154 (c) | 745 (d) |

Common episodes for comparison: 4997
McNemar's Statistic: 2956.7524, p-value: < **0.0001**

**Maze: 24x24 Obstacles 25%**

- *GPS (Algorithm A) Summary:* Total episodes: 5000, Successes: 2443, Failures: 2557, Errors: 0

- *TempoRL (Algorithm B) Summary:* Total episodes: 5000, Successes: 358, Failures: 4642, Errors: 0

*Contingency Table:*

| | | TempoRL (Alg. B) | |
| --- | --- | --- | --- |
| | | **Success** | **Failure** |
| **GPS (Alg. A)** | *Success* | 156 (a) | 2287 (b) |
| | *Failure* | 202 (c) | 2355 (d) |

Common episodes for comparison: 5000
McNemar's Statistic: 1746.5749, p-value: < **0.0001**

This detailed breakdown for each environment when comparing GPS to TempoRL shows the specific data underlying McNemar's tests.

**Results: GPS vs. DAR**

Table 13 summarizes the Average Success Rates (ASR) for GPS and DAR, along with the McNemar test statistics and p-values derived from common evaluation episodes.

All comparisons in Table 13 yield p-values below 0.0001, demonstrating highly statistically significant improvements of GPS over the DAR baseline across all tested maze environments. These consistent outcomes and extremely low p-values robustly support the conclusion that GPS offers superior performance compared to the DAR baseline under these experimental conditions.

Table 13: McNemar's Test Results for GPS vs. DAR. **All differences are statistically significant ($p < 0.0001$).** Results favor GPS in all listed cases.

| Environment | ASR (GPS) | ASR (DAR) | McNemar Stat. |
|---|---|---|---|
| 8×8 | **1.00** | 0.74 | 1385.00 |
| 16×16 | **1.00** | 0.66 | 1749.00 |
| 16×16 w/ 15% obs. | **0.94** | 0.58 | 322.84 |
| 16×16 w/ 25% obs. | **0.57** | 0.16 | 584.31 |
| 16×16 rooms | **0.85** | 0.13 | 2066.76 |
| 16×16 corridors | **0.98** | 0.61 | 1020.16 |
| 24×24 | **0.94** | 0.29 | 3029.20 |
| 24×24 w/ 15% obs. | **0.83** | 0.12 | 3357.55 |
| 24×24 w/ 25% obs. | **0.49** | 0.05 | 1953.79 |

The following sections provide the detailed per-run summaries logged by the script and the specific contingency tables used for McNemar's test for each maze configuration when comparing GPS with DAR.

**Maze: 8x8**

- *GPS (Algorithm A) Summary:* Total episodes: 4978, Successes: 4978, Failures: 0, Errors: 0

- *DAR (Algorithm B) Summary:* Total episodes: 4978, Successes: 3593, Failures: 1385, Errors: 0

*Contingency Table:*

| | | DAR (Alg. B) | |
|---|---|---|---|
| | | **Success** | **Failure** |
| **GPS (Alg. A)** | *Success* | 3593 (a) | 1385 (b) |
| | *Failure* | 0 (c) | 0 (d) |

Common episodes for comparison: 4978
McNemar's Statistic: 1385.0000, p-value: < **0.0001**

**Maze: 16x16**

- *GPS (Algorithm A) Summary:* Total episodes: 4994, Successes: 4994, Failures: 0, Errors: 0

- *DAR (Algorithm B) Summary:* Total episodes: 4994, Successes: 3245, Failures: 1749, Errors: 0

*Contingency Table:*

| | | DAR (Alg. B) | |
|---|---|---|---|
| | | **Success** | **Failure** |
| **GPS (Alg. A)** | *Success* | 3245 (a) | 1749 (b) |
| | *Failure* | 0 (c) | 0 (d) |

Common episodes for comparison: 4994
McNemar's Statistic: 1749.0000, p-value: < **0.0001**

**Maze: 16x16 Obstacles 15%**

- *GPS (Algorithm A) Summary:* Total episodes: 1050, Successes: 981, Failures: 69, Errors: 0

- *DAR (Algorithm B) Summary:* Total episodes: 1050, Successes: 599, Failures: 451, Errors: 0

*Contingency Table:*

|  |  | DAR (Alg. B) | |
|  |  | **Success** | **Failure** |
| **GPS (Alg. A)** | *Success* | 564 (a) | 417 (b) |
|  | *Failure* | 35 (c) | 34 (d) |

Common episodes for comparison: 1050
McNemar's Statistic: 322.8407, p-value: $<$ **0.0001**

**Maze: 16x16 Obstacles 25%**

- *GPS (Algorithm A) Summary:* Total episodes: 2010, Successes: 1143, Failures: 867, Errors: 0

- *DAR (Algorithm B) Summary:* Total episodes: 2010, Successes: 319, Failures: 1691, Errors: 0

*Contingency Table:*

|  |  | DAR (Alg. B) | |
|  |  | **Success** | **Failure** |
| **GPS (Alg. A)** | *Success* | 150 (a) | 993 (b) |
|  | *Failure* | 169 (c) | 698 (d) |

Common episodes for comparison: 2010
McNemar's Statistic: 584.3167, p-value: $<$ **0.0001**

**Maze: 16x16 Rooms**

- *GPS (Algorithm A) Summary:* Total episodes: 2930, Successes: 2496, Failures: 434, Errors: 0

- *DAR (Algorithm B) Summary:* Total episodes: 2930, Successes: 330, Failures: 2600, Errors: 0

*Contingency Table:*

|  |  | DAR (Alg. B) | |
|  |  | **Success** | **Failure** |
| **GPS (Alg. A)** | *Success* | 278 (a) | 2218 (b) |
|  | *Failure* | 52 (c) | 382 (d) |

Common episodes for comparison: 2930
McNemar's Statistic: 2066.7648, p-value: $<$ **0.0001**

**Maze: 16x16 Corridors**

- *GPS (Algorithm A) Summary:* Total episodes: 2725, Successes: 2671, Failures: 54, Errors: 0

- *DAR (Algorithm B) Summary:* Total episodes: 2725, Successes: 1594, Failures: 1131, Errors: 0

*Contingency Table:*

|  | | **DAR (Alg. B)** | |
| --- | --- | --- | --- |
|  | | **Success** | **Failure** |
| **GPS (Alg. A)** | *Success* | 1564 (a) | 1107 (b) |
|  | *Failure* | 30 (c) | 24 (d) |

Common episodes for comparison: 2725

McNemar's Statistic: 1020.1662, p-value: $<$ **0.0001**

## Maze: 24x24

- *GPS (Algorithm A) Summary:* Total episodes: 4999, Successes: 4675, Failures: 324, Errors: 0

- *DAR (Algorithm B) Summary:* Total episodes: 4999, Successes: 1470, Failures: 3529, Errors: 0

*Contingency Table:*

|  | | **DAR (Alg. B)** | |
| --- | --- | --- | --- |
|  | | **Success** | **Failure** |
| **GPS (Alg. A)** | *Success* | 1377 (a) | 3298 (b) |
|  | *Failure* | 93 (c) | 231 (d) |

Common episodes for comparison: 4999

McNemar's Statistic: 3029.2023, p-value: $<$ **0.0001**

## Maze: 24x24 Obstacles 15%

- *GPS (Algorithm A) Summary:* Total episodes: 4997, Successes: 4098, Failures: 899, Errors: 0

- *DAR (Algorithm B) Summary:* Total episodes: 4997, Successes: 571, Failures: 4426, Errors: 0

*Contingency Table:*

|  | | **DAR (Alg. B)** | |
| --- | --- | --- | --- |
|  | | **Success** | **Failure** |
| **GPS (Alg. A)** | *Success* | 482 (a) | 3616 (b) |
|  | *Failure* | 89 (c) | 810 (d) |

Common episodes for comparison: 4997

McNemar's Statistic: 3357.5517, p-value: $<$ **0.0001**

## Maze: 24x24 Obstacles 25%

- *GPS (Algorithm A) Summary:* Total episodes: 5000, Successes: 2443, Failures: 2557, Errors: 0

- *DAR (Algorithm B) Summary:* Total episodes: 5000, Successes: 252, Failures: 4748, Errors: 0

*Contingency Table:*

|  | | **DAR (Alg. B)** | |
| --- | --- | --- | --- |
|  | | **Success** | **Failure** |
| **GPS (Alg. A)** | *Success* | 119 (a) | 2324 (b) |
|  | *Failure* | 133 (c) | 2424 (d) |

Common episodes for comparison: 5000

McNemar's Statistic: 1953.7977, p-value: $<$ **0.0001**

This detailed breakdown for each environment when comparing GPS to DAR shows the specific data underlying McNemar's tests.

### H.1 Paired t-test Analysis

To complement the episode-level McNemar's test analysis, we conducted paired t-tests to assess the statistical significance of the differences in Average Success Rates (ASR) between our proposed method (GPS) and baselines. This analysis treats each random seed as an independent experimental unit.

**Methodology**

The paired t-test is a parametric test used to determine whether the mean difference between two sets of paired observations is significantly different from zero. In our context, each of the five random seeds provides a paired observation: the ASR achieved by GPS and the ASR achieved by a baseline algorithm, when trained and evaluated under identical conditions.

For each maze environment, we computed:

- The ASR for each algorithm across five independent seeds

- The difference in ASR between GPS and the baseline algorithm for each seed

- The mean and standard deviation of these differences

The null hypothesis ($H_0$) is that the mean difference in ASR between GPS and baseline algorithm is zero. The test statistic is calculated as:

$$t = \frac{\bar{d}}{s_d/\sqrt{n}}$$

where $\bar{d}$ is the mean of the paired differences, $s_d$ is the standard deviation of the differences, and $n$ is the number of pairs (seeds).

The significance level was set at $\alpha = 0.05$. We also report Cohen's $d$ as a measure of effect size, calculated as:

$$d = \frac{\bar{d}}{s_d}$$

Effect sizes are interpreted as small ($d \approx 0.2$), medium ($d \approx 0.5$), or large ($d \geq 0.8$).

**Results: GPS vs. DQN**

The paired t-test results corroborate the McNemar's test findings, demonstrating statistically significant improvements of GPS over DQN in the majority of tested environments.

The following sections provide detailed results for each maze configuration.

**Environment: 8×8:**

| GPS (%) | DQN (%) | Difference (%) |
|---|---|---|
| 100.00 | 94.60 | +5.40 |
| 100.00 | 84.80 | +15.20 |
| 100.00 | 84.60 | +15.40 |
| 100.00 | 81.30 | +18.70 |
| 100.00 | 77.80 | +22.20 |
| **Summary Statistics** | | |
| $100.00 \pm 0.00$ | $84.62 \pm 6.27$ | +15.38 (mean) |
| **Paired t-test Results** | | |

$t$-statistic = 5.4864, $p$-value = 0.0054
df = 4, $\alpha = 0.05$ (**significant**)
Cohen's $d$ = 2.4536 (large effect)

**Environment: 16×16:**

| GPS (%) | DQN (%) | Difference (%) |
|---------|---------|----------------|
| 100.00 | 69.90 | +30.10 |
| 100.00 | 67.00 | +33.00 |
| 100.00 | 81.30 | +18.70 |
| 100.00 | 74.00 | +26.00 |
| 100.00 | 75.80 | +24.20 |

**Summary Statistics**

| | | |
|---------|---------|----------------|
| $100.00 \pm 0.00$ | $73.60 \pm 5.51$ | +26.40 (mean) |

**Paired t-test Results**

$t$-statistic $= 10.7093$, $p$-value $= 0.0004$
df $= 4$, $\alpha = 0.05$ (**significant**))
Cohen's $d = 4.7893$ (large effect)

**Environment: 16×16 Obstacles 15%:**

| GPS (%) | DQN (%) | Difference (%) |
|---------|---------|----------------|
| 91.90 | 84.80 | +7.10 |
| 88.60 | 87.60 | +1.00 |
| 97.10 | 81.40 | +15.70 |
| 93.30 | 80.00 | +13.30 |
| 97.10 | 86.20 | +10.90 |

**Summary Statistics**

| | | |
|---------|---------|----------------|
| $93.60 \pm 3.62$ | $84.00 \pm 3.21$ | +9.60 (mean) |

**Paired t-test Results**

$t$-statistic $= 3.7255$, $p$-value $= 0.0204$
df $= 4$, $\alpha = 0.05$ (**significant**))
Cohen's $d = 1.6661$ (large effect)

**Environment: 16×16 Obstacles 25%:**

| GPS (%) | DQN (%) | Difference (%) |
|---------|---------|----------------|
| 48.30 | 79.60 | -31.30 |
| 50.50 | 72.10 | -21.60 |
| 81.60 | 63.90 | +17.70 |
| 41.80 | 71.40 | -29.60 |
| 62.70 | 73.90 | -11.20 |

**Summary Statistics**

| | | |
|---------|---------|----------------|
| $56.98 \pm 15.71$ | $72.18 \pm 5.64$ | $-15.20$ (mean) |

**Paired t-test Results**

$t$-statistic $= -1.6963$, $p$-value $= 0.1651$
df $= 4$, $\alpha = 0.05$ (**not significant**))
Cohen's $d = -0.7586$ (medium effect)

**Environment: 16×16 Rooms:**

| GPS (%) | DQN (%) | Difference (%) |
|---------|---------|----------------|
| 91.00 | 64.70 | +26.30 |
| 80.70 | 61.10 | +19.60 |
| 80.40 | 66.20 | +14.20 |
| 79.70 | 64.00 | +15.70 |
| 94.20 | 66.20 | +28.00 |
| **Summary Statistics** | | |
| $85.20 \pm 6.86$ | $64.44 \pm 2.10$ | +20.76 (mean) |
| **Paired t-test Results** | | |
| $t$-statistic $= 7.5035$, $p$-value $= 0.0017$ df $= 4$, $\alpha = 0.05$ (**significant**)) Cohen's $d = 3.3557$ (large effect) | | |

**Environment: 16×16 Corridors:**

| GPS (%) | DQN (%) | Difference (%) |
|---------|---------|----------------|
| 99.10 | 80.40 | +18.70 |
| 93.80 | 80.40 | +13.40 |
| 99.60 | 84.20 | +15.40 |
| 100.00 | 77.60 | +22.40 |
| 99.40 | 79.80 | +19.60 |
| **Summary Statistics** | | |
| $98.38 \pm 2.58$ | $80.48 \pm 2.38$ | +17.90 (mean) |
| **Paired t-test Results** | | |
| $t$-statistic $= 11.2894$, $p$-value $= 0.0004$ df $= 4$, $\alpha = 0.05$ (**significant**)) Cohen's $d = 5.0488$ (large effect) | | |

**Environment: 24×24:**

| GPS (%) | DQN (%) | Difference (%) |
|---------|---------|----------------|
| 100.00 | 24.20 | +75.80 |
| 77.40 | 27.00 | +50.40 |
| 100.00 | 31.90 | +68.10 |
| 99.90 | 31.30 | +68.60 |
| 90.30 | 25.10 | +65.20 |
| **Summary Statistics** | | |
| $93.52 \pm 9.94$ | $27.90 \pm 3.53$ | +65.62 (mean) |
| **Paired t-test Results** | | |
| $t$-statistic $= 15.6753$, $p$-value $= 0.0001$ df $= 4$, $\alpha = 0.05$ (**significant**)) Cohen's $d = 7.0102$ (large effect) | | |

**Environment: 24×24 Obstacles 15%:**

| GPS (%) | DQN (%) | Difference (%) |
|---------|---------|----------------|
| 81.40 | 15.30 | +66.10 |
| 82.90 | 10.80 | +72.10 |
| 82.50 | 12.70 | +69.80 |
| 83.80 | 10.70 | +73.10 |
| 83.70 | 13.30 | +70.40 |
| **Summary Statistics** | | |
| $82.86 \pm 0.98$ | $12.56 \pm 1.91$ | +70.30 (mean) |
| **Paired t-test Results** | | |

$t$-statistic $= 58.4011$, $p$-value $= 0.0000$
df $= 4$, $\alpha = 0.05$ (**significant**))
Cohen's $d = 26.1178$ (large effect)

**Environment: 24×24 Obstacles 25%:**

| GPS (%) | DQN (%) | Difference (%) |
|---------|---------|----------------|
| 50.50 | 11.10 | +39.40 |
| 22.90 | 10.10 | +12.80 |
| 63.10 | 9.20 | +53.90 |
| 58.70 | 11.60 | +47.10 |
| 51.40 | 9.20 | +42.20 |
| **Summary Statistics** | | |
| $49.32 \pm 15.67$ | $10.24 \pm 1.09$ | +39.08 (mean) |
| **Paired t-test Results** | | |

$t$-statistic $= 5.5703$, $p$-value $= 0.0051$
df $= 4$, $\alpha = 0.05$ (**significant**))
Cohen's $d = 2.4911$ (large effect)

**Results: GPS vs. TempoRL**

The paired t-test results corroborate the McNemar's test findings, demonstrating statistically significant improvements of GPS over TempoRL in the majority of tested environments.

The following sections provide detailed results for each maze configuration.

**Environment: 8×8:**

| GPS (%) | TEMPORL (%) | Difference (%) |
|---|---|---|
| 100.00 | 98.80 | +1.20 |
| 100.00 | 91.00 | +9.00 |
| 100.00 | 92.60 | +7.40 |
| 100.00 | 95.20 | +4.80 |
| 100.00 | 92.80 | +7.20 |
| **Summary Statistics** | | |
| $100.00 \pm 0.00$ | $94.08 \pm 3.04$ | +5.92 (mean) |
| **Paired t-test Results** | | |
| $t$-statistic $= 4.3614$, $p$-value $= 0.0120$ | | |
| df $= 4$, $\alpha = 0.05$ (**significant**)) | | |
| Cohen's $d = 1.9505$ (large effect) | | |

**Environment: 16×16:**

| GPS (%) | TEMPORL (%) | Difference (%) |
|---|---|---|
| 100.00 | 84.80 | +15.20 |
| 100.00 | 86.80 | +13.20 |
| 100.00 | 93.60 | +6.40 |
| 100.00 | 91.60 | +8.40 |
| 100.00 | 82.20 | +17.80 |
| **Summary Statistics** | | |
| $100.00 \pm 0.00$ | $87.80 \pm 4.73$ | +12.20 (mean) |
| **Paired t-test Results** | | |
| $t$-statistic $= 5.7691$, $p$-value $= 0.0045$ | | |
| df $= 4$, $\alpha = 0.05$ (**significant**)) | | |
| Cohen's $d = 2.5800$ (large effect) | | |

**Environment: 16×16 Obstacles 15%:**

| GPS (%) | TEMPORL (%) | Difference (%) |
|---|---|---|
| 91.90 | 81.90 | +10.00 |
| 88.60 | 79.00 | +9.60 |
| 97.10 | 79.00 | +18.10 |
| 93.30 | 80.00 | +13.30 |
| 97.10 | 80.00 | +17.10 |
| **Summary Statistics** | | |
| $93.60 \pm 3.62$ | $79.98 \pm 1.18$ | +13.62 (mean) |
| **Paired t-test Results** | | |
| $t$-statistic $= 7.7640$, $p$-value $= 0.0015$ | | |
| df $= 4$, $\alpha = 0.05$ (**significant**)) | | |
| Cohen's $d = 3.4722$ (large effect) | | |

**Environment: 16×16 Obstacles 25%:**

| GPS (%) | TEMPORL (%) | Difference (%) |
|---|---|---|
| 48.30 | 79.10 | -30.80 |
| 50.50 | 74.10 | -23.60 |
| 81.60 | 63.70 | +17.90 |
| 41.80 | 67.20 | -25.40 |
| 62.70 | 79.60 | -16.90 |

**Summary Statistics**

| | | |
|---|---|---|
| $56.98 \pm 15.71$ | $72.74 \pm 7.10$ | $-15.76$ (mean) |

**Paired t-test Results**

$t$-statistic = -1.8108, $p$-value = 0.1444
df = 4, $\alpha = 0.05$ (**not significant**))
Cohen's $d$ = -0.8098 (large effect)

**Environment: 16×16 Rooms:**

| GPS (%) | TEMPORL (%) | Difference (%) |
|---|---|---|
| 91.00 | 62.50 | +28.50 |
| 80.70 | 55.60 | +25.10 |
| 80.40 | 74.10 | +6.30 |
| 79.70 | 72.70 | +7.00 |
| 94.20 | 68.60 | +25.60 |

**Summary Statistics**

| | | |
|---|---|---|
| $85.20 \pm 6.86$ | $66.70 \pm 7.67$ | $+18.50$ (mean) |

**Paired t-test Results**

$t$-statistic = 3.7959, $p$-value = 0.0192
df = 4, $\alpha = 0.05$ (**significant**))
Cohen's $d$ = 1.6976 (large effect)

**Environment: 16×16 Corridors:**

| GPS (%) | TEMPORL (%) | Difference (%) |
|---|---|---|
| 99.10 | 89.70 | +9.40 |
| 93.80 | 89.40 | +4.40 |
| 99.60 | 93.90 | +5.70 |
| 100.00 | 85.10 | +14.90 |
| 99.40 | 88.30 | +11.10 |

**Summary Statistics**

| | | |
|---|---|---|
| $98.38 \pm 2.58$ | $89.28 \pm 3.16$ | $+9.10$ (mean) |

**Paired t-test Results**

$t$-statistic = 4.8169, $p$-value = 0.0085
df = 4, $\alpha = 0.05$ (**significant**))
Cohen's $d$ = 2.1542 (large effect)

**Environment: 24×24:**

| GPS (%) | TEMPORL (%) | Difference (%) |
|---------|-------------|----------------|
| 100.00 | 50.60 | +49.40 |
| 77.40 | 52.20 | +25.20 |
| 100.00 | 60.80 | +39.20 |
| 99.90 | 46.40 | +53.50 |
| 90.30 | 51.20 | +39.10 |

**Summary Statistics**

| | | |
|---------|-------------|----------------|
| $93.52 \pm 9.94$ | $52.24 \pm 5.27$ | +41.28 (mean) |

**Paired t-test Results**

$t$-statistic = 8.4008, $p$-value = 0.0011
df = 4, $\alpha = 0.05$ (**significant**))
Cohen's $d$ = 3.7570 (large effect)

**Environment: 24×24 Obstacles 15%:**

| GPS (%) | TEMPORL (%) | Difference (%) |
|---------|-------------|----------------|
| 81.40 | 20.30 | +61.10 |
| 82.90 | 12.80 | +70.10 |
| 82.50 | 18.20 | +64.30 |
| 83.80 | 17.60 | +66.20 |
| 83.70 | 17.60 | +66.10 |

**Summary Statistics**

| | | |
|---------|-------------|----------------|
| $82.86 \pm 0.98$ | $17.30 \pm 2.75$ | +65.56 (mean) |

**Paired t-test Results**

$t$-statistic = 44.8201, $p$-value = 0.0000
df = 4, $\alpha = 0.05$ (**significant**))
Cohen's $d$ = 20.0442 (large effect)

**Environment: 24×24 Obstacles 25%:**

| GPS (%) | TEMPORL (%) | Difference (%) |
|---------|-------------|----------------|
| 50.50 | 7.80 | +42.70 |
| 22.90 | 8.00 | +14.90 |
| 63.10 | 6.00 | +57.10 |
| 58.70 | 7.80 | +50.90 |
| 51.40 | 10.40 | +41.00 |

**Summary Statistics**

| | | |
|---------|-------------|----------------|
| $49.32 \pm 15.67$ | $8.00 \pm 1.57$ | +41.32 (mean) |

**Paired t-test Results**

$t$-statistic = 5.7278, $p$-value = 0.0046
df = 4, $\alpha = 0.05$ (**significant**))
Cohen's $d$ = 2.5616 (large effect)

### Results: GPS vs. DAR

The paired t-test results corroborate the McNemar's test findings, demonstrating statistically significant improvements of GPS over DAR across all of the tested environments.

The following sections provide detailed results for each maze configuration.

**Environment: 8×8:**

| GPS (%) | DAR (%) | Difference (%) |
|---|---|---|
| 100.00 | 76.60 | +23.40 |
| 100.00 | 72.20 | +27.80 |
| 100.00 | 78.60 | +21.40 |
| 100.00 | 70.20 | +29.80 |
| 100.00 | 71.20 | +28.80 |

**Summary Statistics**

| | | |
|---|---|---|
| $100.00 \pm 0.00$ | $73.76 \pm 3.65$ | +26.24 (mean) |

**Paired t-test Results**

$t$-statistic $= 16.0960$, $p$-value $= 0.0001$
df $= 4$, $\alpha = 0.05$ (**significant**))
Cohen's $d = 7.1984$ (large effect)

**Environment: 16×16:**

| GPS (%) | DAR (%) | Difference (%) |
|---|---|---|
| 100.00 | 66.40 | +33.60 |
| 100.00 | 60.20 | +39.80 |
| 100.00 | 68.60 | +31.40 |
| 100.00 | 70.40 | +29.60 |
| 100.00 | 65.20 | +34.80 |

**Summary Statistics**

| | | |
|---|---|---|
| $100.00 \pm 0.00$ | $66.16 \pm 3.89$ | +33.84 (mean) |

**Paired t-test Results**

$t$-statistic $= 19.4676$, $p$-value $= 0.0000$
df $= 4$, $\alpha = 0.05$ (**significant**))
Cohen's $d = 8.7062$ (large effect)

**Environment: 16×16 Obstacles 15%:**

| GPS (%) | DAR (%) | Difference (%) |
|---|---|---|
| 91.90 | 63.80 | +28.10 |
| 88.60 | 55.70 | +32.90 |
| 97.10 | 55.20 | +41.90 |
| 93.30 | 52.40 | +40.90 |
| 97.10 | 61.40 | +35.70 |

**Summary Statistics**

| | | |
|---|---|---|
| $93.60 \pm 3.62$ | $57.70 \pm 4.72$ | +35.90 (mean) |

**Paired t-test Results**

$t$-statistic $= 14.0337$, $p$-value $= 0.0001$
df $= 4$, $\alpha = 0.05$ (**significant**))
Cohen's $d = 6.2761$ (large effect)

**Environment: 16×16 Obstacles 25%:**

| GPS (%) | DAR (%) | Difference (%) |
|---|---|---|
| 48.30 | 13.90 | +34.40 |
| 50.50 | 19.40 | +31.10 |
| 81.60 | 11.90 | +69.70 |
| 41.80 | 19.40 | +22.40 |
| 62.70 | 13.90 | +48.80 |

**Summary Statistics**

| | | |
|---|---|---|
| $56.98 \pm 15.71$ | $15.70 \pm 3.47$ | +41.28 (mean) |

**Paired t-test Results**

$t$-statistic $= 4.9846$, $p$-value $= 0.0076$
df $= 4$, $\alpha = 0.05$ (**significant**))
Cohen's $d = 2.2292$ (large effect)

**Environment: 16×16 Rooms:**

| GPS (%) | DAR (%) | Difference (%) |
|---|---|---|
| 91.00 | 14.70 | +76.30 |
| 80.70 | 12.60 | +68.10 |
| 80.40 | 19.80 | +60.60 |
| 79.70 | 8.50 | +71.20 |
| 94.20 | 8.90 | +85.30 |

**Summary Statistics**

| | | |
|---|---|---|
| $85.20 \pm 6.86$ | $12.90 \pm 4.64$ | +72.30 (mean) |

**Paired t-test Results**

$t$-statistic $= 17.5163$, $p$-value $= 0.0001$
df $= 4$, $\alpha = 0.05$ (**significant**))
Cohen's $d = 7.8335$ (large effect)

**Environment: 16×16 Corridors:**

| GPS (%) | DAR (%) | Difference (%) |
|---|---|---|
| 99.10 | 62.80 | +36.30 |
| 93.80 | 67.20 | +26.60 |
| 99.60 | 59.40 | +40.20 |
| 100.00 | 57.40 | +42.60 |
| 99.40 | 59.10 | +40.30 |

**Summary Statistics**

| | | |
|---|---|---|
| $98.38 \pm 2.58$ | $61.18 \pm 3.89$ | +37.20 (mean) |

**Paired t-test Results**

$t$-statistic $= 13.1137$, $p$-value $= 0.0002$
df $= 4$, $\alpha = 0.05$ (**significant**))
Cohen's $d = 5.8646$ (large effect)

**Environment: 24×24:**

| GPS (%) | DAR (%) | Difference (%) |
|---|---|---|
| 100.00 | 30.20 | +69.80 |
| 77.40 | 32.00 | +45.40 |
| 100.00 | 27.80 | +72.20 |
| 99.90 | 31.60 | +68.30 |
| 90.30 | 25.80 | +64.50 |
| **Summary Statistics** | | |
| $93.52 \pm 9.94$ | $29.48 \pm 2.63$ | +64.04 (mean) |
| **Paired t-test Results** | | |
| $t$-statistic $= 13.2731$, $p$-value $= 0.0002$ df $= 4$, $\alpha = 0.05$ (**significant**)) Cohen's $d = 5.9359$ (large effect) | | |

**Environment: 24×24 Obstacles 15%:**

| GPS (%) | DAR (%) | Difference (%) |
|---|---|---|
| 83.50 | 11.60 | +71.90 |
| 82.90 | 11.60 | +71.30 |
| 82.50 | 9.60 | +72.90 |
| 83.80 | 10.80 | +73.00 |
| 83.70 | 14.20 | +69.50 |
| **Summary Statistics** | | |
| $83.28 \pm 0.56$ | $11.56 \pm 1.69$ | +71.72 (mean) |
| **Paired t-test Results** | | |
| $t$-statistic $= 112.2270$, $p$-value $= 0.0000$ df $= 4$, $\alpha = 0.05$ (**significant**)) Cohen's $d = 50.1894$ (large effect) | | |

**Environment: 24×24 Obstacles 25%:**

| GPS (%) | DAR (%) | Difference (%) |
|---|---|---|
| 50.50 | 6.80 | +43.70 |
| 23.60 | 5.20 | +18.40 |
| 63.10 | 4.20 | +58.90 |
| 58.70 | 4.60 | +54.10 |
| 51.40 | 5.00 | +46.40 |
| **Summary Statistics** | | |
| $49.46 \pm 15.37$ | $5.16 \pm 0.99$ | +44.30 (mean) |
| **Paired t-test Results** | | |
| $t$-statistic $= 6.3132$, $p$-value $= 0.0032$ df $= 4$, $\alpha = 0.05$ (**significant**)) Cohen's $d = 2.8233$ (large effect) | | |

# I  Baselines and Architecture

Each baseline is evaluated using a grid search over multiple hyperparameter configurations; Tables 14, 15, 16, 17, 18, 19 and 22 detail the specific value ranges for these parameters.

All baseline models employ the same CNN feature extractor architecture followed by similarly sized linear layers, differing only in the final output layer size. For example, DQN outputs 4 Q-values (one per action), while DAR outputs 12 (4 actions $\times$ 3 repetition heads). TempoRL requires an additional network head to implement the skip policy, adding architectural complexity but gaining flexibility in temporal decision-making. In our GPS method, the actor and critic networks each have their own separate CNN state feature extractors. In future work, we plan to explore a shared CNN feature extraction architecture as implemented in TempoRL, which could potentially improve computational efficiency and state representation learning.

For TempoRL, we configured the model with a maximum skip length between 1..10 to allow variable sequence lengths of action repetition. For DAR, we evaluated possible coarse control values of 1,5,10 to allow the same maximum sequence length and mid-sequence capability, with the fine control value fixed at 1 to allow for actions at every time step. We based our implementations on the publicly available code at `https://github.com/automl/TempoRL` but reimplemented from scratch to enrich with more metrics and employ our evaluation methodology. Detailed architectures, hyperparameter configurations, and implementation specifics can be found in Appendix J, K, L, M and N.

While GPS, like hierarchical methods, targets long-horizon tasks, its approach is fundamentally different. Hierarchical RL learns temporal abstractions via skill discovery and multi-level policies, whereas GPS generates full action sequences directly from state observations without skills or sub-goal decomposition. The proto-sequence decoder is pre-trained and fixed, providing a structured output space rather than a learned controller. Because of these differences and GPS 's focus on discrete action spaces, we compare against DAR and TempoRL, which similarly extend temporal horizons through action repetition and sequence commitment.

## J  DQN Baseline Implementation Details

This section outlines the architecture and configuration of the Deep Q-Network (DQN) agent used as a baseline. It details the neural network structure, hyperparameter settings, exploration strategy, optimization method, and other relevant training aspects.

### J.1  Model Architecture (QNetwork)

The Q-Network is a neural network designed to approximate the action-value function $Q(s, a)$. It consists of a convolutional part for feature extraction from the input observation and a linear part for producing Q-values for each action.

In our maze environments, as depicted in Subsection 4.3, the input observation has a shape $(C, H, W)$, where the number of input channels $C$ is 3. The number of output channels, $n\_output\_channels$, corresponds to the number of available actions, which is 4 (right, left, up, down).

#### J.1.1  Convolutional Neural Network (CNN) Part

The CNN component processes the input observation through a sequence of convolutional layers:

1. **Conv2D Layer 1**:
   - Input channels: 3
   - Output channels: 16
   - Kernel size: 2
   - Stride: 1

2. **Activation**: ReLU

3. **Conv2D Layer 2**:
   - Input channels: 16
   - Output channels: 32

- Kernel size: 2
- Stride: 1

4. **Activation**: ReLU

5. **Conv2D Layer 3**:

    - Input channels: 32
    - Output channels: 64
    - Kernel size: 2
    - Stride: 1

6. **Activation**: ReLU

7. **Flatten Layer**: The output of the convolutional layers is flattened into a 1D vector. The size of this vector, $n\_flatten$, is computed automatically.

### J.1.2 Linear Part

The flattened output from the CNN ($n\_flatten$) is fed into a sequence of fully connected linear layers:

- The hidden layer sizes are configurable via grid search (see Table 14 for details). The activation function for these hidden layers is Leaky ReLU (negative slope 0.1).

- The final linear layer maps the last hidden layer's output to $n\_output\_channels$ (4 actions).

### J.2 Hyperparameters

The agent's behavior and training process are governed by a set of hyperparameters, detailed in Tables 14, 15, and 16.

Table 14: General Experiment Hyperparameters for DQN Baseline

| Parameter | Default Value |
|---|---|
| `seed` | 123 |
| `torch_deterministic` | True |
| `save_model_strategy` | SUCCESS_RATE |
| `val_eval_freq` | 5000 |
| `train_eval_freq` | 5000 |
| `eval_test_dataset_training_freq` | 100000 |

Table 15: Environment-Specific Hyperparameters for DQN Baseline

| Parameter | Default Value |
|---|---|
| max_episode_steps | 75 |
| reward_strategy | NEGATIVE_BASED_ON_MAX_LEVEL_WITH_PENALTIES |
| observation_encoding_strategy | DEFAULT |
| Max Path Length (*max_level*) | Varies (see Env. Def. in Table 1) |
| Min Path Length (*start_level*) | Varies (see Env. Def. in Table 1) |

### J.3 Epsilon-Greedy Exploration

The agent uses an epsilon-greedy strategy for action selection. The value of epsilon ($\epsilon$) is linearly annealed from `start_e` (1.0) to `end_e` (0.1) over a `duration`. This duration is calculated as $\lfloor \exp\_frac \times$ `total_timesteps`$\rfloor$, where exp_frac is the selected `exploration_fraction` (from options in Table 16) and $t$ is the current global timestep. The epsilon at timestep $t$ is:

$$\epsilon_t = \max((((\text{end\_e} - \text{start\_e})/\text{duration}) \times t + \text{start\_e}), \text{end\_e})$$

With probability $\epsilon_t$, a random action is chosen; otherwise, the action with the highest Q-value is selected.

Table 16: Algorithm Specific Hyperparameters for DQN Baseline

| Parameter | Default Value / Options |
|---|---|
| total_timesteps | Environment specific |
| learning_rate | $[1 \times 10^{-3}, 1 \times 10^{-4}]$ |
| buffer_size | $[10000, 50000]$ |
| $\gamma$ (discount factor) | 0.99 |
| $\tau$ (target update rate) | $[0.01, 0.005]$ |
| target_network_frequency | $[10, 100]$ (soft-target update freq.) |
| batch_size | 256 |
| start_e | 1.0 (initial $\varepsilon$) |
| end_e | 0.1 (final $\varepsilon$) |
| exploration_fraction | $[0.1, 0.3, 0.5]$ |
| learning_starts | 1000 (timestep to begin learning) |
| train_frequency | 2 (Q-network update freq.) |
| linear_layers | ["512,128,32", "512,32"] |
| activation_function | Leaky ReLU (slope 0.1) |

## J.4 Optimizer

The Q-Network is trained using the Adam optimizer (`torch.optim.Adam`). The learning rate is controlled by the `learning_rate` hyperparameter (see Table 16).

## J.5 Replay Buffer

A replay buffer (`stable_baselines3.common.buffers.ReplayBuffer`) stores experiences $(s_t, a_t, r_t, s_{t+1}, d_t)$.

The buffer size is specified in Table 16. Key configurations include `optimize_memory_usage = False` and `handle_timeout_termination = False`.

## J.6 Training Details

**Loss Function.** The Q-Network parameters ($\theta$) are updated by minimizing the Mean Squared Error (MSE) loss:

$$L(\theta) = \mathbb{E}_{(s,a,r,s',d)\sim\mathcal{B}} \left[ (y_t - Q(s,a;\theta))^2 \right]$$

where the TD target:

$$y_t = r_t + \gamma \max_{a'} Q_{target}(s_{t+1}, a'; \theta^-)(1 - d_t)$$

Here, $r_t$ is the reward, $\gamma$ is the discount factor, $Q_{target}$ is the target network with parameters $\theta^-$, and $d_t$ indicates if $s_{t+1}$ is terminal. This is implemented via `torch.nn.functional.mse_loss`.

**Target Network.** A separate target network $Q_{target}$ with parameters $\theta^-$ stabilizes training. Its weights are updated using Polyak averaging: $\theta^- \leftarrow \tau\theta + (1 - \tau)\theta^-$. The soft update rate $\tau$ and update frequency `target_network_frequency` are specified in Table 16.

**Training Procedure.**

- **Learning Starts**: Training begins after `learning starts` timesteps (see Table 16).

- **Training Frequency**: The Q-network is updated every `train_frequency` global steps (see Table 16).

- **Batch Size**: Number of experiences sampled per training step is `batch_size` (see Table 16).

## J.7 Evaluation

The agent's performance is evaluated periodically on validation and test datasets.

- Evaluation on the validation dataset occurs every `val_eval_freq` steps.

- Evaluation on the test dataset can occur during training every `eval_test_dataset_training_freq` steps.

- During evaluation, actions are chosen greedily (or with a small fixed epsilon, e.g., 0.05 or 0.0).

- Metrics logged include mean episodic return, success rate, and agent step ratio.

- Model saving is based on performance metrics (e.g., highest success rate or reward on validation) as per `save_model_strategy`.

# K  DAR Baseline Implementation Details

This section outlines the architecture and configuration of the Dyanmic Action Repetition (DAR) agent used as a baseline. The DAR agent builds upon the Deep Q-Network (DQN) architecture and training methodology. Therefore, for aspects not explicitly mentioned here, such as the general experiment configuration (Table 14), environment-specific arguments (Table 15), epsilon-greedy exploration strategy (Section J.3), optimizer (Section J.4), replay buffer (Section J.5), general training procedure (Section J.6), and evaluation methodology (Section J.7), please refer to the corresponding descriptions in the DQN baseline implementation details (Section J).

The primary distinctions of the DAR baseline are its modified network architecture to support an expanded action space and an additional algorithm-specific hyperparameter, `dar_r_l`, related to action repetition.

## K.1  Model Architecture

The DAR network for the DAR agent, similar to DQN, approximates the action-value function $Q(s, a)$. It comprises a convolutional part for feature extraction and a linear part for producing Q-values.

The input observation from the maze environments has a shape $(C, H, W)$, where $C = 3$, identical to the DQN baseline (Section J.1).

### K.1.1  Convolutional Neural Network (CNN) Part

The CNN component is identical to the one used in the DQN baseline. For details on the architecture (number of layers, channels, kernel sizes, strides, and activations), please refer to Section J.1. The output of this part is a flattened 1D vector of size $n\_flatten$.

### K.1.2  Linear Part

The flattened output ($n\_flatten$) from the CNN is processed by a sequence of fully connected linear layers:

- The hidden layer sizes are configurable via grid search, with the same options as the DQN baseline (see Table 17 for `linear_layers`). The activation function for these hidden layers is Leaky ReLU (negative slope 0.1).

- The final linear layer maps the last hidden layer's output to $n\_output\_channels$. For the DAR agent, $n\_output\_channels = 12$, corresponding to 4 base actions (right, left, up, down) each associated with 3 repetition heads/levels.

## K.2  Hyperparameters

The general experimental configuration and environment-specific hyperparameters for the DAR baseline are the same as those for the DQN baseline, as detailed in Table 14 and Table 15, respectively.

### K.2.1 Algorithm Specific Arguments

The algorithm-specific hyperparameters for the DAR baseline, including the newly introduced `dar_r_l` parameter, are listed in Table 17. These parameters are subject to grid search to find the optimal configuration for each environment.

Table 17: Algorithm Specific Hyperparameters for DAR Baseline

| Parameter | Default Value / Options |
|---|---|
| total_timesteps | Environment specific |
| learning_rate | $[1 \times 10^{-3}, 1 \times 10^{-4}]$ |
| buffer_size | [10000, 50000] |
| $\gamma$ (discount factor) | 0.99 |
| $\tau$ (target update rate) | [0.01, 0.005] |
| target_network_frequency | [10, 100] (soft-target update freq.) |
| batch_size | 256 |
| start_e | 1.0 (initial $\varepsilon$) |
| end_e | 0.1 (final $\varepsilon$) |
| exploration_fraction | [0.1, 0.3, 0.5] |
| learning_starts | 1000 (timestep to begin learning) |
| train_frequency | 2 (Q-network update freq.) |
| linear_layers | ["512,128,32", "512,32"] |
| activation_function | Leaky ReLU (slope 0.1) |
| dar_r_l | [1, 5, 10] (repetition level parameter) |

### K.3 Training Details

**Loss Function.** For DAR, the Q-Network parameters ($\theta$) are updated by minimizing Huber loss. This is implemented via torch.nn.SmoothL1Loss.

## L TempoRL Baseline Implementation Details

This section describes the architecture and configuration of the TempoRL agent, a baseline designed for temporal abstraction by learning how long to repeat actions. TempoRL shares several components and procedures with the DQN baseline. For details on the general experiment configuration (Table 14), environment-specific arguments (Table 15), replay buffer (Section J.5), and evaluation methodology (Section J.7), please refer to the corresponding descriptions in the DQN baseline implementation details (Section J).

Key distinctions of the TempoRL agent include its specialized network architecture with separate heads for action selection and skip duration, unique hyperparameters related to these mechanisms (`skip_dim`, `weight_sharing`), and the use of Huber loss for training.

### L.1 Model Architecture

The TempoRL network processes input observations to produce Q-values for primitive actions and Q-values for skip durations. The input observation from the maze environments has a shape $(C, H, W)$, where $C = 3$, identical to the DQN baseline (Section J.1).

### L.1.1 Convolutional Neural Network (CNN) Part

The CNN component used for initial feature extraction is identical to the one in the DQN baseline. For details on its architecture (number of layers, channels, kernel sizes, strides, and activations), please refer to Section J.1. The output of this CNN part is a flattened 1D vector of size $n\_flatten$.

### L.1.2 Linear Heads for Action and Skip Policies

Following the CNN, the network processes features through a structure that leads to two distinct output heads: one for action selection and one for determining the skip duration. The MLP for each pathway (from CNN output to pre-output layer) consists of layers with output units [512, 128, 32].

- **Feature Processing and Weight Sharing**:
  - If `weight_sharing = True` (default configuration): The $n\_flatten$ vector is first processed by a shared linear layer producing 512 output units, followed by a Leaky ReLU activation (negative slope 0.1). This 512-unit feature vector serves as the common input to the subsequent differing layers of the action and skip heads.
  - If `weight_sharing = False`: The $n\_flatten$ vector is independently fed into the first linear layer (512 output units, Leaky ReLU) of both the action and skip processing streams. Each stream then continues with its own [128, 32] layers.

- **Action Head**:
  - Starting from the 512-unit feature vector (either shared or head-specific), it is processed through two subsequent linear layers with 128 and 32 output units, respectively. Each of these hidden layers uses a Leaky ReLU activation (negative slope 0.1).
  - The final linear layer of the action head maps the 32-unit feature vector to $n\_output\_actions$ Q-values, where $n\_output\_actions = 4$ (corresponding to right, left, up, down).

- **Skip Head**:
  - Similarly starting from the 512-unit feature vector, it is processed through two subsequent linear layers with 128 and 32 output units, each followed by a Leaky ReLU activation (negative slope 0.1).
  - The final linear layer of the skip head maps the 32-unit feature vector to `skip_dim`. Each corresponds to the utility of repeating the chosen primitive action for a specific number of steps, from 1 up to `skip_dim`.

### L.2 Hyperparameters

General experimental configuration (Table 14) and environment-specific arguments (Table 15) are consistent with the DQN baseline. Algorithm-specific hyperparameters for TempoRL, including those unique to its architecture, are detailed in Table 18.

### L.3 Action Selection and Exploration

TempoRL employs a two-step $\epsilon$-greedy strategy for exploration and action selection:

1. **Primitive Action Selection**: Given the current state $s_t$, a primitive action $a_t$ (e.g., right, left, up, down) is chosen. With probability $\epsilon$, $a_t$ is selected randomly from the set of $n\_output\_actions$. Otherwise (with probability $1 - \epsilon$), $a_t = \mathrm{argmax}_{a'} Q(s_t, a'; \theta)$, where $Q(s_t, \cdot; \theta)$ are the Q-values produced by the action head of the online network.

2. **Skip Duration Selection**: Conditioned on the current state $s_t$ and the chosen primitive action $a_t$, a skip duration $k_t$ (number of times to repeat $a_t$, from 1 to `skip_dim`) is selected. With probability $\epsilon$, $k_t$ is chosen randomly from $\{1, \ldots, \text{skip\_dim}\}$. Otherwise, $k_t = \mathrm{argmax}_{k'} Q_{skip}(s_t, a_t, k'; \theta_{skip})$, where $Q_{skip}(s_t, a_t, \cdot; \theta_{skip})$ are the Q-values for different skip durations produced by the skip head (which might use shared parameters if `weight_sharing = True`).

The selected primitive action $a_t$ is then executed in the environment for $k_t$ consecutive timesteps. The value of $\epsilon$ is typically linearly annealed from `start_e` to `end_e` over `exploration_fraction` of total timesteps, as detailed for the DQN baseline (see Section J.3 and Table 18).

Table 18: Algorithm Specific Hyperparameters for TempoRL Baseline

| Parameter | Default Value / Options |
|---|---|
| total_timesteps | Environment specific |
| learning_rate | $[1 \times 10^{-3}, 1 \times 10^{-4}]$ |
| buffer_size | [10000, 50000] |
| gamma ($\gamma$) | 0.99 (discount factor) |
| tau ($\tau$) | [0.01, 0.005] (target network update rate) |
| target_network_frequency | [10, 100] (frequency of applying soft target network update) |
| batch_size | 256 |
| start_e | 1.0 (starting epsilon for exploration) |
| end_e | 0.1 (ending epsilon for exploration) |
| exploration_fraction | [0.1, 0.3, 0.5] |
| learning_starts | 1000 (timestep to start learning) |
| train_frequency | 2 (frequency of training the Q-network) |
| activation_function | Leaky ReLU (negative slope 0.1 for hidden layers) |
| skip_dim | 10 (maximum skip size) |
| weight_sharing | True (whether to share the first 512-unit layer) |

## L.4 Optimizer

Separate Adam optimizers (`torch.optim.Adam`) are used for the action Q-network parameters and the skip Q-network parameters. The learning rate for both optimizers is controlled by the `learning_rate` hyperparameter (see Table 18). Gradients for both networks are clipped ( `grad_clip_val = 40.0`).

## L.5 Replay Buffers

TempoRL utilizes two distinct replay buffers with capacity `buffer_size` (see Table 18) to store experiences for training its action and skip policies:

- **Action Replay Buffer**: This is a standard replay buffer ( `ReplayBuffer` from Stable Baselines3) that stores transitions corresponding to individual primitive actions. Each experience tuple is of the form $(s_t, a_t, r_t, s_{t+1}, d_t)$, where:
  - $s_t$: The state at time $t$.
  - $a_t$: The primitive action taken at time $t$.
  - $r_t$: The reward received at time $t + 1$.
  - $s_{t+1}$: The state at time $t + 1$.
  - $d_t$: A boolean flag indicating if $s_{t+1}$ is a terminal state.

  Experiences sampled from this buffer are used to train the action Q-network (the action head).

- **Skip Replay Buffer**: This is a custom replay buffer (referred to as `NoneConcatSkipReplayBuffer` in the implementation) specifically designed to store experiences related to the execution of multi-step skip actions. Each experience tuple is of the form $(s_j, k_j, s_{j+k_j}, R_j, d_{j+k_j}, k_j^{len}, a_j^{behav})$, representing:
  - $s_j$: The state from which the skip action (repeating $a_j^{behav}$) commenced.
  - $k_j$: The selected skip duration (i.e., the 'action' taken by the skip policy).
  - $s_{j+k_j}$: The state reached after the primitive action $a_j^{behav}$ was executed $k_j^{len}$ times.
  - $R_j$: The accumulated (and potentially discounted, depending on exact calculation before storage) reward received over the course of the $k_j^{len}$ steps of the skip.
  - $d_{j+k_j}$: A boolean flag indicating if $s_{j+k_j}$ (the state after the skip) is a terminal state.
  - $k_j^{len}$ (length): The actual number of steps the primitive action $a_j^{behav}$ was repeated (this is equivalent to $k_j$).

     – $a_j^{behav}$: The underlying primitive action that was chosen to be repeated for $k_j^{len}$ steps.

Experiences sampled from this buffer are used to train the skip Q-network (the skip head).

## L.6   Training Details

TempoRL involves separate training updates for the action Q-network and the skip Q-network, both utilizing the Huber loss function.

### L.6.1   Loss Function and Updates

The network parameters are updated by minimizing the Huber loss (This is implemented via torch.nn.SmoothL1Loss) for both action and skip predictions.

- **Action Q-Network Update**: Experiences $(s_j, a_j, r_j, s_{j+1}, d_j)$ are sampled from a standard replay buffer. The target value $y_j^{action}$ is computed using a Double DQN-style approach:

$$y_j^{action} = r_j + \gamma(1 - d_j)Q_{target}(s_{j+1}, \arg\max_{a'} Q(s_{j+1}, a'; \theta); \theta^-)$$

  where $Q$ is the online action Q-network with parameters $\theta$, and $Q_{target}$ is its target network with parameters $\theta^-$. The loss is then:

$$L_{action}(\theta) = \mathbb{E}_{(s_j, a_j, r_j, s_{j+1}, d_j) \sim \mathcal{B}} \left[ \text{HuberLoss}(y_j^{action} - Q(s_j, a_j; \theta)) \right]$$

- **Skip Q-Network Update**: Experiences $(s_j, a_j^{behav}, k_j, R_j, s_{j+k_j}, d_{j+k_j})$ are sampled from a separate replay buffer for skips. Here, $a_j^{behav}$ is the primitive action executed, $k_j$ is the skip duration (number of times $a_j^{behav}$ was repeated), $R_j$ is the accumulated discounted reward during these $k_j$ steps, and $s_{j+k_j}$ is the state after $k_j$ steps. The target value $y_j^{skip}$ is calculated as:

$$y_j^{skip} = R_j + \gamma^{k_j}(1 - d_{j+k_j})Q_{target}(s_{j+k_j}, \arg\max_{a'} Q(s_{j+k_j}, a'; \theta); \theta^-)$$

  Note that the future value component $Q_{target}(s_{j+k_j}, \dots)$ uses the main action Q-network and its target, reflecting the value of the optimal next primitive action after the skip concludes. The current prediction is $Q_{skip}(s_j, a_j^{behav}, k_j; \theta_{skip})$, where $Q_{skip}$ has parameters $\theta_{skip}$ (which may share some parameters with $\theta$ if weight_sharing = True). The loss is:

$$L_{skip}(\theta_{skip}) = \mathbb{E}_{(\dots) \sim \mathcal{B}_{skip}} \left[ \text{HuberLoss}(y_j^{skip} - Q_{skip}(s_j, a_j^{behav}, k_j; \theta_{skip})) \right]$$

Both loss functions are optimized using their respective Adam optimizers, and gradients are clipped to prevent large updates.

### L.6.2   Target Network Updates

To stabilize training, target networks are employed.

- A target network $Q_{target}$ (with parameters $\theta^-$) is maintained for the primary action Q-network $Q$ (parameters $\theta$). This $Q_{target}$ is always updated using Polyak averaging: $\theta^- \leftarrow \tau\theta + (1 - \tau)\theta^-$.

- If weight_sharing is 'False', the distinct skip Q-network $Q_{skip}$ (parameters $\theta_{skip}$) has its own separate target network, $Q_{skip\_target}$ (parameters $\theta_{skip}^-$). This $Q_{skip\_target}$ is similarly updated using Polyak averaging with $\theta_{skip}$ and $\theta_{skip}^-$.

- If weight_sharing is 'True', the parameters of the skip mechanism are part of the overall network structure whose online parameters are $\theta$ (which includes the shared trunk and potentially specific skip head layers not part of the action head). In this scenario, a separate Polyak update for a distinct $Q_{skip\_target}$ is not performed; target values for the skip component's loss are derived using $Q_{target}$ for estimating future state-action values, as shown in the skip target formula.

The soft update rate $\tau$ and the update frequency target_network_frequency are specified in Table 18.

# M  Proto Sequence Decoder (PSD) Implementation Details

This section outlines the architecture and configuration of the Proto-Sequence Decoder (PSD) module used in our model. It describes the decoder network structure, training objectives, latent space regularization, sequence reconstruction process, and other relevant design choices that enable effective decoding of proto-sequence embeddings into action-sequence.

## M.1  Training Dataset Setup

To ensure that the model learns from meaningful and structured data rather than arbitrary noise, we constructed the training dataset according to the following constraints:

1. **Sequence Length Constraint**: All action sequences have lengths between 1 and $L_{max} = 10$ steps.

2. **Action Diversity Constraint**: Each sequence includes at most **two distinct action types**. For example, valid sequences include $[\text{up}, \text{up}]$ or $[\text{up}, \text{left}]$, whereas a sequence like $[\text{up}, \text{left}, \text{down}]$ is considered invalid.

3. **Switch Constraint**: Each sequence may contain at most **one switch between action types**. For instance, $[\text{up}, \text{left}]$ is allowed, but $[\text{up}, \text{left}, \text{up}]$ is not.

4. **Directional Conflict Constraint**: Sequences cannot include **opposite directions**, such as both up and down, or both left and right.

5. **Avoidance of loops**.

Following these criteria, we generated a total of **400 unique action sequences**. These sequences were one-hot encoded and padded with EOS tokens to a fixed length of $L_{max}$ to suit VAE input requirements. These sequences form the basis of the training data for the Proto-Sequence Decoder (PSD).

## M.2  Model Architecture

The Proto-Sequence Decoder (PSD) is implemented as a Variational Autoencoder (VAE) designed to map proto-sequence embeddings into action sequences of varying length. It is trained using reconstruction loss combined with a Kullback-Leibler divergence (KLD) regularization toward a standard Gaussian prior. The decoder operates on flattened one-hot sequence representations of actions and outputs reconstructed sequences over a predefined action vocabulary.

### M.2.1  Input Representation

Each action in the sequence is represented as a one-hot vector over a vocabulary of size $n_{\text{words}} = 5$ corresponding to {up, down, right, left, eos_token}. The decoder models sequences of up to input_length $= 10$ actions, resulting in an input vector of dimension $10 \times 5 = 50$.

### M.2.2  Encoder Network

The encoder receives a flattened 50-dimensional input vector and passes it through a series of fully connected layers:

1. **Linear Layer 1**:
   - Input size: 50
   - Output size: 32
   - Normalization: InstanceNorm1d
   - Activation: LeakyReLU (slope 0.2)

2. **Linear Layer 2**:

    - Output size: 16
    - Normalization: InstanceNorm1d
    - Activation: LeakyReLU

3. **Linear Layer 3**:

    - Output size: 16
    - Normalization: InstanceNorm1d
    - Activation: Tanh

The output is then projected into two parallel linear layers to produce the latent mean $\mu \in \mathbb{R}^{16}$ and log-variance $\log \sigma^2 \in \mathbb{R}^{16}$. A latent sample $z$ is drawn using the reparameterization trick: $z = \mu + \sigma \cdot \epsilon$, where $\epsilon \sim \mathcal{N}(0, 1)$.

### M.2.3  Decoder Network

The sampled latent vector $z \in \mathbb{R}^{16}$ is decoded through a symmetric feedforward network:

1. **Linear Layer 1**:

    - Output size: 16
    - Normalization: InstanceNorm1d
    - Activation: LeakyReLU

2. **Linear Layer 2**:

    - Output size: 32
    - Normalization: InstanceNorm1d
    - Activation: LeakyReLU

3. **Linear Layer 3**:
    x§

    - Output size: 50 (reconstructed sequence)
    - Normalization: InstanceNorm1d
    - Final Activation: Sigmoid (applied element-wise)

### M.3  Training Objective

The PSD is optimized using a combination of:

- **Reconstruction Loss**: Binary cross-entropy loss between the input sequence and its reconstruction, normalized by sequence length.

- **KL Divergence Loss**: Encourages the latent distribution to match a unit Gaussian prior.

All input sequences are EOS-padded to the maximum length of 10 to ensure uniform input dimensionality across batches.

### M.4  Hyperparameters

The training of the Proto-Sequence Decoder (PSD) is governed by a set of fixed hyperparameters, detailed in Table 19. These parameters control aspects such as optimization, batch processing, and reproducibility.

Table 19: Proto-Sequence Decoder (PSD) Training Hyperparameters

| Parameter | Value |
|---|---|
| train_on_entire_dataset | True |
| seed | 42 |
| optimizer | Adam |
| optimizer_learning_rate | $1 \times 10^{-4}$ |
| optimizer_weight_decay | $1 \times 10^{-3}$ |
| batch_size | 32 |

### M.5 Optimizer

The PSD is trained using the Adam optimizer (`torch.optim.Adam`). The learning rate is controlled by the `learning_rate` hyperparameter (see Table 19).

### M.6 Training Details

#### M.6.1 Loss Function

The Proto-Sequence Decoder (PSD) parameters are updated by minimizing a combined loss:

$$L = L_{\text{rec}} + L_{\text{KL}},$$

where:

- $L_{\text{rec}}$ is the label-smoothed binary cross-entropy over the reconstructed sequence:

$$L_{\text{rec}} = -\frac{1}{T} \sum_{t=1}^{T} \Big[ y_t \log \hat{y}_t + (1 - y_t) \log(1 - \hat{y}_t) \Big],$$

  with $y_t$ replaced by $\tilde{y}_t = y_t(1 - \epsilon) + \frac{\epsilon}{2}$, $\epsilon = 0.1$, and $T = 10$ is the sequence length.

- $L_{\text{KL}}$ is the Kullback–Leibler divergence between the approximate posterior and a unit Gaussian:

$$L_{\text{KL}} = D_{\text{KL}}\big(\mathcal{N}(\mu, \sigma^2) \,\|\, \mathcal{N}(0, I)\big).$$

#### M.6.2 Training Procedure

- **Maximum Steps:** Train for up to 20,000 epochs.

- **Batch Composition:** Split sequences into

  - *Short* ($\leq 5$ actions) and
  - *Long* ($> 5$ actions),

  and sample each batch with a 50/50 ratio of short and long sequences.

### M.7 Evaluation

The Proto-Sequence Decoder's (PSD) performance was assessed on the entire training set using two key metrics. Evaluations were conducted every 50 epochs, and the checkpoint yielding the highest exact match accuracy was retained. After 20,000 epochs, the following results were achieved:

- **Exact Match Accuracy:** This metric measures the proportion of sequences reconstructed with zero errors. The PSD achieved an Exact Match Accuracy of 0.978.

- **Per-Step Accuracy:** This metric calculates the fraction of correctly reconstructed actions across all positions within the sequences. The PSD achieved a Per-Step Accuracy of 0.99.

### M.8   Visualization of the Learned Embedding Space

Figure 6 illustrates the two-dimensional t-SNE projection of the learned proto-action-sequence embeddings. Each point corresponds to one sequence from the dataset, with colors representing the effective sequence length. As can be seen, sequences with similar structural properties tend to form dense clusters, indicating that the embedding space preserves meaningful relationships between sequences.

A particularly noteworthy observation is the position of the red star, which represents a previously unseen sequence not included during training. This sequence is located within the cluster of its closest structural neighbors, suggesting that the learned representation generalizes effectively to new data. In other words, the embedding model is able to position novel sequences near the most similar examples from the training set, supporting its potential for robust retrieval, similarity search, and downstream predictive tasks.

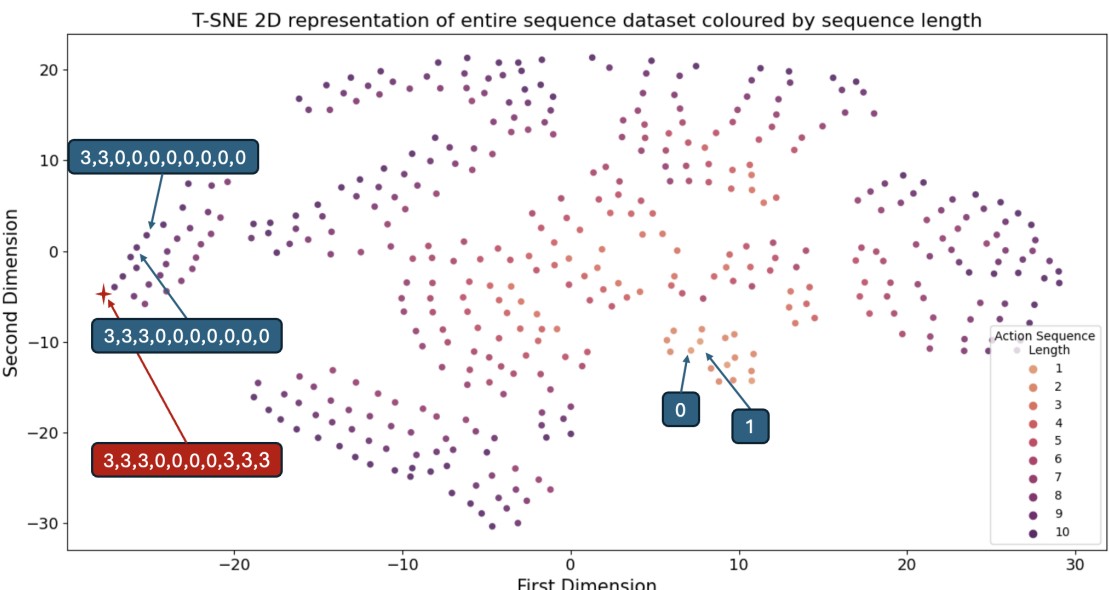

Figure 6: T-SNE 2-D projection of the proto-action-sequence embedding space. The map contains embeddings for the **400 original sequences** in the dataset together with **one previously unseen sequence** (red star). Points are colored by their effective sequence length, and sequences with similar structure form tight neighborhoods. The unseen sequence falls naturally inside the cluster of its closest structural neighbors, showing that the learned representation places **new, out-of-training sequences adjacent to the most similar known examples.**

## N   GPS (Generative Proto Sequence) Implementation Details

This section outlines the architecture and configuration of our GPS (Generative Proto Sequence) method. GPS is an actor-critic based algorithm where the actor network generates a latent representation, termed "proto-sequence." This proto-sequence is then processed by a pre-trained generative decoder model (PSD) to produce a sequence of discrete actions. A critic network evaluates this action sequence to guide learning. The subsequent subsections detail neural network structures, hyperparameter settings, exploration strategy, optimization methods, and other relevant training aspects.

### N.1   Model Architecture and State Representation

The GPS agent consists of three primary neural network components: an Actor, a Critic, and a pre-trained Decoder. It processes observations from the environment. For details on the specific state representation, input shape (e.g., height, width, channels), and preprocessing, please refer to the "State Representation" within Section 4.3.

### N.1.1 Convolutional Neural Network (CNN) Part

Both the Actor and Critic networks use separate but identical Convolutional Neural Network (CNN) architecture to extract features from the input observation. The CNN component used for initial feature extraction is identical to the one in the DQN baseline. For details on its architecture (number of layers, channels, kernel sizes, strides, and activations), please refer to Appendix J.1. The output of this CNN part is a flattened 1D vector of size $n\_flatten$.

Note: in our current implementation, the actor and critic networks each have their own separate CNN state feature extractors. In future work, we plan to explore a shared CNN feature extraction architecture as implemented in TempoRL, which could potentially improve computational efficiency and state representation learning.

### N.1.2 Actor Network

The Actor network takes the extracted features from the CNN and produces a proto-sequence embedding.

- **Input**: The flattened feature vector $n_{\text{features}}$ from the CNN is concatenated with a positional encoding. The embedding dimension of this positional encoding is specified by `pe_embedding_dim`.

- **Architecture**: The combined features are processed through a series of fully connected linear layers, defined by the `actor_linear_layers` parameter (e.g., [512, 128, 32]). The activation function for these hidden layers is specified by `actor_linear_layers_activation_function` which is "leaky_relu" with negative slope 0.1.

- **Output**: The actor generates a single proto-sequence embedding. This embedding is a vector of size `actor_n_output_channels` and serves as input to the Decoder PSD network.

### N.1.3 Position Encoding in Proto-Sequence Generation

To enhance the expressiveness of our action sequence generation, we incorporate positional encoding within the Actor network. This technique, inspired by transformer architectures (Dosovitskiy et al., 2020), helps the Actor generate more contextually aware proto-sequence embeddings by providing explicit spatial information about the agent and goal positions.

**Implementation Details.** Our positional encoding implementation combines both row and column information for each grid cell in the observation space:

1. We create sinusoidal encodings for both dimensions (height and width) separately:

   ```
   pe_row[:, 0::2] = torch.sin(position_row * div_term)
   pe_row[:, 1::2] = torch.cos(position_row * div_term)
   ```

2. These encodings are combined into a unified representation where the first half of each cell's embedding encodes its row position and the second half encodes its column position.

3. During forward passes, we extract the agent and goal positions from the observation and retrieve their respective positional encodings:

   ```
   agent_pe = self.position_encoding[agent_row, agent_col]
   goal_pe = self.position_encoding[goal_row, goal_col]
   ```

4. These position-specific features are concatenated with the CNN-extracted features before being processed by the linear layers of the Actor.

**Motivation and Benefits.** Integrating positional encoding within the Actor network provides several advantages:

Table 20: GPS with vs. without Positional Encoding

| Environment | Positional Encoding | ASR at Training Steps | | | | ASR Converge >0.9 Step | PER | SGF |
|---|---|---|---|---|---|---|---|---|
| | | 100K | 500K | 1M | 1.5M | | | |
| 16x16_obstacles_15% | **With** | 0.22 | 0.96 | 0.99 | 0.96 | **300k** | **0.72** | **10.8** |
| | Without | 0.15 | 0.38 | 0.51 | 0.52 | >1.5M | 0.62 | 17.67 |
| 24x24_obstacles_15% | **With** | 0.12 | 0.45 | 0.80 | 0.91 | **1.5M** | 0.48 | **6.34** |
| | Without | 0.07 | 0.12 | 0.13 | 0.13 | >1.5M | **0.54** | 16.91 |

1. **Enhanced Spatial Reasoning**: By explicitly encoding agent and goal positions, the Actor can better understand spatial relationships, which is crucial for navigation tasks.

2. **Improved Exploration Early in Training**: The position encodings enable the generation of more diverse proto-sequence embeddings in the initial training phases, facilitating better exploration before the CNN features become well-trained.

3. **Direction-Aware Sequence Generation**: The relative positions of agent and goal inform the Actor about the general direction of movement required, allowing it to generate more purposeful action sequences even with limited experience.

4. **Invariance to Visual Feature Quality**: Especially early in training when the CNN features may be unreliable, position encodings provide a stable signal that enables meaningful action sequence generation.

Our manual investigations and targeted experiments suggest that incorporating position encodings enhances the model's capabilities in several ways. We observed that the position-enriched Actor generates proto-sequence embeddings with greater contextual awareness of spatial relationships, which in turn produces more diverse and situation-appropriate action sequences. Without position encoding, the proto-sequence embeddings tended to cluster more closely in the latent space, resulting in less differentiated action patterns. This difference was particularly evident in larger and more complex maze environments, where the position-encoded model demonstrated an improved ability to generate directionally coherent sequences that efficiently navigated toward goals. The positional information appears to provide a structural prior that helps the Actor formulate meaningful navigation strategies even before the CNN features are fully refined through training.

**Quantitative Ablation Study.** To provide empirical evidence for the benefits of positional encoding, we conducted a controlled ablation study comparing GPS with and without positional encoding on two challenging environments. Table 20 presents the comparative results.

The ablation results provide strong quantitative evidence for positional encoding's critical importance. GPS with positional encoding achieves convergence (ASR > 0.9) in 300K-1.5M steps, while the variant without positional encoding fails to converge after 1.5M steps. Performance gaps are substantial: 44 percentage in the 16×16 with 15% obstacles environment (0.96 vs 0.52 final ASR) and 78 percentage in the 24×24 with 15% obstacles environment (0.91 vs 0.13 final ASR). Path efficiency results show mixed patterns, with positional encoding improving PER by 0.10 in the 16×16 with 15% obstacles environment (0.72 vs 0.62) but showing lower PER in the 24×24 with 15% obstacles environment (0.48 vs 0.54). However, this apparent PER advantage for the variant without positional encoding is misleading, as its extremely low ASR (0.13) indicates fundamental task learning failure – it likely only succeeds on the easiest instances, artificially inflating the path efficiency metric. Both environments show substantial reductions in sequence generation frequency with positional encoding: 39% reduction in 16×16 (10.8 vs 17.67 SGF) and 63% reduction in 24×24 (6.34 vs 16.91 SGF), indicating more coherent action sequences requiring fewer decision points.

### N.1.4 Critic Network

The Critic network estimates the Q-value of a state and a decoded action sequence.

- **Input**:
    - The flattened feature vector $n_{\text{features}}$ from the CNN, representing the current state.
    - The action sequence generated by the Decoder from the actor's proto-sequence. The representation used for this action sequence is `ACTION_SEQ_AS_ONE_HOT`. Namely, the input is a tensor where each action in the sequence of length `n_actions_in_seq` is one-hot encoded. Shorter sequences are padded to this length using an End-of-Sequence (EOS) token. Each one-hot vector has a dimension equal to the `action_space_size` plus one (for the EOS token). Consequently, the total input dimension for the action sequence part, `action_seq_dim`, is `n_actions_in_seq` $\times$ (`action_space_size` + 1).

- **Architecture**: The inputs are processed through a series of fully connected linear layers, defined by `critic_linear_layers` (e.g., [512, 128, 32]). The activation function is specified by `critic_linear_layers_activation_function` which is "leaky_relu" with negative slope 0.1.

- **Output**: A single scalar Q-value. The output Q-value can be optionally clipped between `min_qf_value` and `max_qf_value`.

### N.1.5   Decoder Network

A pre-trained generative model, specifically a Variational Autoencoder, acts as the Decoder (PSD).

- **Loading**: The Decoder is loaded from a pre-trained model specified by `decoder_model_path`.

- **Input**: The proto-sequence embedding (size `actor_n_output_channels`) generated by the Actor.

- **Output**: A sequence of `n_actions_in_seq` discrete actions. Each action is selected from a vocabulary of `action_space_size` primitive actions plus an `end_of_sequence_token` token.

- **Generation**: The Decoder can use Gumbel-Softmax for differentiable sampling if `use_gumble_in_decoder` is true, or a deterministic argmax with a Straight-Through Estimator otherwise. See explanation in Section 4.2.

### N.2   Hyperparameters

The GPS method is configured by a wide range of hyperparameters. General experiment settings and environment-specific configurations are typically managed as detailed for the DQN baseline (see Tables 14 and 15). Key algorithm-specific hyperparameters for GPS are listed in Tables 21, 22.

### N.3   Reward Function and Empty Sequence Handling

The GPS agent utilizes a structured reward function designed to encourage efficient navigation while penalizing inefficient or invalid behaviors. The reward function is defined as:

$$R = r_{\text{goal}} - \frac{1}{l_{\text{max}}} \times n_{\text{valid}} - \frac{3}{l_{\text{max}}} \times n_{\text{invalid}}$$

where $r_{\text{goal}}$ is 1 if the agent reached the goal (0 otherwise), $l_{\text{max}}$ is the maximal start-goal distance acting as a regularizer, $n_{\text{valid}}$ is the number of valid actions taken, and $n_{\text{invalid}}$ is the number of invalid actions (e.g., bumping into a wall).

To handle cases during training where the actor generates proto-sequences that are decoded as empty sequences (i.e., where the PSD outputs the EOS token as the first action), we apply a harsh penalty of -20. For these instances, we also hard-code the action to be 1 (DOWN) to ensure the agent always takes some action. This direct negative reinforcement was found to be highly effective in guiding the actor to produce valid proto-sequence embeddings that decode into meaningful action sequences. Without this penalty, the

Table 21: Algorithm Specific Hyperparameters for GPS

| Parameter | Value / Options / Description |
|---|---|
| `total_timesteps` | Env. specific |
| `actor_learning_rate` | Actor LR (e.g., $1 \times 10^{-4}$). |
| `critic_learning_rate` | Critic LR (e.g., $1 \times 10^{-4}$). |
| `buffer_size` | [10000, 50000] |
| `gamma` ($\gamma$) | Discount factor. |
| `tau` ($\tau$) | Target net. soft update rate (e.g., 0.005). |
| `batch_size` | Experiences per train step. |
| `learning_starts` | Timestep train begins. |
| `actor_policy_frequency` | Actor net. update freq. rel. to Critic (e.g., 2). |
| `actor_target_network_frequency` | Target Actor net. update freq. (e.g., 10 steps). |
| `critic_target_network_frequency` | Target Critic net. update freq. (e.g., 10 steps). |
| `start_e` | Initial $\epsilon$ for $\epsilon$-greedy (e.g., 1.0). |
| `end_e` | Final $\epsilon$ value (e.g., 0.1). |
| `total_steps_e` | Timesteps for $\epsilon$ annealing (e.g., 15000). |
| `sub_sequences_move_start_point` | Boolean; varying sub-seq. start points. |
| `sub_sequences_move_end_point` | Boolean; varying sub-seq. end points. |
| `sub_sequences_min_jump_move_start_point` | Min. jump for start-moved sub-seq. gen. |
| `sub_sequences_min_jump_move_end_point` | Min. jump for end-moved sub-seq. gen. |
| `every_one_step_transition_to_buffer` | Boolean; all single-step trans. stored. |
| `actor_n_output_channels` | Proto-action-seq. embed dim. (e.g., 16). |
| `actor_linear_layers` | Actor MLP hidden layer sizes (e.g., [512, 128, 32]). |
| `actor_linear_layers_activation_function` | Actor MLP activation (e.g., "leaky_relu"). |
| `actor_weight_decay` | Actor L2 reg. strength (e.g., $1 \times 10^{-2}$). |
| `pe_embedding_dim` | Positional encoding dim. in Actor (e.g., 16). |
| `critic_linear_layers` | Critic MLP hidden layer sizes (e.g., [512, 128, 32]). |
| `critic_linear_layers_activation_function` | Critic MLP activation (e.g., "leaky_relu"). |
| `critic_weight_decay` | Critic L2 reg. strength (e.g., $1 \times 10^{-3}$). |
| `decoder_model_path` | Path to pre-trained VAE Decoder model. |
| `n_actions_in_seq` | Decoder action seq. length (e.g., 10). |
| `action_space_size` | Unique primitive actions in env. (e.g., 4). |
| `end_of_sequence_token` | Decoder integer token for end of seq. (e.g., 4). |
| `use_gumble_in_decoder` | Boolean; Decoder uses Gumbel-Softmax. |

actor might frequently produce embeddings that map to empty, significantly hampering exploration and learning progress. This approach provides a clear signal to the actor network about the importance of generating proto-sequences that translate to substantive action sequences, accelerating the learning process and improving the overall stability of training. Empirically, we observed that this simple yet effective mechanism substantially reduced the occurrence of empty sequences.

### N.4 Action Selection and Exploration

At each decision step, the Actor network generates a proto-sequence embedding. The Decoder then translates this embedding into a corresponding action-sequence of primitive actions. The agent employs an $\epsilon$-greedy exploration strategy:

- With probability $\epsilon_t$ (where $\epsilon_t$ anneals from `start_e` to `end_e` over `total_steps_e` steps): An exploratory action sequence is selected. This sequence is typically chosen randomly from a pre-defined set of valid action sequences see Section 4.3 for more details.

- With probability $1 - \epsilon_t$ (exploitation): The action sequence generated by the Actor-Decoder pipeline is used for execution.

Table 22: GPS Hyperparameters Settings

| Parameter | Value |
|---|---|
| total_timesteps | Environment specific |
| buffer_size | [10000, 50000] |
| gamma | 0.99 |
| tau | [0.01, 0.005] |
| batch_size | 256 |
| start_e | 1 |
| end_e | 0.1 |
| learning_starts | 1000 |
| actor_learning_rate | [1e-3, 1e-4, 1e-5] |
| critic_learning_rate | 1e-04 |
| actor_policy_frequency | 2 |
| sub_sequences_move_start_point | TRUE |
| sub_sequences_move_end_point | TRUE |
| sub_sequences_min_jump_move_start_point | 1 |
| sub_sequences_min_jump_move_end_point | 1 |
| every_one_step_transition_to_buffer | TRUE |
| actor_target_network_frequency | [10, 100] |
| critic_target_network_frequency | 10 |
| total_steps_e | 15000 |
| actor_n_output_channels | 16 |
| actor_linear_layers | ["512, 32", "512, 128, 32"] |
| actor_linear_layers_activation_function | leaky_relu (negative slope 0.1) |
| actor_weight_decay | 1e-04 |
| pe_embedding_dim | 128 |
| critic_linear_layers | [512, 128, 32] |
| critic_linear_layers_activation_function | leaky_relu (negative slope 0.1) |
| critic_weight_decay | 1e-04 |
| max_level | Environment specific |
| start_level | Environment specific |
| use_gumble_in_decoder | TRUE |

The value of $\epsilon_t$ is linearly annealed:

$$\epsilon_t = \max \left( end\_e, start\_e - (start\_e - end\_e) \cdot (\text{current\_step}/total\_steps\_e) \right)$$

The chosen action sequence is subsequently trimmed using the `end_of_sequence_token` before execution.

## N.5 Optimizer

The Actor and Critic networks are trained using separate Adam optimizers (`torch.optim.Adam`).

- The Actor's optimizer is configured with a learning rate of `actor_learning_rate` and applies L2 weight decay with a coefficient of `actor_weight_decay`.

- The Critic's optimizer uses a learning rate of `critic_learning_rate` and L2 weight decay with a coefficient of `critic_weight_decay`.

## N.6 Replay Buffer

A replay buffer (`ReplayMemory`) with a capacity of `buffer_size` stores past experiences. Each stored transition typically includes: the current state observation ($s_t$), the next state observation ($s_{t+1}$), the selected action sequence (act_seq$_t$), the received reward ($r_t$), a terminal flag ($d_t$), and the actor's proto-action-sequence embedding that generated act_seq$_t$ (emb$_t$). The system may also store sub-sequences derived from executed plans

if parameters such as `push_every_one_step_transition_to_buffer`, `push_sub_sequences_to_buffer_move_start_point`, and `push_sub_sequences_to_buffer_move_end_point` are enabled, potentially enriching the diversity of experiences in the buffer.

### N.7  Training Details

Network training commences after `learning_starts` timesteps have been collected. Updates are performed using batches of `batch_size` experiences sampled from the replay buffer.

#### N.7.1  Critic Network Update

The Critic network parameters ($\theta_C$) are updated by minimizing the Mean Squared Error (MSE) loss:

$$L(\theta_C) = \mathbb{E}_{(s,\text{act\_seq},r,s',d,\text{emb})\sim\mathcal{B}} \left[ (Q(s, \text{emb}, \text{act\_seq}; \theta_C) - y_t)^2 \right]$$

The target Q-value $y_t$ is computed using the target Actor (`target_actor_network`) and target Critic (`target_critic_network`) networks to ensure stability:

$$y_t = r + \gamma(1-d)Q_{\text{target}}\big(s', \text{Actor}_{\text{target}}(s'), \text{Decoder}(\text{Actor}_{\text{target}}(s')); \theta_C^-\big)$$

where $\text{Actor}_{\text{target}}(s')$ is the proto-sequence embedding from the target actor for state $s'$, $\text{Decoder}(\cdot)$ converts it to an action sequence, and $\theta_C^-$ are the parameters of the target critic. The Q-values from the target critic can be clipped using `min_qf_value` and `max_qf_value`.

#### N.7.2  Actor Network Update

The Actor network parameters ($\theta_A$) are updated with a frequency of `actor_policy_frequency` (delayed policy update). The goal is to adjust the actor's parameters to produce a proto-sequence that leads to a higher Q-value as estimated by the current Critic. For a sampled batch of states, the Actor generates a proto-sequence embedding. This is decoded into an action sequence, which is then evaluated by the online Critic network $Q(\cdot; \theta_C)$. The actor loss is designed to maximize this Q-value:

$$L(\theta_A) = -\mathbb{E}_{s\sim\mathcal{B}} \left[ Q(s, \text{Actor}(s), \text{Decoder}(\text{Actor}(s)); \theta_C) \right]$$

#### N.7.3  Target Network Updates

Separate target networks are maintained for both the Actor ($\text{Actor}_{\text{target}}$ with parameters $\theta_A^-$) and the Critic ($Q_{\text{target}}$ with parameters $\theta_C^-$). Their parameters are updated using Polyak averaging with the parameters of their corresponding online networks ($\theta_A, \theta_C$):

$$\theta^- \leftarrow \tau\theta + (1-\tau)\theta^-$$

The soft update rate is $\tau$. Target network updates for the Actor occur every `actor_target_network_frequency` steps, and for the Critic every `critic_target_network_frequency` steps.

#### N.7.4  Training Procedure Summary

- **Initialization**: Networks and target networks are initialized. Replay buffer is empty.

- **Data Collection**: Agent interacts with the environment using the action selection strategy (Section N.4), storing experiences $(s_t, s_{t+1}, \text{act\_seq}_t, r_t, d_t, \text{emb}_t)$ in the replay buffer.

- **Learning Phase** (after `learning_starts` steps):
    1. Sample a `batch_size` of experiences from the replay buffer.
    2. Update Critic network parameters by minimizing the MSE loss with the computed TD targets.
    3. Periodically (every `actor_policy_frequency` steps), update Actor network parameters to maximize the Q-value of the generated sequence as estimated by the Critic.
    4. Periodically (every `actor_target_network_frequency` and `critic_target_network_frequency` steps respectively), update target Actor and target Critic networks using Polyak averaging.

### N.8 Evaluation

The performance of the GPS agent is assessed periodically during training and/or at the end of the training process.

- **Frequency**: Evaluations on validation datasets typically occur every `val_eval_freq` steps, and on subsets of the training data every `train_eval_freq` steps. Less frequent evaluations may occur on a dedicated test dataset (e.g., every `eval_test_dataset_training_freq` steps) or at the end of training.

- **Method**: During evaluation, the actor generates a proto-sequence, the decoder converts it to an action sequence, and this sequence is executed. The Decoder may operate in a deterministic mode (`deterministic_inference = True`).

- **Metrics**: Standard reinforcement learning metrics are logged, such as mean episodic return and success rate. Additional metrics might include the average number of decoder generations per episode or properties of the generated action sequences.

- **Model Saving**: If `save_model` is enabled, the best performing models (actor and critic) are saved based on criteria defined by `save_model_strategy` (e.g., best success rate or mean reward on the validation set).

## O   Maze Evaluation Environments Benchmark

In our research, our evaluation environments consist of procedurally generated mazes with varying structures and complexity. We utilized synthetic maze environments created using Large Language Models (LLMs) to ensure unbiased benchmark construction. The following details our approach to maze generation for the different environment types used in our experiments.

### O.1   Synthetic Maze Generation Process

We generated our maze environments using LLM. For each maze type (rooms, obstacles, and corridors), we provided specific prompts instructing the LLM to generate Python code that would create the maze environments according to our requirements. Importantly, all maze generation was performed programmatically without manual intervention, ensuring reproducibility and eliminating human bias. We specifically used the OpenAI o3-mini model for code generation, which was instructed to create five different variants for each maze type. To avoid experimenter bias in seed selection, we also employed an LLM to generate code for choosing random seeds:

```python
import hashlib
def get_consistent_seed():
"""
Selects a seed number from a list consistently across multiple runs.
Returns:
int: The selected seed number.
"""
seed_list = [42, 1234, 9999, 2024, 2025]

Create a hash of the function name to ensure consistency
hash_object = hashlib.sha256(b'get_consistent_seed')
hash_value = int(hash_object.hexdigest(), 16)

Use the hash value to select a seed from the list
seed_index = hash_value % len(seed_list)
return seed_list[seed_index]
```

```
Get the consistent seed
seed = get_consistent_seed()
print(f"Selected seed: {seed}")
```

The code generated by the LLM for each maze type is available in our code repository. This approach ensured that the maze generation process was fully automated and free from experimenter bias, providing a consistent and fair benchmark for evaluating our GPS algorithm against the baselines.

### O.2  Maze Type Generation Prompts

For each maze type, we provided detailed prompts to the LLM to guide the generation process:

### O.2.1  Obstacles Maze Prompt

**Maze Generation with 15% Obstacles**

Write a Python script that programmatically generates five distinct $16 \times 16$ mazes with randomly placed obstacles. The grid follows these rules:

**Grid & Output Format:**

- The maze is a $16 \times 16$ grid.

- Each cell is either **open space (0)** or a **wall/obstacle (1)**.

- The **outermost border** (first and last rows and columns) **must remain walls (1)**.

- **15% of the inner cells** (excluding the border) should be randomly chosen as **obstacles (1)**, while the remaining are **open spaces (0)**.

- The final maze should be output as a Python dictionary:

```
{
    'maze': [
        "1111111111111111",  # Top row (wall frame)
        "1..............1",  # Use 0 for open spaces;
                             # dots are placeholders here.
        ...
        "1111111111111111"   # Bottom row (wall frame)
    ]
}
```

Replace the dots with appropriate **0s (open spaces)** and **1s (walls/obstacles)**.

**Maze Generation Method: Initialize the Maze:**

- Create a $16 \times 16$ grid where every cell is an **open space (0)**.

- The **outer border (first and last rows/columns) must always remain walls (1)**.

- The **inner** $14 \times 14$ **area** (excluding the border) will contain **open spaces (0) and obstacles (1)**.

**Randomly Place Obstacles:**

- **15% of the inner** $14 \times 14$ **cells** should be converted into **obstacles (1)**.

- The placement of these obstacles should be **random**.

- Ensure that at least one path remains between any two open spaces for potential connectivity.

**Generate Five Distinct Mazes:**

- Use different random seeds to create five unique mazes.

- Ensure that each maze has **exactly 15% obstacles** inside the inner area.

### O.2.2 Rooms Maze Prompt

You are a maze designer responsible for enhancing a predefined base maze structure for a navigation simulation. Your task is to decide where entrances should be located while ensuring the maze meets the following requirements:

**Maze Design Requirements Base Maze Structure:**

- The maze is predefined and consists of a $16 \times 16$ grid.

- The base structure must remain intact, but you will determine the placement of the entrances and ensure connectivity between all rooms and open spaces.

**Entrance and Room Connectivity:**

- The maze is divided into **four equal-sized quadrants (rooms)** separated by walls.

- Each room must have only **two entrances of size 1**. Use a seeded random choice for entrance placement.

- Entrances should be placed strategically to ensure the maze is **fully connected**, meaning an agent can navigate between any two open cells (0) using **up, down, left, or right** movements.

- Passageways between rooms must be **narrow** and preserve the integrity of the maze's challenge.

**Obstacle Coverage:**

- Obstacles (1) must make up **5% of the total grid** ($\approx 13$ cells).

- Don't consider the obstacles that are part of the maze's frame.

- Obstacles may be added or removed **within constraints** to maintain connectivity and alignment with the entrance placement.

- Don't place obstacles in nearby squares close to any entrance. Make sure that an obstacle doesn't block any entrance.

- Use a seeded random choice for obstacles placement.

**Reproducibility:**

- Use a **specific random seed** to ensure the design is reproducible.

**Output Format:**

- Generate code for creating the maze.

- Generate the maze as a Python dictionary with a key (e.g., 'maze') and represent each row as a binary string.

- Output 5 mazes using different seeds and make sure that obstacles don't block entrances.

**Base Maze Layout:** The base structure is as follows:

```
{
    'base_maze': [
        "1111111111111111",
        "1000000100000001",
        "1000000100000001",
        "1000000100000001",
        "1000000100000001",
        "1000000100000001",
        "1000000100000001",
        "1111111111111111",
        "1000000100000001",
        "1000000100000001",
        "1000000100000001",
        "1000000100000001",
        "1000000100000001",
        "1000000100000001",
        "1000000100000001",
        "1111111111111111"
    ]
}
```

**Design Task:** Modify the maze by:

- Placing **entrances** in the walls separating the quadrants.

- The logic should be based on seed for determining the entrances position and obstacles positions.

- Ensuring the maze is **fully connected**.

- Making any minor adjustments to obstacles (1) to meet connectivity and percentage requirements.

- Verify by code that each room must have only **two entrances of size 1**.

- Verify by code that each obstacle doesn't block any entrance.

Output the modified maze as a Python dictionary with the format below:

```
{
    'maze': [
        "updated_row_1",
        "updated_row_2",
        ...,
        "updated_row_16"
    ]
}
```

**General Steps:**

- Choose two entrances for each room by selecting a random square on each wall.

- Randomly select a room and place obstacles within it until the obstacle budget is reached.

### O.2.3 Corridors Maze Prompt

Write a Python script that programmatically generates five distinct $16 \times 16$ mazes. In each maze, start with a grid completely filled with wall cells (represented by 1), then carve out corridors by selecting one or more vertical lines and one or more horizontal lines to convert wall cells to open cells (represented by 0). The corridors will be 1-cell-wide, and they must intersect so that every open cell is reachable from any other via up, down, left, and right moves. The outer border of the maze should always remain as walls.

**Grid & Output Format:**

- The maze is a $16 \times 16$ grid.

- Each cell is either **open (0)** or a **wall (1)**.

- The **outermost border** (first and last rows and columns) **must remain walls**.

- The final maze should be output as a Python dictionary:

```
{
    'maze': [
        "1111111111111111",  # Top row (wall frame)
        "1..............1",  # Use 0 for open spaces;
                             # dots are placeholders here.
        ...
        "1111111111111111"   # Bottom row (wall frame)
```

```
    ]
}
```

Replace the dots with the appropriate 0s and 1s as per the carved corridors.
**Corridor Carving Method: Initialize the Maze:**

- Create a $16 \times 16$ grid where every cell is a **wall (1)**, with the outer border fixed as walls.

**Select Corridor Lines:**

- **Vertical Corridors:** Choose **2 up to 4** vertical columns (not including the outer borders) that will serve as corridors.

- **Restriction:** Ensure that no consecutive vertical columns are selected – there must be at least one wall column between any two chosen corridor columns.

- **Horizontal Corridors:** Choose **2 up to 4** horizontal rows (again, not including the outer borders) that will serve as corridors.

- **Restriction:** Ensure that no consecutive horizontal rows are selected – there must be at least one wall row between any two chosen corridor rows.

- These lines will form a network of corridors that cross each other.

**Carve the Corridors:**

- For each selected vertical column, change all cells in that column (except the outer border) from 1 (wall) to 0 (open space).

- Similarly, for each selected horizontal row, change all cells in that row (except the outer border) from 1 to 0.

- The intersections of these corridors (where a selected vertical column crosses a selected horizontal row) will naturally be open, ensuring connectivity.

**Ensure Full Connectivity:**

- The chosen vertical and horizontal corridors should intersect, guaranteeing that every open cell (in the corridors) is reachable from any other open cell.

- Optionally, you can add additional corridor "branches" (by clearing cells adjacent to the main corridors) to create a more interesting maze layout, as long as all open cells remain interconnected.

**Randomness:**

- Generate five distinct mazes by using different random seeds and varying the selected vertical and horizontal corridor positions.

