# OpenReview forum: "Generative Proto-Sequence: Sequence-Level Decision Making for Long-Horizon Reinforcement Learning"
_TMLR — Accepted by TMLR_

### Review · Reviewer_yP6u · 2025-08-31

**Summary Of Contributions:**

The paper proposes a method that generates a sequence of actions by VAE (encoder is fixed) along with an actor-critic architecture. Authors compared their methods with Deep Q Network (DQN), Dynamic Action Repetition (DAR), and TempoRL (Temporal Reinforcement Learning), and showed that the proposed methods achieve better performance and learns faster than the baselines in the grid environment.

**Audience:**

Yes

**Audience Explanation:**

The topic of long-horizon reinforcement learning has gained increasing attention along with embodied intelligence. Although the authors did not conduct any experiments on robot simulation rather only focused on 2D bird-eye view grid world maze environments, I think it can still gain attention from TMLR’s audience.

**Broader Impact Concerns:**

No broader impact concern that requires a Broader Impact Statement

**Claims And Evidence:**

No

**Claims Explanation:**

* A detailed explanation of the implementation is provided in the appendix.

* The authors briefly mention the rich literature on hierarchical reinforcement learning but do not sufficiently compare their method to existing approaches in either the related works or the experiments section.

* On page 4, the authors claim that generating unseen action sequences is an important aspect of their work. However, any non-rule-based model is expected to generate novel action sequences when presented with novel observations. Therefore, it is unclear why this should be considered a significant contribution.

* The proposed methods, GPS and GPS-D, do not demonstrate improved path efficiency, although they achieve a better success rate. Furthermore, in Section 4.4.2, the authors claim that GPS can self-correct via sequential decision points but provide no supporting evidence for this claim.

* According to the supplementary materials, the authors used only a single, different seed for each experimental method. Since reinforcement learning results are known for their high variance, reporting the mean and standard deviation over multiple seeds is necessary for a robust evaluation.

* The experiments were only conducted in simple, 2D grid-world environments. I recommend that the authors evaluate their method in more complex environments. It is also worth noting that the environment used appears to be a fully observable Markov Decision Process (MDP), not a Partially Observable MDP (POMDP).


Bacon, Pierre-Luc, Jean Harb, and Doina Precup. "The option-critic architecture." Proceedings of the AAAI conference on artificial intelligence. Vol. 31. No. 1. 2017.


Botvinick, Matthew M., Yael Niv, and Andew G. Barto. "Hierarchically organized behavior and its neural foundations: A reinforcement learning perspective." cognition 113.3 (2009): 262-280.


McClinton, Willie, Andrew Levy, and George Konidaris. "Hac explore: Accelerating exploration with hierarchical reinforcement learning." arXiv preprint arXiv:2108.05872 (2021)
.
Osa, Takayuki, Voot Tangkaratt, and Masashi Sugiyama. "Hierarchical Reinforcement Learning via Advantage-Weighted Information Maximization." International Conference on Learning Representations.

**Requested Changes:**

I recommend moving Appendices A-G to the end of the appendix since these are implementation details, while starting from Appendix H, which supplements the experimental results mentioned in the main paper.


Please differentiate \citet and \citep. Also, please change the authors of \citep to just \citet.


In Section 3.1, embedding space $k \in K$: $K$ is not defined, and $k$ is the embedding, not the embedding space mentioned in the previous sentence.


In Figure 2 caption, please add the meaning of each symbol in the figure.

---

> ### Author Response · Authors · 2025-10-01
> **Response to Reviewer**
>
> **Weaknesses:**
>
> **1)	A detailed explanation of the implementation is provided in the appendix.**
>
> As suggested by the reviewer, we re-organized the Appendix. The new order  improves the paper's flow, making the experimental validation materials more accessible while maintaining implementation details for reproducibility. The new order is as follows:
>
> A. Analysis of Reward Strategy Impact on ASR
>
> B. Evaluating the Average Success Rate (ASR) With Larger Train dataset
>
> C.	Evaluation on Stochastic Environments
>
> D.	Evaluation on Partial Observability Environments
>
> E.	Self-Correction Analysis
>
> F.	Computational Cost Analysis
>
> G.	ASR Statistical Significance Testing
>
> H.	Baselines and Architecture
>
> I.	DQN Baseline Implementation Details
>
> J.	DAR Baseline Implementation Details
>
> K.	TempoRL Baseline Implementation Details
>
> L.	Proto Sequence Decoder (PSD) Implementation Details
>
> M.	GPS (Generative Proto Sequence) Implementation Details
>
> N.	Maze Evaluation Environments Benchmark
>
> **2)	The authors briefly mention the rich literature on hierarchical reinforcement…**
>
> We thank the reviewer for raising the point on hierarchical RL. To clarify, GPS is not a hierarchical method: it does not learn high- and low-level policies, discover reusable skills, perform sub-goal decomposition, or use temporal abstractions via learned skill primitives. Instead, GPS directly maps states to complete action sequences in one forward pass, with a pre-trained, frozen Decoder serving as a fixed mapping rather than a lower-level policy.
>
> We cited hierarchical approaches (e.g., TACO-RL, ASPiRe, SHRL) to acknowledge the broader literature on long-horizon decision-making, and Dulac-Arnold et al. (2015) inspired our proto-sequence embedding, not to claim hierarchical lineage.
>
> Our comparisons focused on DAR and TempoRL because (1) they align with GPS’s contribution of generating variable-length action sequences, (2) the hierarchical methods we referenced require continuous actions or pre-learned skills, making comparison infeasible, and (3) GPS differs fundamentally in learning full action sequences without sub-goals or skill decomposition
>
> We revised Section H in the appendix to more clearly delineate GPS from hierarchical RL methods and better justify our experimental scope (page 35, last paragraph of Section H).
>
>
> **3)	..the authors claim that generating unseen action sequences…it is unclear why this should be considered a significant contribution.**
>
> We thank the reviewer for pointing out the need to clarify this point. We do not claim that the Decoder’s ability to generalize is novel or a contribution. We wanted to emphasize the point that GPS can generate novel sequences in order to ensure that this point is clear to the reader. Without this capability – if our Decoder was only able to memorize action sequences – then our approach would not be applicable to domains with large and complex action spaces.
>
> To clarify this point, we have added the following paragraph to Section 3:
> "Despite using a frozen decoder pre-trained on a limited set of sequences, GPS can generate action sequences that differ from those in the decoder's training data. The ability to generalize is important to our method, since sequence memorization is not applicable to domains with large action spaces ."
>
>
> **4)	…the authors claim that GPS can self-correct…but provide no supporting evidence for this claim.**
>
> We thank the revier for suggesting that we quantify GPS’s self-correction. At each decision point, progress toward the goal is tracked via BFS distance; deviations are flagged when (d_after ≥ d_before) , and GPS’s ability to generate corrective sequences over multiple horizons is measured.
>
> Key findings:
>
> a) High immediate correction: 62.4%–99.6% across environments at the next decision point.
>
> b) Environment adaptation: open spaces achieve >89% immediate correction, while rooms/obstacles yield 62%–77% initially but recover within 2–4 sequences.
>
> c) Path efficiency trade-off: GPS favors directional correctness and robustness over strict optimality, improving success via self-correction.
>
> These results substantiate that GPS performs sequence-level closed-loop control, enabling effective course corrections while preserving temporal abstraction.
>
> We made the following revisions in our paper:
>
> a)	We added Section E in our appendix, which contains our analysis of this behavior.
>
> b)	We added the following text to Section 4.4.2, to reference our analysis:
> “Empirical validation of this self-correction capability is provided in Appendix E, demonstrating GPS’s ability to generate correction sequences with high success rates (62.4-99.6% immediate correction) across diverse maze environments)”

---

> ### Author Response · Authors · 2025-10-01
> **Continuation of response**
>
> **5)	…reporting the mean and standard deviation over multiple seeds is necessary...**
>
> We thank the reviewer for rasing this important point. We now use three seeds for each method across all environments.
>
> All main results tables and analyses in the paper have been updated to reflect the use of multiple seeds. We report mean and standard deviation across these runs for all key metrics including Average Success Rate (ASR), Path Efficiency Ratio (PER), Sequence Generation Frequency (SGF), and convergence rates.
>
> *Consistency of Results*: The updated multi-seed results confirm the robustness of our findings. GPS consistently outperforms baselines in most cases with the same general patterns observed in our previously-reported experiments.
> Our updates were performed in the following Sections and Tables: Tables 2-3, Sections 4.4.1-4.4.3.
>
> **Please note**: Due to time constraints, we have not been able to complete all experimental runs.  Specifically, we need to complete the runs for the 24x24 obstacles environment . Our tables have several missing values that will be updated in the next few days. The current incomplete results clearly show that our approach maintains its performance.
>
> **6)	..experiments were only conducted in simple, 2D grid-world environments. I recommend that the authors evaluate their method in more complex environments…**
>
>
> We acknowledge these limitations and updated our submission accordingly. First, we clarified in Section 4.1 that our primary experiments use fully observable MDPs, where the agent has complete visibility of the maze. Second, we added two new experimental setups:
>
> *a) Stochastic environment (sticky actions)*: A 25% chance of repeating the previous action was introduced in four maze environments. Results (Section 5.3, Appendix C, Table 5) show GPS consistently outperforms all baselines, even surpassing baselines trained and evaluated on non-sticky environments.
>
> *b) Partial observability*: Agents only view a 7×7 area around them, instead of full observability. Results (Section 5.4, Appendix D, Table 6) show GPS excels in empty mazes, is comparable in corridors, but underperforms in obstacles and rooms, as expected given the limited information.
>
> Due to time constraints, we are currently reporting partial results. We will add the results for additional environments in the coming days.  Despite this partiality, we report sufficient results to demonstrate clear performance trends: GPS is robust under stochastic dynamics but less effective when observability is restricted.
>
>
> **Requested changes:**
>
> **1) I recommend moving Appendices A-G to the end of the appendix...**
>
> This request has been addressed in our response to #1
>
> **2) Please differentiate \citet and \citep. Also, please change the authors of \citep to just \citet.**
>
> This issue has been resolved.
>
> **3) In Section 3.1, embedding space  is not defined, and  is the embedding, not the embedding space mentioned in the previous sentence.**
>
> We thank the reviewer for pointing out this unclear notation. We have clarified the proto-sequence representation in Section 3.1:
>
> *Revised text*: "The proto-sequence k is a latent embedding of a sequence of actions, represented as a vector in the embedding space k ∈ ℝᵈ, where d is the dimensionality of the embedding space. This representation provides our Actor with significant flexibility, as it can create action sequences of varying length using a fixed-size representation."
>
> **4) In Figure 2 caption, please add the meaning of each symbol in the figure.**
>
> We thank the reviewer for pointing this out. We updated the Figure 2 caption to include symbol meanings:
>
> *Updated caption*: "Examples of our generated mazes (16×16). We use four maze environments (left to right): EMPTY – open space; 15% Obstacles – random obstacle placement; ROOMS – structured rooms with doorways; CORRIDORS – narrow paths requiring precise navigation. Red squares indicate start positions, green squares mark goals, and yellow circles show the current agent position."

---

> > ### Comment · Reviewer_yP6u · 2025-10-09
> >
> > I appreciate the authors for their response and revisions. As Reviewer Vp97 mentioned, the new experiments on robustness, POMDP, and detailed analysis strengthen the paper.

---

### Review · Reviewer_Vp97 · 2025-09-14

**Summary Of Contributions:**

This paper introduces **Generative Proto-Sequence (GPS)**, a novel actor-critic reinforcement learning architecture designed to tackle long-horizon, sparse-reward tasks. The core contribution is a method that generates **variable-length, discrete action sequences in a single, atomic decision step**. Instead of a traditional policy that outputs one action per timestep, the GPS Actor generates a latent "proto-sequence" embedding. This embedding is then translated into an executable action sequence by a pre-trained VAE-based Proto-Sequence Decoder (PSD). The Critic evaluates the value of the entire state-sequence pair, allowing for end-to-end training.

---

**Key Strengths:**
* **Novel Architecture:** The proposed method of generating a complete, variable-length action sequence via a latent embedding in a single forward pass is a novel and elegant approach to temporal abstraction. The end-to-end differentiability, achieved by passing gradients from the Critic through the Decoder to the Actor, is a significant technical strength.
* **Strong Empirical Performance:** The paper provides extensive evidence across a variety of procedurally generated maze environments, demonstrating that GPS consistently and significantly outperforms strong baselines (DQN, DAR, TempoRL) in terms of sample efficiency (faster convergence) and final performance (higher success rates), especially in larger and more complex mazes.
* **Effective Exploration and Credit Assignment:** By making decisions at the sequence level, GPS effectively reduces the decision horizon, which appears to directly address the challenges of temporal credit assignment and strategic exploration in sparse-reward settings. The analysis of subsequence buffering provides direct support for this claim.
* **Generalization:** The paper shows that GPS can generate novel action sequences not seen during the decoder's training, suggesting it learns a structured representation of behaviors rather than simply memorizing effective trajectories. This is a strong argument for the model's ability to generalize.

---

**Key Weaknesses:**
* **Limited Evaluation Domain:** While the maze environments are well-controlled and systematically test navigation, they represent a single domain. The claims of general applicability to long-horizon tasks would be stronger if tested on other benchmark domains.
* **Decoder Pre-training Overhead:** The method requires a separate pre-training phase for the Proto-Sequence Decoder (PSD) on a dataset of synthetic action sequences. A new PSD must be trained for every unique action space, which adds complexity and may limit transferability.
* **Path Efficiency Trade-off:** The main (stochastic) GPS model often achieves a higher success rate at the cost of path efficiency (PER) compared to its deterministic variant (GPS-D) and some baselines. While the authors frame this as a reasonable trade-off for robustness, it is a limitation in scenarios where optimality is critical.

**Audience:**

Yes

**Audience Explanation:**

1.  **Addresses a Fundamental RL Problem:** The paper tackles the long-standing challenges of exploration and credit assignment in long-horizon, sparse-reward environments. This is a core problem in reinforcement learning, and novel methods that show significant progress are of high interest to the research community.
2.  **Novel and Elegant Method:** The concept of generating entire action sequences as single decisions through a latent space is a conceptually clean and powerful idea. It moves beyond simple action repetition or pre-defined skills, offering a more flexible form of temporal abstraction.
3.  **Strong and Clear Results:** The paper presents unambiguous and statistically significant improvements over relevant and well-established baselines on a challenging benchmark. The clarity and strength of the results make the paper's contribution easy to understand and appreciate.
4.  **Sparks Future Research:** The proto-sequence idea opens up several interesting avenues for future work, such as applying it to continuous control, using more powerful generative models for the decoder, or exploring ways to learn the decoder's action vocabulary online. This potential to inspire follow-up research makes it a valuable contribution to the field.

**Broader Impact Concerns:**

I have no broader impact concerns with this work. The research is foundational and focuses on algorithmic improvements for decision-making in simulated environments (mazes). It does not engage with sensitive data, and the potential applications are not directly tied to societal or ethical issues that would require a Broader Impact Statement.

**Claims And Evidence:**

Yes

**Claims Explanation:**

1.  **Claim of Superior Performance:** The primary claim that GPS "consistently outperforms the baselines in most cases" is convincingly demonstrated in **Tables 2 and 3**. In nearly all environments, GPS and its deterministic variant, GPS-D, achieve higher Average Success Rates (ASR) and converge in significantly fewer training steps than DQN, TempoRL, and DAR. The authors validate these results with **McNemar's tests (Appendix J)**, showing that the improvements over all baselines are statistically significant ($p < 0.0001$), which adds a strong layer of confidence to the results.

2.  **Claim of Improved Credit Assignment & Exploration:** This is supported by multiple pieces of evidence.
    * **Indirectly:** The dramatic speed-up in learning (Table 3, "ASR Converge>0.9 Step") is strong evidence that the learning signal is being propagated more effectively.
    * **Directly:** The ablation study on subsequence buffering (**Figure 3**) clearly shows that training the Critic on prefixes and suffixes of action sequences significantly accelerates learning. This technique directly leverages the structure of the sequence-based decisions to provide denser, more effective learning signals, supporting the credit assignment claim.
    * **Exploration:** The lower Sequence Generation Frequency (SGF) in **Table 3** indicates that GPS commits to longer, more coherent action sequences, which prevents dithering and promotes more structured exploration compared to step-by-step methods.

3.  **Claim of Better Generalization:** The ability of the trained agent to achieve high success rates on a held-out test set of unseen mazes is the primary evidence for generalization. Furthermore, the analysis in **Section 5**, which shows the model generating plausible and novel action sequences that were not in the decoder's training set, is a powerful demonstration that GPS learns a flexible, compositional representation in its latent space rather than merely memorizing training trajectories.

**Requested Changes:**

The paper is well-written and the research is solid. The following changes would strengthen it further.

**Critical to Acceptance:**
* None. The paper in its current form meets the bar for acceptance. The claims are well-supported by the evidence provided.

**Strengthening the Work (Recommended):**
* **Discuss Computational Cost:** Please add a brief discussion of the computational trade-offs. While GPS is more sample-efficient, what is the wall-clock time cost? This includes the one-time cost of pre-training the VAE decoder and the per-step inference time of the Actor-Decoder pipeline compared to the baselines. This would provide a more complete picture of the method's practicality.
* **Expand on the Path Efficiency Trade-off:** The results clearly show a trade-off between the stochastic GPS (better ASR, worse PER) and the deterministic GPS-D (often slightly lower ASR, much better PER). This is an interesting finding. I recommend moving a sentence or two from the results section into the main discussion or conclusion to explicitly highlight this "exploration-exploitation" trade-off at the sequence level and provide guidance on when one variant might be preferred over the other.
* **Clarify the Exploration Mechanism:** In Section 4.3, it is stated that exploration uses an $\epsilon$-greedy strategy with "random sequences being sampled from a predefined pool". For clarity, please explicitly state that this pool is the same set of 400 synthetic sequences used to train the PSD, as described in Appendix E.1. This closes the loop for the reader on where these exploratory sequences originate.
* **Position Encoding Ablation:** The use of positional encoding is well-motivated in Appendix F.1.3. However, the authors state that "manual investigations and targeted experiments suggest that incorporating position encodings enhances the model's capabilities". This is a strong claim that would be more convincing with a quantitative ablation study. Adding a small table or plot comparing the ASR of GPS with and without positional encoding on a challenging environment (e.g., 24x24 with obstacles) would provide direct evidence and significantly strengthen this architectural design choice.

---

> ### Author Response · Authors · 2025-10-01
> **Response to Reviewer**
>
> **Weaknesses**
>
> **1)	Limited Evaluation Domain…**
>
> We acknowledge these limitations and updated our submission accordingly. First, we clarified in Section 4.1 that our primary experiments use fully observable MDPs, where the agent has complete visibility of the maze. Second, we added two new experimental setups:
>
> *a) Stochastic environment (sticky actions)*: A 25% chance of repeating the previous action was introduced in four maze environments. Results (Section 5.3, Appendix C, Table 5) show GPS consistently outperforms all baselines, even surpassing baselines trained and evaluated on non-sticky environments.
>
> *b) Partial observability*: Agents only view a 7×7 area around them, instead of full observability. Results (Section 5.4, Appendix D, Table 6) show GPS excels in empty mazes, is comparable in corridors, but underperforms in obstacles and rooms, as expected given the limited information.
>
> Due to time constraints, we are currently reporting partial results. We will add the results for additional environments in the coming days. Despite this partiality, we report sufficient results to demonstrate clear performance trends: GPS is robust under stochastic dynamics but less effective when observability is restricted.
>
> **2) Decoder Pre-training Overhead: The method requires a separate pre-training phase for the Proto-Sequence Decoder (PSD)…**
>
>
> The reviewer is correct in their analysis. While a new Decoder would have to be trained for every unique action space, the same applies to all the components of our proposed approach. Training the Decoder separately has, in our view, several important advantages:
>
> *a)	Simplicity* - training the Decoder in advance reduces the number of “moving parts” in our approach, thus simplifying the training and facilitating faster convergence.
>
> *b)	Robustness* - training the Actor and Decoder together may result in a “moving target” scenario, where both components adapt at the same time, thus harming stability and making it harder for the models to converge [Sutton et al., 1999].
>
> *c)	Preventing mode collapse* - training the Actor and the Decoder together may result in the Decoder focusing on a subset of the action space, making the selection of some actions unlikely in later training steps. By freezing the decoder after training, we ensure it preserves the ability to generate diverse actions.
>
> *d)	Diversity* - by separately training the Decoder, we can ensure that its training set contains diverse sequences of actions. Moreover, by freezing the Decoder we ensure that it does not “forget” how to generate some action types during the training of the other components. This setup also enables us to deploy the same Decoder to every problem with the same action space.Case in point: the same Decoder architecture was used in all experiments, regardless of maze size, including the new stochastic and partial observability experiments described in our response to question #1.
>
> To better explain our reasoning, we elaborated on our reasoning for the separate training in Section 3.2 (last paragraph).
>
> **3)	Path Efficiency Trade-off…a limitation in scenarios where optimality is critical.**
>
> The reviewer is correct in their analysis. While GPS is better at completing larger tasks, this sometimes comes at the cost of optimality. We agree that this trade-off is not suitable for tasks where optimality is critical, but there exist multiple domains where finding a solution quickly or finding one at all is preferable to finding none. This topic has been studied in existing literature, and we update our discussion in **Section 6 (third paragraph)** to address this important point.
>
>
> Furthermore, we updated our future work in Section 6 to specifically mention the need for  automatically adapting our approach so as to handle this trade-off:
>
> *“Finally, we will explore modifications to our approach that will enable us to define desired trade-offs between path optimality and success. By doing so, our aim is to automatically adapt GPS's strategy to match the requirements of various domains.”*

---

> > ### Author Response · Authors · 2025-10-01
> > **Continuation of response**
> >
> > **Strengthening the Work (Recommended):**
> >
> > **1)	Discuss Computational Cost…**
> >
> > We thank the reviewer for raising this important point. *We added Section 5.6* to our paper, which provides a summary of our analysis, and Section F in the appendix, where we provide a more comprehensive analysis.
> >
> > **Summary of our analysis**: While GPS incurs computational overhead, it offers favorable trade-offs in challenging environments:
> >
> > *a)	Pre-training*: VAE decoder requires 10-15 minutes (one-time cost per action space, reusable across environments).
> >
> > *b)	Inference Overhead*: GPS requires 1.6-3.3x wall-clock time per decision vs. baselines due to the Actor-Decoder pipeline. However, since GPS makes decisions less frequently due to its sequence-level abstraction, we measure normalized episode efficiency (total episode wall-clock time divided by optimal path length). In 16x16 environments with 15% obstacles, GPS (4.6ms ± 0.19ms per optimal step) performs comparably to baselines, indicating the per-decision overhead is largely offset by reduced decision frequency. In 24x24 environments, GPS shows higher normalized costs, though this metric favors baselines that only solve easier instances (GPS: 90% success vs. baselines: 23-46%).
> >
> > *c)	Training Convergence*: Despite per-step overhead, GPS achieves faster wall-clock convergence in complex environments due to superior sample efficiency. For example, GPS reaches 90% success in 3.3 hours vs. DQN's 5.83 hours in 16x16 mazes with obstacles.
> >
> > *d)	Practical Impact*: GPS becomes most practical when learning effectiveness is prioritized over raw speed, particularly where baseline methods struggle. The computational cost is offset by substantially superior learning outcomes in challenging scenarios.
> >
> > **2)	Expand on the Path Efficiency Trade-off: …**
> >
> > We thank the reviewer for this valuable suggestion. To address this comment, we have revised the final paragraph of Section 4.4.2 (Path Efficiency Ratio) to explicitly highlight this sequence-level trade-off and provide practical guidance on variant selection.
> >
> > **Revised text:** “These findings reveal a conceptual trade-off in our approach: GPS's stochasticity boosts exploration and rapid convergence to high ASR, while GPS-D's determinism excels in path efficiency once a good policy is learned. This sequence-level exploration-exploitation trade-off offers practitioners a choice between prioritizing solution discovery (GPS) or execution efficiency (GPS-D) based on their specific requirements.”
> >
> > **3)	Clarify the Exploration Mechanism: …**
> >
> > We thank the reviewer for pointing out this ambiguity. We have revised the text in Section 4.3 to explicitly clarify that the exploratory sequences come from the same set used to train the PSD, making the connection clear for readers.
> >
> > **Revised text:** "Exploration employed an ε-greedy strategy, with random sequences being sampled from the same pool of 400 synthetic sequences used to train the PSD (see Appendix E.1 for generation details). This ensures exploratory sequences follow similar structural patterns to those the decoder was trained to generate."
> >
> > **4)	Position Encoding Ablation: The use of positional encoding is well-motivated in Appendix F.1.3. … Adding a small table or plot comparing the ASR of GPS with and without positional encoding on a challenging environment (e.g., 24x24 with obstacles) would provide direct evidence and significantly strengthen this architectural design choice.**
> >
> > We agree with the reviewer that additional analysis was required. Therefore, we conducted the suggested ablation study, comparing GPS with and without positional encoding on challenging environments (16×16 with 15% obstacles and 24×24 with 15% obstacles). We have added a new quantitative ablation study to the appendix with detailed results (*Appendix M.1.3*).
> >
> > In summary, the ablation results provide strong quantitative evidence for positional encoding's critical importance. GPS with positional encoding achieves convergence (ASR > 0.9) in 300K-1.5M steps, while the variant without positional encoding fails to converge after 1.5M steps. Performance gaps are substantial: 44 percent in the 16×16 with 15% obstacles environment (0.96 vs 0.52 final ASR) and 78 percent in the 24×24 with 15% obstacles environment (0.91 vs 0.13 final ASR).

---

> > > ### Comment · Reviewer_Vp97 · 2025-10-06
> > >
> > > Thank you for your detailed response and targeted revisions. The new experiments on stochasticity and partial observability, along with the computational cost analysis and the positional encoding ablation study, are substantial additions that provide valuable empirical support for your method. The clarifications regarding the decoder's design and the path-efficiency trade-off have also effectively addressed my earlier points.

---

### Review · Reviewer_kSJ6 · 2025-09-17

**Summary Of Contributions:**

This work proposes a new method to generate action sequences for reinforcement learning rather than predicting individual actions.

Strengths:
- clear writing
- the model is compared to strong baselines

Weaknesses:
- The environment used is increadibly simple. Specifically, it does not feature any stochasticity. This means that the agent can always predict the maximal number of steps and does not need to trade off prediction horizon against accuracy. I recommend trying this work on more standard control tasks from e.g. gymnasium.
- The split of pretraining the decoder and training the policy is poorly motivated: pretraining the decoder only works if one has very similar structure and action spaces since otherwise it cannot even be guaranteed that a specific action sequence is generatable. In fact, one cannot guarantee that the space reachable by the decoder is spanning the complete set of all A^K action-sequences.
- The reward function used looks very "reward shaped", and - more importantly - is not markov since the reward at state $s_k$ depends on all previous steps $s_0,\dots,s_k$.
- The paper, to our knowledge, does not list the reward and only displays surrogate metrics. Ultimately the reward would be the interesting part since this is the only thing directly selected for by the RL agent, while everything else depends on tenous connections between reward and surrogate and the implicit biases encoded in the metrics. This means even if this method works well in this setting it just might be due to implicit biases favouring the GPS module which would not generalize to other domains
- Multi-step prediction is poorly motivated: Of course receding horizon methods used in MPC have advantages due to being able to anticipate the future, but if one has to commit to a sequence _without_ being able to update it later this actively reduces the amount of information available to the agent. In some sense, this is the opposite of MPC since the agent has to be able to perfectly anticipate the future states K steps into the future without being able to re-ground itself. This only works in simple, completely determinstic settings: If the environment is not deterministic, one might get into an unanticipated state and continue from there without knowing it. If the environment is too complex, the model will be unable to capture it (it will become stochastic from the model's POV). Any sort of partial observability is also expected to break this. In general, since we do not have a receding horizon the problem you end up modelling is over an augmented MDP with size state S and action space $A^K$ and transition function $T^K=T\circ T\circ\dots\circ T$ (with $K$ being the stepsize). It is unclear why this problem would every be easier than the original MDP. I recommend improving the motivation.

**Audience:**

Yes

**Audience Explanation:**

The composition of actions as part of the network output is theoretically interesting.

**Broader Impact Concerns:**

We do not have broader impact concerns

**Claims And Evidence:**

No

**Claims Explanation:**

Only deterministic environments are being tested. No reward statistics are disclosed. Lack of usage of standardized benchmarks.

**Requested Changes:**

- benchmark other environments (particularly stochastic ones)
- adress the motivation problem discussed in the weaknesses
- discuss why the decoder has to be pretrained
- disclose the reward (behaviour over time and at the end of training)
- use a markov reward formulation

---

> ### Author Response · Authors · 2025-10-01
> **Response to Reviewer**
>
> **Weaknesses:**
>
> **1)	The environment used is incredibly simple… does not feature any stochasticity…**
>
> We acknowledge these limitations and updated our submission accordingly. First, we clarified in Section 4.1 that our primary experiments use fully observable MDPs, where the agent has complete visibility of the maze. Second, we added two new experimental setups:
>
> *a) Stochastic environment (sticky actions)*: A 25% chance of repeating the previous action was introduced in four maze environments. Results (Section 5.3, Appendix C, Table 5) show GPS consistently outperforms all baselines, even surpassing baselines trained and evaluated on non-sticky environments.
>
> *b) Partial observability*: Agents only view a 7×7 area around them, instead of full observability. Results (Section 5.4, Appendix D, Table 6) show GPS excels in empty mazes, is comparable in corridors, but underperforms in obstacles and rooms, as expected given the limited information.
>
> Due to time constraints, we are currently reporting partial results. We will add the results for additional environments in the coming days. Despite this partiality, we report sufficient results to demonstrate clear performance trends: GPS is robust under stochastic dynamics but less effective when observability is restricted.
>
> **2)	The split of pretraining the decoder and training the policy is poorly motivated..**
>
> The reviewer raises an important point. While training end-to-end is certainly a commonly used option, we had several reasons for our decision:
>
> *1)	Simplicity* - training the Decoder in advance reduces the number of “moving parts” in our approach, thus simplifying the training and facilitating faster convergence.
>
> *2)	Robustness* - training the Actor and Decoder together may result in a “moving target” scenario, where both components adapt at the same time, thus harming stability and making it harder for the models to converge [Sutton et al. (1999)].
>
> *3)	Preventing mode collapse* - training the Actor and the Decoder together may result in the Decoder focusing on a subset of the action space, making the selection of some actions unlikely in later training steps. By freezing the decoder after training, we ensure it preserves the ability to generate diverse actions.
>
> *4)	Diversity* - the reviewer is correct in stating that we have no guarantees of our Decoder being able to generate all action sequences. However, such a guarantee also would not exist if we trained all components at the same time. By separately training the Decoder, we can ensure that its training set contains diverse sequences of actions. Moreover, by freezing the Decoder we ensure that it does not “forget” how to generate some action types during the training of the other components.
>
> *5)	Transferability* - the reviewer is correct in stating that the Decoder would have to be re-trained for tasks with different action spaces. However, that is true for all components of our model. Separating the training ensures that we can use the same Decoder for all tasks with identical action spaces (e.g., environments with four basic movement actions). Case in point, the same Decoder architecture was used in all experiments, regardless of maze size, including the new stochastic and partial observability experiments described in our response to question #1.
>
> To better explain our reasoning, we updated our reasoning for the separate training in Section 3.2 (the last paragraph).
>
> **3)	...cannot guarantee that the space reachable by the decoder is spanning the complete set...**
>
> The reviewer is correct that such a guarantee does not exist in our approach. However, the same is true for any other deep learning-based solution. We argue that training the Decoder separately actually increases the likelihood of this component maintaining its ability to generate diverse sequences. By providing a diverse training set to the decoder (see Section 5.1), and then “freezing” the model, we prevent the possibility of the Decoder “forgetting” to generate certain kinds of trajectories.
>
> In Section 5.1, we present an analysis of GPS’s ability to generate novel action sequences (i.e., sequences that were not in its training set). An expansion of this analysis is presented in *Section L.8 in the appendix*.
>
> To clarify the important point raised by the reviewer, we revised the first paragraph of Section 5.1:
>
> *“While we provided our Decoder with a diverse training set, the latter did not include all possible trajectories. Our reasons were twofold. First, while including all possible action combinations was feasible in our (relatively small) action space, doing the same for larger, more complex action spaces would be infeasible or very costly. Secondly, we wanted to evaluate GPS’s ability to generalize and produce trajectories that were not in the training set. We consider the ability to generalize important, because the lack of it may limit the usefulness of our approach in large action spaces.”*

---

> > ### Author Response · Authors · 2025-10-01
> > **Continuation of response**
> >
> > **4)	The reward function..is not markov..**
> >
> > We thank the reviewer for pointing out the need to clarify this point. This concern stems from our presentation of aggregated rewards over entire action sequences, which may appear non-Markovian. However, the underlying step-level rewards are fully Markovian, and the apparent non-Markovian nature comes from standard RL practice when dealing with multi-step actions or temporal abstractions.
> >
> > To address this comment, we clarified the Markovian nature of our reward function in Section 4.3 by modifying the reward function description:
> >
> > *"We define this reward function identically across all methods (GPS, DQN, TempoRL, and DAR) where rgoal is 1 if the agent reached the goal (0 otherwise), lmax is the maximal start-goal distance acting as a regularizer, nvalid is the number of valid actions taken, and ninvalid is the number of invalid actions (e.g., bumping into a wall). The underlying step-level reward computation is fully Markovian: each step returns +1.0 for reaching the goal, -1.0/lmax for valid moves, or -3.0/lmax for invalid actions (wall collisions), depending only on the current state and action. When GPS executes a sequence of length L, we aggregate these Markovian step rewards: R_total = Σ(r_t) for t=1 to L. This reward structure encourages goal achievement while penalizing excessive steps and invalid actions, ensuring fair comparison across all methods."*
> >
> > **5)	…does not list the reward and only displays surrogate metrics…**
> >
> > This point is strongly connected to the one above. We revised our paper according to the reviewer’s suggestions to ensure that our reward function is well presented to the reader. We made the following revisions:
> >
> > *a)	We revised and expanded the reward function description in Section 4.3*:
> >
> > "We define this reward function identically across all methods (GPS, DQN, TempoRL, and DAR) where rgoal is 1 if the agent reached the goal (0 otherwise), lmax is the maximal start-goal distance acting as a regularizer, nvalid is the number of valid actions taken, and ninvalid is the number of invalid actions (e.g., bumping into a wall). The underlying step-level reward computation is fully Markovian: each step returns +1.0 for reaching the goal, -1.0/lmax for valid moves, or -3.0/lmax for invalid actions (wall collisions), depending only on the current state and action. When GPS executes a sequence of length L, we aggregate these Markovian step rewards: R_total = Σ(r_t) for t=1 to L. This reward structure encourages goal achievement while penalizing excessive steps and invalid actions, ensuring fair comparison across all methods."
> >
> > *b)	We revised our description of our evaluation metrics in Section 4.3*:
> >
> > “These metrics are complementary, as they allow us to evaluate the policy’s effectiveness, efficiency, and decision frequency, under identical reward optimization across all approaches”
> > These changes emphasize that all methods optimize the same objective function, ensuring that any performance differences stem from architectural approaches rather than reward design biases.
> >
> > **6)	Multi-step prediction is poorly motivated…**
> >
> > The reviewer raises a point that is at the core of our approach: the trade-off between maximizing the available information on the one hand, and enabling the agent to generalize over long sequences on the other. This tradeoff is the reason for our use of Path Efficiency Ratio (PER) metric in our evaluation.
> >
> > We agree with the reviewer that evaluation in additional settings is required to fully assess the merits of our approach. We have revised our paper in the following ways:
> >
> > ●	We address this by introducing two additional experimental setups as been has been addressed in our response to #1.
> >
> > ●	We added the topic of partial information to our discussion in Section 6 (“Conclusions, Limitations, and Future Work):
> > “Furthermore, our analysis suggests that a partial information setting (e.g., limited visibility) sometimes reduces the effectiveness of our proposed approach.”
> >
> > ●	We revised our description of our planned future work in Section 6 to describe our plans to comprehensively address partial information and stochastic settings. The revision is in the beginning of the last paragraph.
> >
> > **Requested changes**
> >
> > **1)	Benchmark other environments (particularly stochastic ones)**
> >
> > This request has been addressed in our response to #1.
> >
> > **2)	Address the motivation problem discussed in the weaknesses**
> >
> > This request has been addressed in our response to #2.
> >
> > **3)	Discuss why the decoder has to be pretrained**
> >
> > This request has been addressed in our response to #2.
> >
> > **4)	Disclose the reward (behaviour over time and at the end of training)**
> >
> > This request has been addressed in our responses to #4 and #5.
> >
> > **5)	Use a markov reward formulation**
> >
> > This request has been addressed in our responses to #4 and #5.

---

> > > ### Comment · Reviewer_kSJ6 · 2025-10-09
> > >
> > > Thank you for your thorough additional work. However, I still question the choice to pretrain the decoder.
> > >
> > > The problem with pretraining the decoder is that one can easily sneak in additional information into the model: While few other papers use such a decoding structure, this is well known when using pretrained embeddings for vision (Parisi et al., "The Unsurprising Effectiveness of Pre-Trained Vision Models for Control" shows 2-4x performance when using a pretrained model).
> > > Of course the advantage in your method would be smaller, but that doesn't mean it's zero. For instance, if you train on feasible action sequences you automatically encode a notion of feasibility into the decoder. I.e. even if you might be technically able to produce any action sequence, your pretraining biases towards specific types of trajectories (even having diverse sequences is such a bias: you implicitly encode information that such sequences exist)
> > >
> > > The normal argument for pre-training is that you have a common data source (e.g. images) for which you can train a single good representation ahead of time to save yourself some effort when moving to RL (or supervised learning). However, in your case the decoder is really specific, so you don't have the same advantage of being able to continuously re-use the decoder.
> > >
> > >
> > > Regarding the reward function: Maybe it would be easier to just write the reward function as a big "choice":
> > >
> > > $$\\begin{cases} 1 & \\text{if goal is reached}\\\\ -1/l_{max} & \\text{if a valid action is executed}\\\\ -3/l_{max} & \\text{if an invalid action is executed}\\end{cases}$$
> > >
> > > I think that would be more understandable.
> > >
> > > Finally, regarding stochasticity: I don't think sticky actions are enough. If you have a very empty gridworld, then sticky actions barely matter (you can split your movement into 3 or 4 cardinal direction moves. repeats barely matter). If you have a really constrained gridworld, then sticky actions also barely matter since any sticky action has a high liklihood of walking "into" a wall, in which case the movement is void. If you have e.g. the corridors setting, sticky action don't do anything or might even be adventagous when walking through a long corridor.
> > >
> > > Maybe dropping actions could also help? The original paper proposing sticky actions also listed "Uniformly random action noise" (i.e. have a chance an action gets replaced by a random one) as an option.

---

> > > > ### Author Response · Authors · 2025-10-15
> > > > **Response to Reviewer**
> > > >
> > > > We thank the reviewer for their useful comments, as well as time and effort.
> > > >
> > > > 1) **"The problem with pretraining the decoder is that one can easily sneak in additional information into the model...your pretraining biases towards specific types of trajectories (even having diverse sequences is such a bias: you implicitly encode information that such sequences exist)"**
> > > >
> > > > We thank the reviewer for raising this important point. We have not previously considered the possibility that pre-training our Decoder could enhance the performance of our proposed approach, or give it an advantage compared to other approaches. As we explained in our earlier response, our reasons for training the Decoder separately were simplicity, reducing the chances of mode collapse and transferability (i.e., using the same Decoder for all maze sizes and setups, including the new stochastic setups).
> > > >
> > > > To address the concerns raised by the reviewer, we re-ran our experiments and compared the performance of our modular GPS to one that was trained end-to-end. Our results, presented in detail in Section G and Table 10 in the Appendix, can be summarized as follows:
> > > >
> > > > a) The overall ASR performance (percentage of successfully completed mazes) of the two variants is nearly identical.
> > > >
> > > > b) The end-to-end variant of GPS achieves a slightly higher PER (meaning its path is a little closer to the optimal path). This is likely due to the model’s ability to modify the generated trajectories so that they are a better fit for specific maze types.
> > > >
> > > > c) The slight improvement in PER comes at the cost of more frequent sequence generations, i.e. the end-to-end GPS produces shorter sequences on average.
> > > >
> > > > Our analysis enables us to rule out the possibility that GPS’s superior performance is the result of the additional information we provided to the Decoder during pre-training.
> > > >
> > > > In response to the reviewer’s comment, we have made the following revisions to our submission:
> > > >
> > > > a) Added Section G, which consists of the analysis summarized above.
> > > >
> > > > b) Referred to Section G at the end of Section 3.2:
> > > > *“It should be noted that training GPS end-to-end achieves comparable final results, with some differences in path optimality and efficiency (see Section G in the Appendix).”*
> > > >
> > > > c) Updated our future work in Section 6 to include the possibility of using more advanced pre-training strategies for domains with large and complex action-space:
> > > > *“As part of this research direction, we plan to explore advanced initialization strategies for our Decoder, so that our approach can more efficiently explore large action spaces.”*
> > > >
> > > > **2) Regarding the reward function: Maybe it would be easier to just write the reward function as a big "choice"...**
> > > >
> > > > We thank the reviewer for this suggestion. We have revised Section 4.3 to present the reward function as a conditional statement for improved clarity. Our revised text is as follows:
> > > >
> > > > *"We define the reward function identically across all methods as follows:
> > > > $$R = \begin{cases} 1 & \text{if goal is reached} \ -1/l_{max} & \text{if a valid action is executed} \ -3/l_{max} & \text{if an invalid action is executed} \end{cases}$$
> > > > where l_max is the maximal start-goal distance (see Table 1) acting as a regularizer. When GPS executes a sequence of length L, we sum these Markovian step rewards: R_total = Σ(r_t) for t=1..L. This reward structure encourages goal achievement while penalizing excessive steps and invalid actions, ensuring fair comparison across all methods."*
> > > >
> > > > This presentation more clearly shows:
> > > >
> > > > a) The Markovian nature of the underlying reward function
> > > >
> > > > b) The three distinct cases that determine the reward at each step
> > > >
> > > > c) How sequence-level rewards are derived from step-level rewards through simple summation
> > > >
> > > > **3) Finally, regarding stochasticity: I don't think sticky actions are enough...Maybe dropping actions could also help? The original paper proposing sticky actions also listed "Uniformly random action noise"**
> > > >
> > > > Based on the reviewer’s suggestion, we performed a set of experiments in another stochastic setting. We implemented the “uniform random action noise” proposed by the reviewer: for every action, there is a 25% chance of it being replaced by a randomly-selected action. This setup is indeed more challenging, which is reflected in the lower performance of the evaluated algorithms compared to the “sticky actions” setup.
> > > >
> > > > The results of our experiments are discussed in Section 5.3 and presented in full in Section C and Table 6 in the Appendix. GPS is the top performer in all evaluated environments.

---

### Decision · Action_Editor_7xpe · 2025-10-25

**Recommendation:** Accept with minor revision

**Additional Comments:**

In my opinion, the paper tackles a relevant problem in RL with an interesting solution, which has some relation with previous research but seems to have important novel aspects. The reviewers are also unanimously favoring acceptance, noting the quality of the research in general and praising the effort the authors made in addressing their initial concerns about the limited empirical evaluation. I do not see any main reason against the paper and I am happy to propose acceptance.

I am still flagging the paper for minor revision as I think that further action on the aspects above could make the submission even more compelling:
- The experiments are conducted over three seeds. If the authors can still dedicate some computational resources to this submission, I would suggest them to run a few additional seeds to support the statistical significance of the findings;
- The discussion of related works is already substantial, but I feel some recent trends in imitation learning and robotics (action chuncking transformers and diffusion policies referenced above) could also be mentioned to further motivate the research. In the same bucket of action repetition, the following works could also be mentioned:
   - https://proceedings.neurips.cc/paper/2020/hash/216f44e2d28d4e175a194492bde9148f-Abstract.html
   - https://ojs.aaai.org/index.php/AAAI/article/view/26156

**Audience:**

Yes

**Audience Explanation:**

I think the paper covers an interesting topic in RL, which may interest a substantial portion of the corresponding community. Although action repetition and generating action sequences is not new in RL and a standard method in imitation learning (see action chuncking transformers https://arxiv.org/abs/2304.13705 and diffusion policies https://journals.sagepub.com/doi/full/10.1177/02783649241273668 https://arxiv.org/abs/2409.00588) it is still not that common in practical RL, which may indicate that further research is relevant.

**Claims And Evidence:**

Yes

**Claims Explanation:**

All the reviewers agree that the claims made in the submission are sufficiently supported. While the manuscript indeed provides empirical evaluation to support the claims, the authors may consider rephrasing parts of the abstract/introduction to account for the limited set of considered domains (e.g., "extensive evaluations") and the weak statistical significance from averages of three runs (e.g., "consistently outperforms").

---

> ### Author Response · Authors · 2025-11-18
> **Response to the AE and a request for a short extension**
>
> We sincerely thank the reviewers and action editor for their constructive feedback and for recommending to accept our paper.
>
> Following the Action Editor’s decision, and before we address each of the points they raised, we have a request and a question:
>
> 1) **Request for extension:** as requested by the AE, we increased the number of seeds in our experiments. While all previously reported results currently remain correct, we require a little more time to complete all our runs (see extended description below). Therefore, we respectfully request an **extension of 10 days** to the original deadline (until 5th of December).
>
> 2) **Question about anonymity:** should the revised manuscript we submit remain anonymous, or may we de-anonymize it at this stage? This question relates to the revision that contains the changes requested by the AE, as well as the required presentation and all other materials.
>
> Thank you for your consideration.
>
> The Authors
>
> Below we address each of the points raised by the Action Editor.
> 1) *“The experiments are conducted over three seeds…”*
>
> We thank the editor for the useful suggestion. We increased the number of seeds to five, and are currently in the process of running the additional experiments. Due to the large number of runs required  (9 maze environments × 5 methods × hyperparameter configurations), we requested an extension of the submission deadline.
>
> It is important to note that, based on the additional partial results we already have, all previously reported results and conclusions remain unchanged.
>
> 2) *“The discussion of related works is already substantial, but I feel some recent trends…”*
>
> We thank the AE for the useful suggestion. We have the following revisions to our submission:
>
> a) In Section 2.1 (Temporal Abstraction Through Action Repetition) we added the following paragraph:
>
> "Other recent work includes multi-frequency control, where Lee et al. (2020) introduce Action-Persistent Actor-Critic (AP-AC) to handle cases where different action dimensions persist for different durations (e.g., repeating leg movements every two steps while repeating arm movements every four steps). Additionally, Sabbioni et al. (2023) introduce the All-Persistence Bellman Operator to enable agents to learn from experiences collected with varying repetition durations simultaneously, by decomposing and bootstrapping across different temporal scales. They provide theoretical guarantees of convergence and empirically demonstrate improved exploration and sample efficiency through their ability to reuse experience collected at any repetition duration. Despite showing promise, these studies.."
>
> b) In Section 2.2 (Multi-Step Action Sequence Generation), we revised and added the following text:
>
> b.1) “Beyond single-action repetition, several methods focus on generating and partially committing to multi-step action sequences—an approach often called action chunking, where multiple consecutive actions are predicted simultaneously rather than one at a time”
>
> b.2) “Action chunking approaches have also been explored in imitation learning, where methods like Action Chunking Transformers (ACT) (Zhao et al., 2023) and Diffusion Policy (Chi et al., 2025) generate multi-step behaviors from expert demonstrations."
>
> 3) *“The authors may consider rephrasing parts of the abstract/introduction to account…”*
>
> We thank the editor for this feedback. In response to the editor's comment, we have reviewed the entire paper and revised similar claims throughout to use more measured and precise language. Below are **representative examples** from key sections:
>
> *Abstract revisions:*
>
> a) Changed "Extensive evaluations" → "Evaluations across diverse maze navigation tasks"
>
> b) Changed "consistently outperforms" → "outperforms leading action repetition and temporal methods in the large majority of tested configurations"
>
> *Introduction section revisions:*
>
> a) Changed "We evaluated GPS on a large set of" → "We evaluated GPS on a set"
>
> b) Changed “Our results demonstrate that GPS consistently learns more efficiently” → “Our results demonstrate that GPS learns more efficiently across most configurations”
>
> c) Changed “significantly outperforms” → “outperforms”
>
> d) Changed “superior generalization and faster convergence” → “improved generalization and faster convergence”
>
> e) Changed “We provide extensive empirical results on challenging maze benchmarks” → “We provide empirical results on challenging maze benchmarks”
>
> f) Changed “showing significant improvements" → “showing improvements”
>
> g) Added explicit acknowledgment of our experimental scope: "faster convergence in maze navigation tasks"
>
> *Conclusion section revisions:*
>
> Changed "Our evaluation shows GPS consistently surpasses leading…" → "Our evaluation on maze navigation tasks shows GPS surpasses leading action repetition and temporal methods across the majority of tested configurations…"

---

> > ### Comment · Action_Editor_7xpe · 2025-11-18
> >
> > Dear Authors,
> >
> > 1. I understand that running the additional seeds may take some time. We granted the extension til December the 5th.
> >
> > 2. The revised manuscript shall be intended as the final camera-ready version. As such it has to be de-anonymized.
> >
> > I have gone through the planned changes briefly. It looks like you are on the right direction for incorporating reviewers feedback. Looking forward to see the final versione.
> >
> > Best regards,
> >
> > AE

---

> > > ### Author Response · Authors · 2025-12-05
> > > **Camera-ready manuscript submitted**
> > >
> > > Dear AE,
> > >
> > > We have submitted the revised (camera ready) version of our manuscript. All comments and requests have been addressed, and we also uploaded our video and code.
> > >
> > > We would like to again thank the reviewers and AE for their thoughtful and useful feedback.
> > >
> > > Best regards,
> > > The Authors

---

> > > > ### Comment · Action_Editor_7xpe · 2025-12-10
> > > >
> > > > Dear Authors,
> > > >
> > > > Thank you for submitting a revised version of the manuscript. Happy to see reviewers' suggestions have been incorporated, a copule of seeds have been added to the experimental results, and the related work discussion is even more complete.
> > > >
> > > > Just a few minor notes:
> > > > - The entries for Seo & Abbeel and Zhang, Xu, Yu appear twice in the references.
> > > > - Consider mentioning also https://proceedings.mlr.press/v119/metelli20a/metelli20a.pdf in the related works.
> > > >
> > > > Please resolve before I accept the camera-ready version.
> > > >
> > > > Best regards,
> > > >
> > > > AE

---

> > > > > ### Author Response · Authors · 2025-12-10
> > > > > **Minor corrections to the camera-ready version**
> > > > >
> > > > > Dear AE,
> > > > >
> > > > > Thank you for your useful feedback.
> > > > >
> > > > > We revised our submission in accordance with the suggestions listed above:
> > > > >
> > > > > 1) The redundant entry for Seo & Abbeel and Zhang, Xu, Yu was removed.
> > > > >
> > > > > 2) We added the suggested citation to our related work (Section 2.1):
> > > > >
> > > > > "Metelli et al. (2020) instead study action persistence in batch reinforcement learning, treating repetition duration as an environmental configuration parameter that modifies the effective control frequency. They introduce Persistent Fitted
> > > > > Q-Iteration (PFQI), which learns value functions for different fixed repetition factors from a single batch
> > > > > of experience and provides theoretical bounds on the performance loss induced by persistence as well as
> > > > > a heuristic for selecting an approximately optimal repetition duration."
> > > > >
> > > > > The revised version has been uploaded.
> > > > >
> > > > > Please let us know if you have additional questions or suggestions.
> > > > >
> > > > > Best regards,
> > > > >
> > > > > The Authors

---

> > > > > > ### Comment · Action_Editor_7xpe · 2025-12-11
> > > > > >
> > > > > > Dear Authors,
> > > > > >
> > > > > > Thanks for updating the manuscript once again.
> > > > > >
> > > > > > Just a final comment: The appendix can be attached to the main paper if you wish to. I suppose it is not mandatory, so splitting in two files as you did is also accepted. Can you please send me a confirmation that you intend to have the appendix in the separate file (or update the manuscript with the appendix)?
> > > > > >
> > > > > > Best regards,
> > > > > >
> > > > > > AE

---

> > > > > > > ### Author Response · Authors · 2025-12-11
> > > > > > > **Minor corrections to the camera-ready version**
> > > > > > >
> > > > > > > Dear AE,
> > > > > > >
> > > > > > > We uploaded a new version of our manuscript, where the main paper and appendices are in the same file. Therefore, our submission now consists of a single file without supplementary material.
> > > > > > >
> > > > > > > Best regards,
> > > > > > >
> > > > > > > The authors